# Ballistic macroscopic fluctuation theory

Benjamin Doyon[1], Gabriele Perfetto[2], Tomohiro Sasamoto[3], Takato Yoshimura[4,5]

**1** Department of Mathematics, King's College London, Strand, London WC2R 2LS, United Kingdom
**2** Institut für Theoretische Physik, Eberhard Karls Universität Tübingen, Auf der Morgenstelle 14, 72076 Tübingen, Germany.
**3** Department of Physics, Tokyo Institute of Technology, Ookayama 2-12-1, Tokyo 152-8551, Japan
**4** All Souls College, Oxford OX1 4AL, U.K.
**5** Rudolf Peierls Centre for Theoretical Physics, University of Oxford, 1 Keble Road, Oxford OX1 3NP, U.K.

November 30, 2022

## Abstract

We introduce a new universal framework describing fluctuations and correlations in quantum and classical many-body systems, at the Euler hydrodynamic scale of space and time. The framework adapts the ideas of the conventional macroscopic fluctuation theory (MFT) to systems that support ballistic transport. The resulting "ballistic MFT" (BMFT) is solely based on the Euler hydrodynamics data of the many-body system. Within this framework, mesoscopic observables are classical random variables depending only on the fluctuating conserved densities, and Euler-scale fluctuations are obtained by deterministically transporting thermodynamic fluctuations via the Euler hydrodynamics. Using the BMFT, we show that long-range correlations in space generically develop over time from long-wavelength inhomogeneous initial states in interacting models. This result, which we verify by numerical calculations, challenges the long-held paradigm that at the Euler scale, fluid cells may be considered uncorrelated. We also show that the Gallavotti-Cohen fluctuation theorem for non-equilibrium ballistic transport follows purely from time-reversal invariance of the Euler hydrodynamics. We check the validity of the BMFT by applying it to integrable systems, and in particular the hard-rod gas, with extensive simulations that confirm our analytical results.

# 1 Introduction

Determining the emergent, universal principles of non-equilibrium dynamics in many-body systems is an extremely active subject of current research. The emergence of hydrodynamics for the description of fluxes and large-scale motion is one of the most far-reaching ideas, and has been been extremely successful recently (see, e.g., [1–3]). However, a crucial question is to go beyond, and develop a statistical theory for fluctuations and correlations. An important result is the Gallavotti-Cohen fluctuation theorem [4, 5], which relates fluctuations of currents to equilibrium properties of reservoirs or driving forces, and which is valid no matter how far from equilibrium the system is. In this spirit, at large scales of space and time, $x, t \to \infty$, one would expect only few properties of the microscopic model to be required in order to understand the full spectrum of fluctuations. What are these properties and what is the statistical theory for large-scale dynamics?

In the last two decades, a universal approach to accessing the diffusive scale $x \sim \sqrt{t}$ has been proposed and developed: macroscopic fluctuation theory (MFT) [6, 7]. It is a large-deviation theory for many-body systems out of equilibrium whose hydrodynamics is purely diffusive, taking as only input the hydrodynamic equations of the system (thus, transport quantities such as the diffusion matrix). MFT has given an understanding of both current fluctuations and correlations, explaining long-range effects observed in driven-diffusive non-equilibrium steady states (NESS) [7]. It has been successfully applied to many classical many-body diffusive systems, such as the symmetric simple exclusion process (SSEP) [8–12]. It is expected to hold for stochastic or deterministic, classical or (at nonzero temperature) quantum systems, as on large scales one always finds hydrodynamics and classical fluctuations around it (although to our knowledge, no quantum application of the MFT has been reported so far). Quantum effects on fluctuations in diffusive systems have also been studied, with the construction of a quantum version of the MFT being considered [13].

A natural question is as to the nature of correlations and fluctuations in systems that admit *ballistic* transport. Ballistic transport means that persistent currents can exist even with vanishing gradients. It is realised in many important systems including the totally asymmetric exclusion process (TASEP) [14], anharmonic chains [15], and integrable many-body systems [16, 17]. In all cases, the leading hydrodynamic equation is the (general form of the) Euler hydrodynamic equation, which is solely based on the assumption of maximisation of entropy in local fluid cells and which arises at the ballistic (or hyperbolic) scaling $x \sim t$ [3, 18]. It takes the form

$$\partial_t \mathsf{q}_i(x,t) + \mathsf{A}_i{}^j(x,t)\partial_x \mathsf{q}_j(x,t) = 0, \tag{1}$$

where $\mathsf{q}_i(x,t)$ are averages of the local conserved densities admitted by the model, and $\mathsf{A}_i{}^j(x,t)$ is the "flux Jacobian", the variation of the fluxes with respect to the densities (see (8)),

$$\mathsf{A}_i{}^j = \frac{\partial \mathsf{j}_i}{\partial \mathsf{q}_j}, \tag{2}$$

in the entropy-maximised state at $x, t$. But relatively little is known about the structure of fluctuations at the ballistic scale as compared to what is available at the diffusive scale

with the MFT, the main works being [19–23]. One is thus looking for an adaptation of the framework of the MFT to the universal description of correlations and fluctuations of ballistic modes based solely on the Euler equation.

In Ref. [24], this theory, which we refer to as the ballistic MFT (BMFT), has been first introduced. The theory requires only the Euler hydrodynamics data of the system (the flux Jacobian $\mathsf{A}_i^{\ j}$ as a function of maximal entropy states), and accounts for the presence of any number of ballistic modes. The BMFT is a large-deviation theory, based on an action principle and saddle-point analysis much like the diffusive MFT, but it gives access to the ballistic scale instead of the diffusive scale. The BMFT provides the space-time probability distribution of fluctuating local observables in states that are subject to large-wavelength variations and motion and described by Euler hydrodynamics. In Ref. [24], the application of the BMFT to Euler scale correlations [19, 25] has been, in particular, considered. It has been therein fundamentally shown that the BMFT generically predicts that *long-range correlations develop over long times in non-stationary states*. For initial states with spatial variations on large wavelengths $\ell$, these are correlations which develop over time $t \sim \ell$, which extend on regions of size $x \sim \ell$, and with strength $\sim 1/\ell$. A physical interpretation is that correlated ballistic modes are emitted continuously at points where the state vary in time and scatter; alternatively, the one may see long-wavelength profiles in the fluid as being formed of a bath of ballistic modes that scatter and correlate continuously. See the pictorial representation in Fig. 1.

In this manuscript we systematically present and develop the results of the companion manuscript [24]. We discuss in details and in completely general terms the physical assumptions on which the BMFT relies and the associated implications. The BMFT states that Euler-scale fluctuations of time-evolved observables are obtained by deterministically transporting fluctuations of conserved quantities in the initial state via the Euler hydrodynamic equations of the model. This follows from a simple principle of "local relaxation of fluctuations" within fluid cells, similar in spirit to, but a refinement of, the principle of local relaxation that justifies the emergence of the Euler hydrodynamic equations themselves [18]. The principle stems from a separation of scales: non-conserved degrees of freedom vary quickly in time as they are affected by every interaction within the volume of a cell, and the local state rapidly covers the microcanonical shell; while conserved quantities vary more slowly as they are only affected by exchanges through the surface of fluid cells, and the local state slowly fluctuates amongst different microcanonical shells. We further develop the application of the BMFT to the ballistic large-deviation theory of current fluctuations. Using the BMFT, we show that the Gallavotti-Cohen fluctuation theorem for the large-deviation function of total currents holds independently from the details of the microscopic dynamics, purely as a consequence of a time-reversal symmetry of the Euler hydrodynamic equations. Considering the applications to correlations of the BMFT, we show that the latter naturally retrieves the whole structure of Euler-scale correlation functions, such as hydrodynamic projection formulas [18, 19, 26–30]. The emergence of long-range correlations in interacting many-body systems subjected to long-wavelength dynamics, even from states which have only short-range correlations, is further discussed. The differences between this phenomenon and correlations stemming from linear-response theory [19, 22, 23, 31] are presented (see Fig. 1). Both in the case of current fluctuations and of correlations, we give explicit results for integrable systems, based on generalised hydrodynamics (GHD) [16, 17]. In order to confirm these results, we will compare with simulations in the classical model of hard rods on the line.

We believe the BMFT is a framework with wide applicability, appropriate for the study

of a vast range of correlations and large deviations in many-body systems, integrable or not, quantum or classical, deterministic or stochastic. It is a universal tool capable of describing the rare fluctuations of any (mesoscopic) observable, including charge densities and currents, in a unified way. In principle the BMFT should work in arbitrary dimensionality; however, in order to lay out its foundation without any unnecessary complication, we shall focus on the one-dimensional case.

$$\ell \langle \overline{q_0}(\ell x, \ell t)\overline{q_0}(0,\ell t)\rangle^{\mathrm{c}}_{\ell}$$

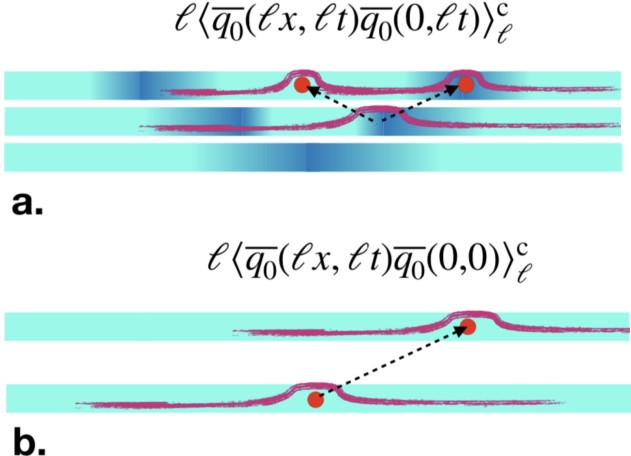

$$\ell \langle \overline{q_0}(\ell x, \ell t)\overline{q_0}(0,0)\rangle^{c}_{\ell}$$

Figure 1: **Correlations from the BMFT and linear response.** Pictorial representation of the Euler-scale evolution from a state with a density bump varying on a macroscopic length scale $\ell$ around the origin. Dark and light blue denote high and low particle density $\langle \overline{q_0}(\ell x, \ell t)\rangle_{\ell}$, respectively. Red lines are sketches of fluid waves while dotted arrows indicate their trajectories. Red points denote the space-time points of the observables involved in the correlations. (a) Long-range Euler-scaled correlations $\ell \langle \overline{q_0}(\ell x, \ell t)\overline{q_0}(0, \ell t)\rangle^{c}_{\ell}$ develop over time because of coherent wave emissions at positions where the state is inhomogeneous and non stationary, throughout the time evolution. This nonlinear effect is described by the BMFT and it necessarily requires an inhomogeneous and non stationary state, and two (as in the Figure), or more, interacting modes. Initially uncorrelated Euler-scale fluid cells therefore develop correlations over time. (b) Euler-scaled correlations at different times $\ell \langle \overline{q_0}(\ell x, \ell t)\overline{q_0}(0,0)\rangle^{c}_{\ell}$ also occur because of normal modes emitted by the perturbation of the state at the insertion of the earlier observable, and probed by the later observables. This mechanism is described by linear response and it can take place also in homogeneous and stationary states, such as the one sketched in the figure.

## 1.1   Previous works on the ballistic scale

There is an extensive literature on the Euler scale of various types of many-body systems, from stochastic particle systems to quantum spin chains. We provide here only a partial overview of some of the works relevant to this paper.

In stationary, homogeneous states, many results are available at the ballistic scale; this includes not only equilibrium states, but also generalised Gibbs ensembles in integrable models [32] and ballistic NESS emerging at long times from the partitioning protocol, where constant flows exist [16, 17, 33]. Here, the partitioning protocol refers to a particular initial condition that consists of two semi-infinite subsystems that are prepared at different (generalised) Gibbs

ensembles. Correlations have been studied by a variety of techniques, such as linear and non-linear response and hydrodynamic projection methods. Under ballistic scaling of space and time, they are purely controlled by the Euler hydrodynamics of the model. See, e.g., the recent reviews [25, 34].

The Boltzmann-Gibbs principle, that mesoscopic observables "project", in some way, onto mesoscopic conserved quantities, has a long history starting with Mori [35] and Zwanzig [36]. Until recently it had been studied mainly in stochastic particle systems assuming few conservation laws of the conventional form, see, e.g., the books [18, 26] and the paper [27] for a recent result. For correlation functions, the Boltzmann-Gibbs principle leads to hydrodynamic projection formulae. The first proposal for a general hydrodynamic projection formula, for two-point correlation functions in Hamiltonian systems with many-component Euler hydrodynamics, were written in [28] and [19] (in homogeneous, and long-wavelength inhomogeneous states, respectively). Hydrodynamic projections lead in a simple way to linearised Euler equations for two-point functions. In stationary, homogeneous states, the linearised Euler equation for two-point functions has been shown rigorously in the hard-rod gas [29]. Further, both the general hydrodynamic projection formula, and the linearised Euler equation, have been shown rigorously in every finite-range quantum spin chains [30]. There, the complete space of conserved quantities is defined rigorously, and so is the Euler scaling limit of two-point correlation functions.

For higher-point functions, nonlinear response, as first proposed in [19], can be used, as developed for integrable systems in [22], but the structure is less well understood.

In macroscopically inhomogeneous initial states of integrable systems, also much less is known. A theory has been developed for two-point correlation functions at the ballistic scale in [19], and numerical checks in the hard rod gas have confirmed the expression [31].

Fluctuations at the ballistic scale have also been studied in specific models by various hydrodynamic techniques, which differ from the BMFT developed here. In the TASEP, the earliest is that by Jensen [37] and Varadhan [38], culminating in an exact rate function (see, e.g., [39] for recent developments on the rate function of the TASEP from first principles). Another approach is via a ballistic extension of the diffusive MFT (similar to, but differing from, that proposed in section 3.3), which has been written [8] for the so-called weakly asymmetric simple exclusion process (WASEP), whose totally asymmetric limit gives the TASEP. By taking the totally asymmetric limit the Jensen-Varadhan formulation is re-obtained. In the TASEP, exact current large deviation functions (and SCGFs) are in fact known in various situations [40, 41].

The recently developed ballistic fluctuation theory (BFT) [20, 21], which provides the large-deviation theory for total current fluctuations in stationary states, is the first general theory for ballistic-scale fluctuations. The BFT is based on modifying the local states by accounting for the insertion of total currents, through a "flow equation" derived using hydrodynamic linear response. It shows that these fluctuations are also controlled by the Euler hydrodynamics of the model, and it accounts for an arbitrary number of ballistic modes. It implies, for instance, that "dynamical phase transitions" occur whenever the initial state admits a hydrodynamic mode with vanishing velocity. The first four cumulants of energy transport have been checked against numerical simulations in the hard rod gas [21], and the BFT reproduces the known exact SCGF in homogeneous states of the TASEP [42, Sec 4.2], obtained by first-principle calculations earlier [41]. It has been confirmed by exact calculations in the box-ball system [43]. The BFT was generalised to long-wavelength, non-stationary situations in integrable systems in [23], but again this is less understood.

The Gallavotti-Cohen fluctuation theorem (GCFT) has been studied widely, see the review [44] for basic results. In particular, in ballistic NESS, the GCFT was analytically proved for several models, see the review [33]. This includes (interacting) conformal field theories [45]. It has also been verified for integrable systems [21] by numerically evaluating the expression obtained from the BFT, although no proof yet has been provided. A general argument based on time-reversal symmetry of the microscopic model was proposed [46] for the full current fluctuations, including the early times of the partitioning protocol. For the latter state, the GCFT has been checked for various non-interacting models, see, e.g., [47,48].

## 1.2 Organisation of the paper

This paper is organised as follows. In Sec. 2, we set the stage by introducing the two main quantities that we will study: the scaled cumulant generating function (SCGF) for time-integrated currents and the Euler-scale dynamical correlation functions. In particular, we shall emphasise that the BMFT predicts a novel phenomenon that had been hitherto not known, which is the existence of long-range correlations in equal-time Euler-scale correlators in generic ballistic many-body systems. In Sec. 3, we then discuss the general idea of the BMFT, which rests upon the assumption of *local relaxation of fluctuations*, culminating in the expression of local equilibrium averages in terms of path-integrals. The relation to diffusive MFT is also explained. In Sec. 4, the actual implementations of the BMFT to compute the above two objects will be then carried out. The associated equations are hard to solve in general. In Sec. 5, we discuss integrable systems, where it turns out that all the difficulties presented in generic cases can be circumvented. The main predictions for this class of systems, as well as some technical details, are provided. We eventually conclude with future directions in Sec. 6. Technical aspects of the presentation and details about the calculations and the numerical simulations are consigned to the Appendix sections.

The notations used throughout the manuscript are as follows: microscopic system's observables (or operators in quantum systems) are denoted with a hat as $\hat{o}(x,t)$, and their fluid-cell mean, defined in (23), as $\overline{o}(x,t)$ (these are still operators in quantum systems). The integration of a microscopic observable over the full, infinite length of the system is denoted with a capital letter, $\hat{O} = \int_{\mathbb{R}} \mathrm{d}x\, \hat{o}(x)$. Note that the space and time used as arguments in $\hat{o}(x,t)$ and $\overline{o}(x,t)$ are *microscopic* ones. We will also define below *classical fluctuating variables* $o(x,t)$ in Eqs. (48)-(51) which are identified, via their Euler-scale correlation functions, with the mesoscopic means $\overline{o}(\ell x, \ell t)$. Recall that $\ell$ is the macroscopic scale, and thus in $o(x,t)$ space and time coordinates are *macroscopic* ones. Finally, we will also take ensemble averages of the microscopic observables $\hat{o}$ within homogeneous, stationary states (that is, within (generalised) Gibbs ensembles, see below), which we will denote in typewriter font, $\mathtt{o}$. With macroscopically varying states, such as in Eq. (12) below, the Lagrange multipliers (or generalised temperatures) $\beta^i(x)$, or $\beta^i(x,t)$, give rise to space or space-time dependent averages, $\mathtt{o}(x)$ or $\mathtt{o}(x,t)$, where space-time coordinates are again macroscopic.

# 2 Main physical predictions and numerical checks

We now overview the setup for the BMFT, the two main applications and physical predictions from the theory (the current large deviations and the CGFT in Subsec. 2.1, the Euler-scale correlation functions and long-range correlations in Subsec. 2.2), and the explicit results in

integrable systems and numerical checks we performed to verify these predictions (Subsec. 2.3).

The setup is an extensive model, quantum or classical, supported on the line $\mathbb{R}$, with a dynamics that admits a certain number of extensive conserved quantities,

$$\hat{Q}_i = \int_{\mathbb{R}} \mathrm{d}x \, \hat{q}_i(x). \tag{3}$$

Here and below, the set of values the index $i$ can take is kept arbitrary; it can be finite or infinite. Extensivity is intuitively understood as the fact that $q_i(x)$ is local – it probes the system at, or around, position $x$ only (this includes quasi-local densities as constructed in integrable models [49]). The set of charges $\hat{Q}_i$ is assumed to be complete[1].

One of the representative models that belongs to the former case is the classical anharmonic chain whose Hamiltonian reads [15]

$$\hat{H}_{\mathrm{AHC}} = \sum_{j=1}^{N} \left( \frac{1}{2} p_j^2 + V(r_j) \right), \quad r_j = x_{j+1} - x_j, \tag{4}$$

where $x_j$ and $p_j$ give the position and the momentum of $j$-th particle. Barring some exceptions, the model possesses the three conservation laws (3) for $i = 0, 1, 2$, corresponding to the number of particles $\hat{q}_0(x) = \sum_{j=1}^{N} \delta(x - x_j)$, the momentum $\hat{q}_1(x) = \sum_{j=1}^{N} p_j \delta(x - x_j)$, and the energy $\hat{q}_2(x) = \sum_{j=1}^{N} e_j \delta(x - x_j)$ with $e_j = \frac{1}{2} p_j^2 + V(r_j)$. Some exceptions occur when the potential is fine-tuned, giving a number of conservation laws that grows like $N$, e.g., the system of hard rods of lengths $a$, with $V_{\mathrm{HR}}(r) = \infty$ for $|r| \leq a$ and $V_{\mathrm{HR}}(r) = 0$ for $|r| > a$ [50], and the Toda chain $V_{\mathrm{Toda}}(r) = e^{-r}$ [51–53]. Quantum mechanically, one of the prototypical integrable models is the Lieb-Liniger model, which is given by [54]

$$\hat{H}_{\mathrm{LL}} = -\frac{1}{2} \sum_{j=1}^{N} \frac{\partial^2}{\partial x_j^2} + c \sum_{i<j} \delta(x_i - x_j), \tag{5}$$

where $c > 0$. Time evolution may, however, not necessarily be generated by a Hamiltonian; in stochastic models, a conserved quantity is to be understood as a martingale for the stochastic dynamics.

Under time evolution, continuity equations for local densities $\hat{q}_i(x, t)$ and currents $\hat{\jmath}_i(x, t)$ hold:

$$\partial_t \hat{q}_i(x, t) + \partial_x \hat{\jmath}_i(x, t) = 0. \tag{6}$$

Crucial to the structure of the BMFT and the expression of our main results are the set of stationary, homogeneous states where entropy is maximised with respect to the available conserved quantities. The states are characterised by a set of inverse "generalised temperatures" (temperature, chemical potential, Galilean or relativistic boosts, etc.), or Lagrange multipliers, $\beta^i$,

$$\langle \bullet \rangle_{\underline{\beta}} = \frac{1}{Z} \mu \left( \exp\left[ -\sum_i \beta^i \hat{Q}_i \right] \bullet \right), \quad \hat{Q}_i = \int_{\mathbb{R}} \mathrm{d}x \, \hat{q}_i(x). \tag{7}$$

---

[1]Completeness and extensivity are difficult to define rigorously in general. Although we are not looking for rigour in the present paper, we nevertheless mention that these concepts are given fully unambiguous definitions for the linearised Euler hydrodynamics of quantum spin chains in [30].

$\mu$ is any "flat *a priori* measure" that is homogeneous (invariant under space translation) and stationary (invariant time evolution). Physically, it is an infinite-temperature ensemble, such as the trace in quantum mechanics $\mu = \mathrm{Tr}$, or the flat phase-space integral $\mu = \int \prod_n \mathrm{d}p_n \mathrm{d}q_n$ in classical mechanics. Here and henceforth, $Z$ is the constant normalising the resulting state $\langle \bullet \rangle_{\underline{\beta}}$. Note that in integrable systems, where an infinity of conserved quantities may be involved in (7), these states are usually referred to as generalised Gibbs ensembles (GGE).

Also crucial are the flux Jacobian and Euler hydrodynamic equations. We recall that the flux Jacobian is the variation of the average currents $\mathsf{j}_i = \langle \hat{j}_i \rangle_{\underline{\beta}}$ with respect to the average densities $\mathsf{q}_i = \langle \hat{q}_i \rangle_{\underline{\beta}}$ within homogeneous stationary states,

$$\mathsf{A}_j{}^i = \mathsf{A}_j{}^i[\underline{\beta}] = \frac{\partial \mathsf{j}_j}{\partial \mathsf{q}_i} = -\sum_k \frac{\partial \mathsf{j}_j}{\partial \beta^k} \mathsf{C}^{ki}, \tag{8}$$

where $\mathsf{C}^{ki} = (\mathsf{C}^{-1})_{ki}$ is the inverse of the static covariance matrix, or susceptibility matrix, $\mathsf{C}_{ij}$,

$$\mathsf{C}_{ij} = -\frac{\partial \mathsf{q}_i}{\partial \beta^j} = \int_{\mathbb{R}} \mathrm{d}x \ \langle \hat{q}_i(x) \hat{q}_j(0) \rangle^{\mathrm{c}}_{\underline{\beta}}. \tag{9}$$

By positive-definiteness of the matrix $\mathsf{C}$ there is a bijection $\underline{\mathsf{q}} \leftrightarrow \underline{\beta}$ (in appropriate domains of values of these quantities). The Euler equation is obtained by assuming that the state is slowly varying in space and time so that in every fluid cell local entropy maximisation to the state (7) takes place. A fluid cell may be seen as a "mesoscopic" region: a region of extent $L$, large compared to microsocpic scales $\ell_{\mathrm{micro}}$, set by the mean inter-particle distance and interaction lengths, but small compared to macroscopic spatial variation scales $\ell$ (we take all the "scales" to have dimension of length),

$$\ell_{\mathrm{micro}} \ll L \ll \ell. \tag{10}$$

Thus all quantities associated to the state (7) now acquire $x, t$ dependence, $\beta^i(x,t)$, $\mathsf{q}_i(x,t)$ and $\mathsf{j}_i(x,t)$, and the resulting equation takes a number of equivalent forms: Eq. (1), and (using the property $\mathsf{AC} = \mathsf{CA}^T$, see, e.g., [28])

$$\begin{aligned} \partial_t \mathsf{q}_i(x,t) + \partial_x \mathsf{j}_i(x,t) &= 0, \\ \partial_t \beta^i(x,t) + \sum_j \mathsf{A}_j{}^i[\underline{\beta}(x,t)] \partial_x \beta^j(x,t) &= 0. \end{aligned} \tag{11}$$

Clearly, the Euler hydrodynamic equations are linear if the flux Jacobian is in fact independent of the state. We name henceforth, in agreement with Ref. [20], "interacting" those hydrodynamic systems where the flux Jacobian $\mathsf{A}_i{}^k(x,t) = \mathsf{A}_i{}^k[\underline{\beta}(x,t)]$ depends non-trivially on the state $\underline{\beta}(x,t)$.

The BMFT is concerned with out-of-equilibrium phenomena that happen at long wavelengths and long times. For definiteness, we will look at time evolution from initial states with long-wavelength $\ell$ variations

$$\langle \bullet \rangle_\ell = \frac{1}{Z} \mu \left( \exp \left[ -\sum_i \int_{-\infty}^{\infty} \mathrm{d}x \ \beta^i_{\mathrm{ini}}(x/\ell) \hat{q}_i(x) \right] \bullet \right). \tag{12}$$

For an extensive discussion of such long-wavelength states and the ensuing Euler hydrodynamics of many-body systems admitting an arbitrary number of conserved quantities, see, e.g., [3].

## 2.1 Current large deviations and fluctuation theorem

The BMFT gives access to the large-deviation theory for total observables integrated on macroscopic regions of space and time. Two natural examples are the scaled cumulant generating functions (SCGF) $\tilde{F}(\lambda, T)$ and $F(\lambda, T)$ of total charges

$$\left\langle e^{\lambda \hat{Q}(\ell X)} \right\rangle_\ell \asymp e^{\ell X \tilde{F}(\lambda, X)}, \quad \hat{Q}(\ell X) = \int_0^{\ell X} \mathrm{d}x\, \hat{q}_{i_*}(x, 0), \tag{13}$$

and, most interestingly, of total currents,

$$\left\langle e^{\lambda \hat{J}(\ell T)} \right\rangle_\ell \asymp e^{\ell T F(\lambda, T)}, \quad \hat{J}(\ell T) = \int_0^{\ell T} \mathrm{d}t\, \hat{j}_{i_*}(0, t), \tag{14}$$

as $\ell \to \infty$. The notation "$\asymp$" means, e.g., $F(\lambda, T) = \lim_{\ell \to \infty} \frac{1}{\ell T} \log \left\langle e^{\lambda \hat{J}(\ell T)} \right\rangle_\ell$, cf. Ref. [23]. Here $\hat{q}_{i_*}$ and $\hat{j}_{i_*}$ are one particular pair of charge density and current of the model; as the basis of conserved charges is kept arbitrary, this is without loss of generality. Note that Eq. (14) is equivalently the large-deviation theory for the total amount of the conserved quantity $i_*$ transported from left to right in time $\ell T$, that is $\hat{J}(\ell T) = \int_0^\infty \mathrm{d}x\, (q_{i_*}(x, \ell T) - q_{i_*}(x, 0))$. The SCGF's are generating functions for the scaled cumulants of $\hat{Q}(\ell X)$ and $\hat{J}(\ell T)$; for instance $F(\lambda, T) = \sum_{n \geq 0} \lambda^n c_n(T)/n!$ with

$$c_n(T) = \lim_{\ell \to \infty} (\ell T)^{-1} \int_0^{\ell T} \mathrm{d}t_1 \cdots \int_0^{\ell T} \mathrm{d}t_n \, \langle \hat{j}_{i_*}(0, t_1) \cdots \hat{j}_{i_*}(0, t_n) \rangle_\ell^{\mathrm{c}}. \tag{15}$$

In (13) and (14), the SCGF is evaluated in general in the macroscopically inhomogeneous state (12), where variations occur over the length scale $\ell$; thus the results depend on the scaled position $X$ and scaled time $T$. Many studies have concentrated on the special case of homogeneous initial states, where $F(\lambda, T) = F(\lambda, 1) =: F(\lambda)$, but our theory gives access to the general situation.

It is explained in [20] that $\tilde{F}(\lambda, X)$ is simply a difference of thermodynamic free energies (see also Appendix A),

$$\tilde{F}(\lambda, X) = \frac{1}{X} \int_0^X \mathrm{d}x\, (f[\underline{\beta_{\mathrm{ini}}}(x)] - f[\beta_{\mathrm{ini}}^\bullet(x) - \delta_{i_*}^\bullet \lambda]), \tag{16}$$

where $f[\underline{\beta}]$ is the specific free energy for the stationary state (7), defined as a generating function for averages of conserved densities, $\langle \hat{q}_i \rangle_{\underline{\beta}} = \partial f / \partial \beta^i$.

The quantity $F(\lambda, T)$ is more involved, as it requires understanding the dynamics of the model. In homogeneous states, where $\beta_{\mathrm{ini}}$ is indepedent of position, a general solution in terms of the Euler hydrodynamics is given by the BFT [20, 21]. However, the BMFT developed here is, to our understanding, the first general framework that applies to many-body systems both for homogeneous stationary states and long-wavelength inhomogeneous and non-stationary states.

Of particular interest is the partitioning protocol. In this setup, the initial state takes the Gibbs form with different, but otherwise constant, Lagrange multipliers on the right ($x > 0$)

and the left $(x < 0)$, for instance[2]:

$$\exp\left[-\sum_{i\in\mathcal{C}}\left(\beta_L^i\int_{-\infty}^0 \mathrm{d}x\,\hat{q}_i(x) + \beta_R^i\int_0^\infty \mathrm{d}x\,\hat{q}_i(x)\right) + \sum_{i\notin\mathcal{C}}\beta_0^i\hat{Q}_i\right], \tag{17}$$

where $\mathcal{C}$ specifies the set of the charges for which there is imbalance in the state. With this particular form of the initial state, choosing a single $i_*$ in the SCGF is no longer the most general case, and instead one may consider

$$F(\underline{\lambda}, T) = \lim_{\ell\to\infty}\frac{1}{\ell T}\log\left\langle e^{\sum_{i\in\mathcal{C}}\lambda^i\hat{J}_i(\ell T)}\right\rangle_\ell. \tag{18}$$

In this equation, $\hat{J}_i(\ell T)$ is defined analogously as $\hat{J}(T)$ after (14) with the replacement $i_* \to i$. The Gallavotti-Cohen fluctuation theorem (GCFT) gives a symmetry relation for this (generalised) current SCGF. As the initial state (17) is scale invariant, $F(\underline{\lambda}, T) = F(\underline{\lambda}, 1) =: F(\underline{\lambda})$, and the GCFT is

$$F(\underline{\beta}_L - \underline{\beta}_R - \underline{\lambda}) = F(\underline{\lambda}). \tag{19}$$

We will show, using the BMFT, that the GCFT holds solely as a consequence of a *time-reversal symmetry of the Euler hydrodynamics*. With $\mathsf{A}_i{}^j[\underline{\beta}]$ the flux Jacobian within the state (7), the symmetry is an appropriate time-reversibility of the Euler hydrodynamic solution, including the existence of a collection of signs $\mathcal{S}_i \in \{+1, -1\}$ such that

$$\mathsf{A}_i{}^j[\underline{\beta}] = -\mathcal{S}_i\mathcal{S}_j\mathsf{A}_i{}^j[\underline{\tilde{\beta}}], \tag{20}$$

where $\tilde{\beta}^i = \mathcal{S}_i\beta^i$ (no sum over repeated indices), along with the requirements $\mathcal{S}_i = 1$ and $\langle\hat{j}_i\rangle_{\underline{\beta}} = -\langle\hat{j}_i\rangle_{\underline{\tilde{\beta}}}$ for all $i \in \mathcal{C}$. This arises if there exists a time-reversal symmetry of the microscopic model, under which densities of conserved charges transform diagonally, with signs $\mathcal{S}_i$, see Eq. (84).

The result shows that, in ballistic many-body systems, the GCFT is a consequence of general principles of large-scale fluctuations, and does not depend on the microscopic details. In particular, by combining with generalised hydrodynamics, this provides, as far as we are aware, the first proof of the GCFT in all integrable many-body systems. The application to models of conventional hydrodynamic type, such as TASEP and the anharmonic chains, would require a more in-depth study of "weak solutions" to the BMFT equations (see below).

## 2.2 Long-range correlations

The BMFT also gives access to connected correlation functions of any number of mesoscopic observables at different, macroscopically separated points in space-time, again in general from large-wavelength states. Let us first recall the main elements of Euler-scale correlation functions.

The *Euler scaling limit* of correlation functions is the limit where space, time and wavelengths all tend to infinity simultaneously. The scaling limit in fact requires two additional

---

[2]This is not a large-wavelength initial state. However, it quickly settles to a large-wavelength state, and the effect of the initial transient on the cumulants vanish in the large-scale limit, as confirmed by our numerical results in Sec. 5.

operations: one must take *fluid-cell means*, and the correlation function must be rescaled appropriately. See [19]. For the $n$-point function, the Euler scaling limit is

$$S_{\hat{o}_1,\dots,\hat{o}_n}(x_1,t_1;\cdots;x_n,t_n) := \lim_{\ell\to\infty} \ell^{n-1} \langle \overline{o_1}(\ell x_1,\ell t_1)\cdots\overline{o_n}(\ell x_n,\ell t_n)\rangle_\ell^{\rm c}, \tag{21}$$

where $\langle\cdots\rangle_\ell^{\rm c}$ is the connected correlation function in the initial state (12), again with its dependence on the scale of spatial variations $\ell$ explicitly written. For the two point function, explicitly, $\langle \overline{o_1}(x_1,t_1)\overline{o_2}(x_2,t_2)\rangle_\ell^{\rm c} = \langle \overline{o_1}(x_1,t_1)\overline{o_2}(x_2,t_2)\rangle_\ell - \langle \overline{o_1}(x_1,t_1)\rangle_\ell \langle \overline{o_2}(x_2,t_2)\rangle_\ell$, and

$$S_{\hat{o}_1,\hat{o}_2}(x_1,t_1;x_2,t_2) := \lim_{\ell\to\infty} \ell \langle \overline{o_1}(\ell x_1,\ell t_1)\overline{o_2}(\ell x_2,\ell t_2)\rangle_\ell^{\rm c}. \tag{22}$$

In the Euler-scaling limit (21), the quantity $\overline{o_i}(x,t)$ is the fluid-cell means of $\hat{o}_i$ around the space-time point $x,t$; the fluid-cell mean can be taken as a mesoscopic space-time average

$$\overline{o}(x,t) = \frac{1}{vL^2} \int_{-L/2}^{L/2} {\rm d}y \int_{-vL/2}^{vL/2} {\rm d}s\, \hat{o}(x+y,t+s), \tag{23}$$

for some $v > 0$. The fluid-cell mean (23) is generically required in order for the limit expressed in (22) to be described by Euler hydrodynamics. Time-averaging is, in general, necessary in order to wash out oscillations; see for instance the discussion in Ref. [55] for the XX spin chain and in Ref. [56] for the classical sinh-Gordon field theory. For the hard-rod model, however, we find that averaging in space (as in Eq. (35)) works well (see also Ref. [31]). See [19,25] for discussions of fluid-cell means.

In general, $S_{\hat{o}_{i_1},\dots,\hat{o}_{i_n}}(x_{i_1},t_{i_1};\cdots;x_{i_n},t_{i_n})$ is expected to be *symmetric under permutations of indices* $i_k \mapsto i_{\sigma(k)}$. This is clear in classical systems, but should also hold in quantum systems thanks to the taking of the Euler scaling limit, as shown by the rigorous results of [30] for two-point functions in stationary homogeneous states of quantum spin chains.

As discussed in [23], the SCGF $F(\lambda,T)$ is related to Euler-scale correlation functions. Indeed, from Eqs. (14) and (15), and assuming that we can neglect the spatial part of the fluid-cell mean,

$$c_n(T) = T^{-1} \int_0^T {\rm d}t_1 \cdots \int_0^T {\rm d}t_n\, S_{\hat{j}_{i_*},\dots,\hat{j}_{i_*}}(0,t_1;\cdots;0,t_n). \tag{24}$$

Therefore, conceptually, Euler-scale correlation functions (21) encode all Euler-scale information.

We note that the existence (in some sense) of the limit (21) does not mean that the $n$-point correlation function of microscopic observables decays as $\ell^{1-n}$. In non-integrable systems the result of the limit is expected to be a distribution, where delta-functions are found along the characteristics of the fluid, representing a decay of microscopic correlations that is slower than $\ell^{1-n}$. In integrable systems, because of the continuum of fluid modes, $n$-point correlation functions generically indeed decay as $\ell^{1-n}$. See the review [25].

Before presenting our results, we recall that using hydrodynamic linear response arguments and other related principles, one arrives, from the Euler hydrodynamic equations (1), (11), at predictions for Euler-scale correlations. This is based on the fact that the hydrodynamic equations, with initial condition $\beta^i(x,0) = \beta^i_{\rm ini}(x)$ or equivalently $\mathsf{q}_i(x,0) = \langle \hat{q}_i\rangle_{\underline{\beta}_{\rm ini}(x)}$, predict the averages of all mesoscopic observables [3,18]:

$$\lim_{\ell\to\infty} \langle \overline{o}(\ell x,\ell t)\rangle_\ell = \langle \hat{o}\rangle_{\underline{\beta}(x,t)} =: \mathsf{o}(x,t). \tag{25}$$

The main lines of linear and nonlinear response arguments in multi-component Euler hydrodynamics (see Appendix C for a short review) are expressed in [19, 22], where they are worked out for integrable models.

One prediction is the hydrodynamic projection formula for two-point functions [19]:

$$S_{\hat{o}_1,\hat{o}_2}(x_1,t_1;x_2,t_2) = \frac{\partial \mathsf{o}_1}{\partial \mathsf{q}_i}\Big|_{\underline{\beta}(x_1,t_1)} \frac{\partial \mathsf{o}_1}{\partial \mathsf{q}_j}\Big|_{\underline{\beta}(x_2,t_2)} S_{\hat{q}_i,\hat{q}_j}(x_1,t_1;x_2,t_2), \tag{26}$$

where for lightness of notation, here and below, the Einstein convention of summation over repeated indices is used. That is, it is sufficient to know the Euler-scale correlation functions of conserved densities, in order to access other Euler-scale correlation functions. In stationary homogeneous states (7) of short-range quantum spin chains, this has been shown rigorously in Ref. [30]. In particular,

$$S_{\hat{j}_k,\hat{q}_j}(x_1,t_1;x_2,t_2) = \mathsf{A}_k{}^i(x_1,t_1)S_{\hat{q}_i,\hat{q}_j}(x_1,t_1;x_2,t_2). \tag{27}$$

Combining (27) with the conservation laws (6), one obtains a linear equation for $S_{\hat{q}_i,\hat{q}_j}(x,t;x',t')$:

$$\partial_t S_{\hat{q}_i,\hat{q}_j}(x,t;x',t') + \partial_x \Big( \mathsf{A}_i{}^k(x,t)S_{\hat{q}_k,\hat{q}_j}(x,t;x',t') \Big) = 0. \tag{28}$$

This is a linearised version of the Euler equations, which physically represents the propagation of a small disturbance on top of a (generically) inhomogeneous, non-stationary background. Eq. (28) says that the hydrodynamic modes are transported via the local flux Jacobian $\mathsf{A}_i{}^k(x,t) = \mathsf{A}_i{}^k[\beta(x,t)]$ viewed as a "propagator", being emitted by the observable at $x',t'$ and probed by the observable at $x,t$. Of course, in homogeneous and stationary states, Eq. (28) can be solved explicitly by a simple Fourier transform as the flux Jacobian can be taken out of the derivative.

We will show that the BMFT gives the full structure of Euler-scale correlation functions. First, we show that (26) and (28) indeed hold. We also obtain a generalisation of hydrodynamic projections to higher-point functions, and related linearised Euler hydrodynamic equations. For three-point functions, omitting the explicit space-time arguments for lightness of notation, hydrodynamic projection is

$$S_{\hat{o}_1,\hat{o}_2,\hat{o}_3} = \frac{\partial \mathsf{o}_1}{\partial \mathsf{q}_i}\frac{\partial \mathsf{o}_2}{\partial \mathsf{q}_j}\frac{\partial \mathsf{o}_3}{\partial \mathsf{q}_k}S_{\hat{q}_i,\hat{q}_j,\hat{q}_k} + \frac{\partial^2 \mathsf{o}_1}{\partial \mathsf{q}_i \partial \mathsf{q}_j}\frac{\partial \mathsf{o}_2}{\partial \mathsf{q}_k}\frac{\partial \mathsf{o}_3}{\partial \mathsf{q}_l}S_{\hat{q}_i,\hat{q}_k}S_{\hat{q}_j,\hat{q}_l} + \text{cyclic perm. of } 1,2,3, \tag{29}$$

and the evolution equation is

$$\partial_{t_1}S_{\hat{q}_i,\hat{q}_j,\hat{q}_k} + \partial_{x_1}\Big( \mathsf{A}_i{}^l S_{\hat{q}_l,\hat{q}_j,\hat{q}_k} + \mathsf{H}_i{}^{lr} S_{\hat{q}_l,\hat{q}_j}S_{\hat{q}_r,\hat{q}_k} \Big) = 0, \tag{30}$$

where $\mathsf{H}_i{}^{lr} = \partial^2 \mathsf{j}_i/\partial \mathsf{q}_l \partial \mathsf{q}_r$. The evolution equation is in agreement with nonlinear response results of [22], and a similar structure has been found in the hydrodynamic limit of the quantum single exclusion process [57, 58]. These, as we will show, in fact follow quite directly from the scaling (21) and the BMFT principle of relaxation of fluctuations.

We further show one of the most striking physical predictions of the BMFT: *the lack of correlations between separate fluid cells that is often assumed in hydrodynamic response theory, is in fact not generically preserved under macroscopic time evolution.*

By standard arguments, an initial state of the form (12) has quickly decaying correlations. With finitely-many local densities $\hat{q}_i(x)$, exponential decay is typically found on microscopic length scales $\ell_{\text{micro}} \ll \ell$, much like in equilibrium states. In integrable systems, as infinitely-many charges may be involved, $1/x^2$ correlations may appear, as is found in NESS [59,60]. But in all cases, correlations are expected to decay faster than $1/|x|$. This means that equal-time Euler-scale correlations in this state vanish at different points,

$$S_{\hat{o}_1,\hat{o}_2}(x_1, 0; x_2, 0) = 0 \quad \text{if } x_1 \neq x_2. \tag{31}$$

In this states, fluid cells are not correlated at macroscopic scales. In particular, if $t' = 0$, the initial condition for (28), as taken in [19], is

$$S_{\hat{q}_i,\hat{q}_j}(x, 0; x', 0) = \mathsf{C}_{ij}(x, 0)\,\delta(x - x'). \tag{32}$$

In the conventional view of Euler hydrodynamics, under time evolution, local entropy maximisation occurs with respect to the mesoscopic conserved quantities in fluid cells, and the form (12), with time-dependent $\beta^i(x,t)$, describes the state later in time. This indeed correctly describes averages of local observables. But the BMFT generically *invalidates the time-evolved form of (12) at nonzero macroscopic times for the description of Euler-scale correlations.* That is, the set of states of the form (12) is not preserved under time evolution. Long-range correlations appear,

$$S_{\hat{o}_1,\hat{o}_2}(x_1, t; x_2, t) \neq 0 \quad \text{for } |x_1 - x_2| > 0, \ t > 0, \tag{33}$$

and the BMFT gives a quantitative prediction.

In fact, it is possible to argue for long-range correlations directly from hydrodynamic projection. Indeed, if the flux Jacobian depends on the state and has a nontrivial matrix structure, and the state is space-time dependent, the evolution equation

$$\partial_t S_{\hat{q}_i,\hat{q}_j}(x, t; x', t) + \partial_x \Big( \mathsf{A}_i{}^k(x, t) S_{\hat{q}_k,\hat{q}_j}(x, t; x', t) \Big) + \partial_{x'} \Big( \mathsf{A}_j{}^k(x', t) S_{\hat{q}_i,\hat{q}_k}(x, t; x', t) \Big) = 0, \tag{34}$$

which follows from (26) for $t_1 = t_2 = t$ and the conservation laws, does not preserve the initial delta-function structure. Although hydrodynamic projection for two-point functions was already known, this observation, it appears, had not been made before.

Long-range correlations occur whenever the system is interacting (that is, the flux Jacobian depends on the state), spatio-temporal variations are present, and there are more than one fluid velocity. They are built by macroscopically separated fluid modes going at different velocities, that have been emitted coherently in small regions with spatio-temporal variations, and scatter with each other. More generally, correlations at the Euler scale in space-time, such as (22), are not only due to fluid modes propagating between observables as obtained by linear response theory, but also receive contributions from fluid modes coherently emitted in the past. As a consequence, the initial condition (32) does not hold at times $t' > 0$, in contrast to what is normally assumed in linear- and nonlinear-response studies.

The above mechanism for long-range correlations is of hydrodynamic origin. In certain situations there may be other mechanisms at play, giving rise to long-range correlations of similar strength, but not described by the BMFT. For instance, if the initial condition is not of large-scale variations, say it admits a step density profile, then it can be expected that during the initial stage of the dynamics some nontrivial correlations build up from the microscopic

physics. Thus dynamical correlations of observables at different spatial positions, from such initial conditions, will be given by the combination of the early time microscopic ones and hydrodynamics ones as above. We shall indeed numerically observe for the hard-rod model evolving from a step initial condition, that correlations that stem from microscopic physics give additional contributions to those from coherent production of normal modes described by the BMFT.

Lastly, we remark that the observed long-range correlations are *not* in contradiction with the Lieb-Robinson bound [61], which constrains the spatial extent of time-evolved operators in lattice systems. By the Lieb-Robinson bound, nontrivial long-range correlations (33) can exist only if the two Lieb-Robinson light cones that fan out from $x = x_1$ and $x = x_2$ overlap; beyond this, correlations must decay exponentially (or as fast as in the initial state). Our explicit formulae for long-range correlations in integrable systems (132) show that correlations indeed decay quickly when the light cones determined by the maximum fluid velocity do not overlap; the maximum fluid velocity is bounded by the Lieb-Robinson velocity.

## 2.3   Numerical checks: integrable systems

As mentioned, the BMFT is based on an action principle. The BMFT action turns out to be rather simple. It yields a set of hydrodynamic-type equations of motion that describe the representative, "typical" dynamics which encodes the rare but large fluctuations at the root of Euler-scale correlations. Handling the resulting BMFT equations is however tricky. First, obtaining exact, analytic solutions is usually difficult, much like in conventional MFT (see however the recent exact result [12, 62]). Second, generically the profile of the solution would display points of non-continuity where entropy is not conserved, as in Euler hydrodynamics. Due care is therefore needed in determining which weak solution to choose. This is a problem that has been addressed in simple models by first formulating the MFT in the presence of viscosity terms and taking the vanishing viscosity limit [8]. In the present paper, we do not address this problem more generally, although we provide some pointers.

In order to avoid these difficulties and assess our new theory in models which are simple enough yet which present all the structures of interacting Euler hydrodynamics, we concentrate on one-dimensional integrable systems. Their hydrodynamics, which has been coined generalised hydrodynamics (GHD), has come under intensive scrutiny over the last five years [16, 17, 63, 64] (see also the review [3] and the special issue [65]). GHD is based on the idea that fluid cells are described by generalised Gibbs ensemble (GGE). These are, nominally, Eq. (7) with, in general, infinitely many charges involved. These are the ensembles emerging in integrable systems under relaxation [32], instead of the usual Gibbs ensembles. In GHD, one in fact forgoes the set of explicit, extensive conserved charges, and describes the relaxed states more universally in terms of distributions of asymptotic states, see [3]. GHD has been observed in recent experiments on cold atomic gases [66–68], and it is also the framework at the heart of the theory of soliton gases, structures observed in water-wave and light-guide experiments [69–71].

In integrable systems, an intricate analysis of entropy-non-conserving discontinuous points is not necessary, as their hydrodynamic equations are known to have no shock solutions [72], something that is attributed to *complete linear degeneracy* of these equations [3, 73, 74]. Linear degeneracy means that the hydrodynamic velocity of a given normal mode does not depend on the value of this normal mode. This allows for contact discontinuities, instead of shocks, to develop, where entropy is preserved, as seen for instance in non-equilibrium steady states

[16]. Further, one of the salient features of GHD is that it admits exact solutions of its hydrodynamic equations by means of the method of characteristics [75]. This fact greatly facilitates the application of the BMFT to integrable systems, allowing us to understand the structure of the BMFT equations to a much larger extent than in usual interacting many-body systems.

We concentrate on the two types of physical quantities discussed above: the scaled cumulants $c_n(T)$ for total time-integrated currents, Eqs. (14), (15), and the Euler-scale, equal-time two-point functions $S_{\hat{q}_i,\hat{q}_j}(x_1, t_1; x_2, t_2)$, Eq. (22). The latter is numerically evaluated as:

$$S_{\hat{q}_0,\hat{q}_0}(x_1, t; x_2, t) = \lim_{\ell \to \infty} \frac{\ell}{L^2} \int_{-L/2}^{L/2} \mathrm{d}y \int_{-L/2}^{L/2} \mathrm{d}y' \ \langle \hat{q}_0(\ell x_1 + y, \ell t)\hat{q}_0(\ell x_2 + y', \ell t)\rangle_\ell^{\mathrm{c}}. \tag{35}$$

The integrals over $y$ and $y'$ give the fluid-cell means $\overline{q}_0(\ell x_1, \ell t)$ and $\overline{q}_0(\ell x_2, \ell t)$ over the mesoscopic scale $L$ in Eq. (10)[3]. The numerical evaluation of $c_n(T)$ is detailed in Appendix I. The calculations are performed in the generality of the universal GHD formalism, and thus the results apply to all integrable many-body systems, quantum or classical. The numerical comparisons are done against simulations of the hard rod model. This model is simple enough to simulate with good statistics, yet non-trivial enough to present all the properties of generic interacting intergable systems. The hydrodynamic theory of the hard rod model [50] is known to be a special case of GHD [63,76], and thus our GHD result can immediately be specialised to it (see Appendix B).

First, the scaled cumulants are evaluated both in homogeneous, stationary states (7), and in the non-stationary configurations emanating from an initial partitioning of the system into two homogeneous halves as in Eq. (17). As in both cases there is scale invariance, the scaled cumulants are not time dependent, $c_n(T) = c_n$. For homogeneous states, we show that the BMFT results for the first nontrivial scaled cumulants $c_2$ and $c_3$ agree with the known expressions from the BFT [21]. For the partitioning protocol, we show that our BMFT results also agree with the recent inhomogeneous BFT proposal [23]. We further compare the $c_2$ result against molecular-dynamics simulations of the hard rod model, with excellent agreement.

As an illustration, we find that $c_2^{\mathrm{part}}$, the second cumulant associated to particle transport ($i_* = 0$ in our convention) in the partitioning protocol of the hard-rod model, is given by

$$c_2^{\mathrm{part}} = (1 - a\rho)^3 \int_{\mathbb{R}} \mathrm{d}\theta \, n^\theta |v_\theta^{\mathrm{eff}}|. \tag{36}$$

Here $a$ is the rod length, $\rho$ is the density of the rods in the NESS, and $n^\theta$ is the "normal mode" density in the NESS, which is simply the velocity distribution function, labeled by the velocity $\theta$, normalised in such a way that $\rho(1 - a\rho)^{-1} = \int_{\mathbb{R}} \mathrm{d}\theta \, n^\theta$. The effective velocity of the rods $v_\theta^{\mathrm{eff}}$ is given by

$$v_\theta^{\mathrm{eff}} = \frac{1}{1 - a\rho} \left( \theta - (1 - a\rho) \int_{\mathbb{R}} \mathrm{d}\phi \, \phi n^\phi \right). \tag{37}$$

The NESS is the fluid state on the ray $x/t = 0$ of the partitioning protocol; see the solution to the hydrodynamic Riemann problem of the hard rods in [76]. Interestingly, expression (36) is precisely the same as its homogeneous version except that each thermodynamic quantity is

---

[3]One may take $L = \epsilon\ell$ for some $\epsilon > 0$, and take $\epsilon \to 0$ after the limit on $\ell$. Numerical simulations were done with $\epsilon = 0.05$. In the hard rod gas, we expect $L = 0$, no fluid-cell averaging, would also work, but this is hard to verify numerically.

now evaluated with respect to the NESS. The reason for this resemblance will be explained in later sections. We find perfect agreement with numerics, thereby confirming the validity, at least for this cumulant, of both the inhomogeneous version of BFT, and of our new BMFT.

Second, we provide, using the GHD normal modes, explicit sets of integral equations for two-point correlation functions of conserved densities in space-time, from arbitrary large-wavelength initial configurations, in the Euler scaling limit, $S_{\hat{q}_i,\hat{q}_j}(x_1, t_1; x_2, t_2)$.

In particular, we focus on the equal-time case $t_1 = t_2 = t$. The assumption that the state is of locally entropy-maximised form at that time, Eq. (12), would only give a delta function; as discussed already, the BMFT generically invalidates this assumption.

For the density-density dynamical correlation function $S_{\hat{q}_0,\hat{q}_0}(x, t; 0, t)$ in the hard-rod gas, the BMFT yield the following exact expression:

$$S_{\hat{q}_0,\hat{q}_0}(x, t; 0, t) = \rho(x,t)(1 - a\rho(x,t))^2 \delta(x) - (1 - a\rho(0,t))^2 \int_{\mathbb{R}} \mathrm{d}\theta \, [n\mathcal{E}]^\theta(0,t), \qquad (38)$$

where $\mathcal{E}^\theta(x,t)$ satisfies the integral equation

$$\mathcal{E}^\theta(x,t) = \mathcal{E}_0^\theta(x,t) + w^\theta(x,t) \int_{-\infty}^x \mathrm{d}y \, (1 - a\rho(y,t))[n\mathcal{E}]^{\mathrm{dr};\theta}(y,t). \qquad (39)$$

In the previous equation, $\rho(x,t)$ and $n^\theta(x,t)$ are defined as before except that this time they are functions of space and time, and evaluated in the fluid cell at $x, t$. The objects $\mathcal{E}_0^\theta(x,t)$ and $w^\theta(x,t)$ are some (cumbersome) functionals of $n^\theta(x,t)$, which can be obtained from Eqs. (132)-(138) by specialising to the hard-rod gas (see Appendix B). Finally, $\mathsf{a}^{\mathrm{dr};\theta}(x,t)$ for any function $\mathsf{a}^\theta(x,t)$ is defined by

$$\mathsf{a}^{\mathrm{dr};\theta}(x,t) := \mathsf{a}^\theta(x,t) - a(1 - a\rho(x,t)) \int_{\mathbb{R}} \mathrm{d}\phi \, n_\phi \mathsf{a}^\phi(x,t). \qquad (40)$$

We verify numerically with the hard rod simulations that long-range correlations indeed develop, that is, that $S_{\hat{q}_0,\hat{q}_0}(x_1, t; x_2, t)$ gives, for all $t > 0$, a function of $x$ that is nonzero on an extended region. Furthermore, we also confirm that this function agrees with the above BMFT prediction. We have done this from Eqs. (38) and (39) (and Eqs. (132)-(138)) for $S_{\hat{q}_0,\hat{q}_0}(x, t; 0, t)$ in Fig. (1) of the companion manuscript [24]. In the case of the correlator $S_{\hat{q}_0,\hat{q}_0}(x, t; -x, t)$, we present our analysis in Sec. 5 (see Fig. 4).

## 3  MFT for ballistic transport

In this Section, we introduce the BMFT formalism, which will be used in the calculations presented in the next sections of the manuscript. In particular, in Subsec. 3.1, we give the main statements of the BMFT in Eqs. (46)-(49), along with the predictions (50), (51). In Subsec. 3.2, we provide a justification for these statements. In Subsec. 3.3, a connection between BMFT and the conventional diffusive MFT is presented. The ideas developed in Subsec. 3.3 are not used in later sections, hence for a basic understanding of the BMFT, this subection can be skipped.

### 3.1  Formulation of the BMFT

The starting point for our implementation of the BMFT is the set of initial states (12). Below we sometimes write $\mathsf{q}_i = \mathsf{q}_i[\underline{\beta}]$ to emphasise its functional dependence on the $\beta^i$'s; as

mentioned just before Eq. (11), there is a bijection $\mathsf{q} \leftrightarrow \underline{\beta}$. Likewise, we write $\mathsf{A} = \mathsf{A}[\underline{\beta}]$. For local observables $\hat{o}$, including the currents $\hat{o} = \hat{j}_i$, we write

$$\mathsf{o}[\underline{q}] := \langle \hat{o} \rangle_{\underline{\beta}} \quad (\text{under } q_i = \mathsf{q}_i[\underline{\beta}]). \tag{41}$$

As is clear from (25) at $t = 0$ (and the initial condition $\beta^i(x, 0) = \beta^i_{\text{ini}}(x)$), the marginal distributions of fluid cells in the initial state, or the fluid-cell reduced density matrices in the quantum language, give, at least for mesoscopic means, GGEs with the local values of $\beta^i$. By the fast correlation decay discussed above, the full measure induced on mesoscopic means is a product measure over these local fluid-cell marginals.

It is a simple statistical mechanics exercise [77] to express the product measure representing (12) on the mesoscopic conserved quantities. In order to do so, one sees the coarse-grained, mesoscopic means $\overline{q_i}(\ell x, 0)$, whose correlations are obtained from (12), as classical fluctuating variables, $q_i(x, 0)$, whose correlations are obtained from an appropriate measure $d\mathbb{P}_{\text{ini}}[\underline{q}(\cdot, 0)]$; that is, one sets the equivalence

$$\overline{q_i}(\ell x, 0) \equiv q_i(x, 0) \quad (\text{at Euler scale}). \tag{42}$$

The equivalence hold in the Euler scaling limit in Eq. (10) with $L, \ell \to \infty$. In macroscopic coordinates, the fluid cell scale $L$ shrinks to a point, thus the measure $d\mathbb{P}_{\text{ini}}[\underline{q}(\cdot, 0)]$ is pointwise factorised. One finds that the following measure represents well the initial state (12): $d\mathbb{P}_{\text{ini}}[\underline{q}(\cdot, 0)] = d\mu[\underline{q}(\cdot, 0)] \, e^{-\ell \mathcal{F}[\underline{q}(\cdot, 0)]}$ where $\mathcal{F}[\underline{q}(\cdot, 0)]$ can be expressed as follows[4]:

$$\mathcal{F}[\underline{q}(\cdot, 0)] = \int_{\mathbb{R}} dx \left( \beta^i_{\text{ini}}(x) q_i(x, 0) - f[\underline{\beta}_{\text{ini}}(x)] - s[\underline{q}(x, 0)] \right). \tag{43}$$

Here $d\mu[\underline{q}(\cdot, 0)] = \prod_{x \in \mathbb{R}} \prod_i dq_i(x)$ is the flat measure, and $s[\underline{q}]$ is the entropy density, which is defined (up to an unimportant constant) by the equations $\partial s[\underline{q}] / \partial q_i |_{q_i = \mathsf{q}_i[\underline{\beta}]} = \beta^i$. The function $\mathcal{F}[\underline{q}(\cdot, 0)]$ can be interpreted as a relative entropy[5], and it is clearly pointwise factorised.

The measure $d\mathbb{P}_{\text{ini}}[\underline{q}(\cdot, 0)]$ represents the initial state (12) at the Euler scale. Indeed, the generating function of equal-time Euler-scale correlations takes the form of a difference of integrated free energies, generalising (16) (see Appendix A):

$$\left\langle \exp \int_{\mathbb{R}} dx \, \lambda^i(x/\ell) \hat{q}_i(x) \right\rangle_\ell \asymp \exp \ell \left( \int_{\mathbb{R}} dx \left( f[\underline{\beta}_{\text{ini}}(x)] - f[\underline{\beta}_{\text{ini}}(x) - \underline{\lambda}(x)] \right) \right), \tag{44}$$

and the Legendre transform, of a difference of free energies is a relative entropy, giving the large-deviation functional $e^{-\ell \mathcal{F}[\underline{q}(\cdot, 0)]}$ in Eq. (43) (see e.g. Ref. [78]). In particular, the saddle-point of this measure indeed gives the correct averages of conserved densities for (12); defining $\beta^i(x, 0)$ via the relation $q_i(x, 0) = \mathsf{q}_i[\underline{\beta}(x, 0)]$, the saddle-point equation is

$$\beta^i(x, 0) = \beta^i_{\text{ini}}(x, 0) \quad (\text{saddle-point of } d\mathbb{P}_{\text{ini}}[\underline{q}(\cdot, 0)]). \tag{45}$$

The main purpose of the BMFT is to give an action principle that describes how the probability distribution $d\mathbb{P}_{\text{ini}}[\underline{q}(\cdot, 0)]$ *extends to a probability distribution for the full space-time mesoscopic conserved densities*, $d\mathbb{P}[\underline{q}(\cdot, \cdot)]$ for $q_i(\cdot, \cdot)$'s on $\mathbb{S} := \mathbb{R} \times [0, T]$, for any time

---

[4]One can also write $\mathcal{F}[\underline{q}(\cdot, 0)] = \int_{\mathbb{R}} dx \int_{\mathsf{q}_{\text{ini}}(x)}^{q(x, 0)} dr_i \, \mathsf{C}^{ij}[\underline{r}](q_j(x, 0) - r_j)$ where $\mathsf{C}[\underline{r}]$ is the static covariance matrix as a function of average conserved densities $r_i$'s.

[5]Indeed, $\mathcal{F}[\underline{q}(\cdot, 0)] = \int_{\mathbb{R}} dx \, \mu \left[ \hat{\varrho}[\underline{\beta}(x, 0)] \log \left( \hat{\varrho}[\underline{\beta}(x, 0)] / \hat{\varrho}[\underline{\beta}_{\text{ini}}(x)] \right) \right]$, where $\hat{\varrho}[\underline{\beta}] = \exp \left[ - \sum_i \beta^i \hat{Q}_i \right] / Z(\underline{\beta})$.

$T$. This action principle is derived from the main assertion of the BMFT that the initial distribution $\mathrm{d}\mathbb{P}_{\mathrm{ini}}[\underline{q}(\cdot,0)]$ propagates in time according to the Euler equations (11). We will justify this assertion below in Subsec. 3.2.

Specifically, the resulting measure on space-time configurations is

$$\mathrm{d}\mathbb{P}[\underline{q}(\cdot,\cdot)] = \mathrm{d}\mu[\underline{q}(\cdot,\cdot)]\, e^{-\ell\mathcal{F}[\underline{q}(\cdot,0)]}\delta[\partial_t\underline{q} + \partial_x\underline{j}[\underline{q}]]. \tag{46}$$

$\mathrm{d}\mu[\underline{q}(\cdot,\cdot)]$ is the flat measure for functions on $\mathbb{S}$. The delta functional, which we represent as an integral over an auxiliary fields (as is done in the diffusive MFT [10])

$$\delta[\partial_t\underline{q} + \partial_x\underline{j}[\underline{q}]] = \int_{(\mathbb{S})} \mathrm{d}\mu[\underline{H}(\cdot,\cdot)] \exp\left[-\ell\int_{\mathbb{R}}\mathrm{d}x\int_0^T\mathrm{d}t\, H^i(\partial_t q_i + \partial_x j_i[\underline{q}])\right], \tag{47}$$

enforces the Euler equations. Here on the integral symbol we indicate the range of the functions in the function space over which we integrate.

Euler-scale correlation functions involving mesoscopic fluid-cell means of any observables $\overline{o}(\ell x, \ell t)$ are obtained with the above measure by identifying them with appropriate classical random variables. They are identified with the functions of $q_i(x,t)$'s representing their GGE averages in the local states characterised by $q_i(x,t)$'s:

$$\overline{o}(\ell x, \ell t) \equiv \mathsf{o}[\underline{q}(x,t)] \quad \text{(at Euler scale)}. \tag{48}$$

Explicitly, the BMFT average is given by

$$\langle\!\langle\bullet\rangle\!\rangle_\ell = \frac{1}{Z}\int_{(\mathbb{S})}\mathrm{d}\mu[\underline{q}(\cdot,\cdot)]\, e^{-\ell\mathcal{F}[\underline{q}(\cdot,0)]}\delta(\partial_t\underline{q} + \partial_x\underline{j}[\underline{q}])\,\bullet\,. \tag{49}$$

The main BMFT predictions are expressions for Euler-scale correlations (21)

$$S_{\hat{o}_1,\ldots,\hat{o}_n}(x_1, t_1;\cdots;x_n, t_n) = \lim_{\ell\to\infty}\ell^{n-1}\left\langle\!\left\langle\mathsf{o}_1[\underline{q}(x_1, t_1)]\cdots\mathsf{o}_n[\underline{q}(x_n, t_n)]\right\rangle\!\right\rangle_\ell^{\mathrm{c}}, \tag{50}$$

and in particular

$$F(\lambda, T) = \lim_{\ell\to\infty}\frac{1}{\ell T}\log\left\langle\!\left\langle\exp\lambda\ell J(T)\right\rangle\!\right\rangle_\ell, \tag{51}$$

where

$$J(T) = \int_0^T\mathrm{d}t\,\mathsf{j}_{i_*}[\underline{q}(0, t)], \tag{52}$$

see Eqs. (21) and (14). (Recall that the expansion of the right-hand side of (51) in powers of $\lambda$ boils down to correlation functions as in the right-hand side of (50).) We will write (50), for simplicity, as

$$\langle\bullet\rangle_\ell \sim \langle\!\langle\bullet\rangle\!\rangle_\ell. \tag{53}$$

Although we have formulated the theory using functional integration on appropriate measures, the result as $\ell\to\infty$ on the right-hand side of (53) is in fact obtained by taking a saddle point. For an observable $O[\underline{q}]$ in space-time,

$$-\lim_{\ell\to\infty}\ell^{-1}\log\left\langle\!\left\langle\exp\bigl(\ell O[\underline{q}]\bigr)\right\rangle\!\right\rangle_\ell = \mathcal{F}_O[\underline{q}^*], \tag{54}$$

where

$$\mathcal{F}_O[\underline{q}(\cdot,\cdot)] = \mathcal{F}[\underline{q}(\cdot,0)] - O[\underline{q}(\cdot,\cdot)], \tag{55}$$

is the "observable action", and $q^*$ is the minimiser of the "BMFT action"

$$S_O[\underline{q}, \underline{H}] = \mathcal{F}_O[\underline{q}] + \int_{\mathbb{R}} dx \int_0^T dt \, H^i(\partial_t q_i + \partial_x \mathsf{j}_i[\underline{q}]), \tag{56}$$

with respect to $\underline{q}$ and $\underline{H}$. Equivalently, one can see the fields $H^i(\cdot, \cdot)$ as Lagrange parameters enforcing the Euler equations viewed as constraints on the minimisation of the observable action $\mathcal{F}_O$. That is:

$$\left. \frac{\delta S_O[\underline{q}, \underline{H}]}{\delta q_i(x,t)} \right|_{\underline{q}=\underline{q}^*} = 0, \quad \partial_t q_i^* + \partial_x \mathsf{j}_i[\underline{q}^*] = 0. \tag{57}$$

We take $O = \lambda J(T)$ for the current SCGF, and $O = \sum_{a=1}^n \lambda_a \mathsf{o}_a[\underline{q}(x_a, t_a)]$ for the generator of Euler-scale correlation functions, with $n$ independent generating parameters $\lambda_a$. The set of equations (57) – the BMFT equations – will form the basis for our analysis below.

From Eq. (57), it is evident that the space-time configuration minimizing the BMFT action satisfies the Euler equation (although, as we will see, the initial condition is not $\mathsf{q}[\beta_{\mathrm{ini}}(x)]$, but a $\lambda$- or $\lambda_a$-dependent initial condition accounting for the rare fluctuations we are focusing on). We shall therefore use for the minimizer $\underline{q}^*$ the notation $\underline{q}^*(x,t) = \mathsf{q}(x,t)$. A sketch of the space-time configurations whose weight is given by the BMFT action in Eq. (56), together with the saddle point minimizing configuration, is given in Fig. 2.

We note that it is not necessary to have as initial state the form (12) (which gives the initial measure in factorised form (43)). The initial state must be slowly varying, but may otherwise have long-range correlations, thus giving a non-factorised initial measure. The BMFT principle, which simply says to evolve the initial fluctuating state with the Euler equation, will work all the same, with in (46) that measure instead of $e^{-\ell \mathcal{F}[\underline{q}(\cdot,0)]}$. However, the state (12), giving the measure (43), is natural to assume in local equilibrium, and is simple yet nontrivial enough for this presentation.

## 3.2 Local relaxation of fluctuations

The BMFT probability distribution, Eqs (46) and (49), results from a single assumption about fluctuations of mesoscopic variables. This is the assumption that

*Mesoscopic means of local observables (coarse-grained observables), $\bar{o}(\ell x, \ell t)$, do not fluctuate independently from the conserved densities, but are fixed functions of these, $o[\underline{q}(x,t)]$.*

That is, the form (50) must hold, for some, yet unknown, functions $o_a[\underline{q}(x,t)]$, which we show below must be the GGE average $\mathsf{o}_a[\underline{q}(x,t)]$. This means that the mesoscopic fluctuating degrees of freedom are *reduced to the conserved quantities*: all other degrees of freedom quickly relax, and only the "slowly-decaying modes", which are the extensive conserved quantities of the model, remain as fluctuating variables in the Euler scaling limit. In phase space, the interpretation is that the fixed-$q$ shell is quickly covered in fluid cells (as per the principle of ergodicity), and large-scale fluctuations are fluctuations between different shells.

The principle can be justified by a separation of fluctuation scales between variables that change due to interactions within the fluid cell, and those that are only affected by exchanges between fluid cells. Take the picture of classical particles with short-range interactions. If $\tau$ is the mean free time and $\rho$ the spatial density, then in a length $L$ and time $T$ there will be $\rho L T / \tau$ collisions (interactions). Thus an observable $(LT)^{-1} \int_0^L dx \int_0^T dt \, o(x,t)$ affected by the few-body interaction with nearby particles will have fluctuations due to in-cell process of

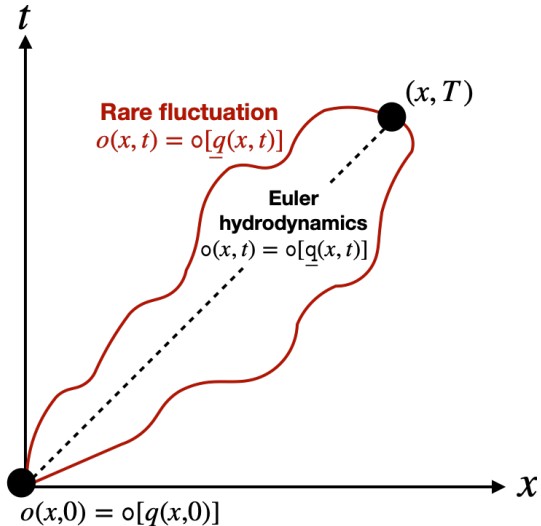

Figure 2: Space-time configurations (Feynman history) of fluctuating variables $o(x,t)$ that contribute to the BMFT path integrals (49). Due to local relaxation of fluctuations, the functional form of the classical fluctuating variables $o(x,t)$ on rare trajectories (sketched in red in the Figure) is completely determined by the fluctuating variables for the conserved densities $q(x,t)$ through the GGE expression, i.e., $o(x,t) = \mathsf{o}[\underline{q}(x,t)]$. The trajectory minimizing the BMFT action is sketched in black dashed. Along this trajectory, Euler hydrodynamics is recovered and the fluctuating variables $o(x,t)$ take the value given by the $\lambda$-dependent GGE at the corresponding space-time point $(x,t)$, i.e., $o(x,t) = \mathsf{o}[\underline{\mathsf{q}}(x,t)]$.

order $1/\sqrt{\rho LT/\tau}$. But conserved quantities $(LT)^{-1}\int_0^L \mathrm{d}x \int_0^T \mathrm{d}t\, q_i(x,t)$ are only affected by exchange of particles through the boundaries of the cells, and exchange of energy and other quantities by interactions through these boundaries. Naturally, such processes take place less frequently than the in-cell processes, giving contributions to fluctuations only of order $1/\sqrt{T/\tau}$ and $1/\sqrt{\rho\ell_{\mathrm{micro}}T/\tau}$, respectively. Therefore, with $T \propto L$ inter-cell fluctuations of fluid-cell means $\Delta_{\mathrm{inter\text{-}cell}}\overline{o} \propto 1/\sqrt{L}$ dominate over in-cell fluctuations $\Delta_{\mathrm{in\text{-}cell}}\overline{o} \propto 1/L$, and as $\Delta_{\mathrm{in\text{-}cell}}\overline{q_i} = 0$, in-cell processes simply cover the phase-space shell with fixed values of $\overline{q_i}$'s.

Tacitly assumed in the above assumption is that fluid-cell means of observables can be seen as classical random variables, in agreement with the general expectation that the Euler-scale correlation functions are symmetric as mentioned above. In classical systems this is clearly the case. In quantum systems, this assumes that such fluid-cell means become, in the Euler scaling limit, classical fluctuating variables. The idea is that such observables are examples of the "macroscopic observables" introduced by von Neumann in the context of the quantum ergodic theorem [79] (see an analysis of this paper in [80]). An ergodicity result has recently been shown [81], which implies that, in a large class of stationary, homogeneous states of quantum lattices with short-range Hamiltonians, averages of observables over space-time indeed project onto the state average times the identity operator; this further support taking fluid-cell means as classical variables.

The principle of local relaxation of fluctuations is of course closely related to the principle used normally to justify hydrodynamics: the hydrodynamic equations can be derived by assuming that state averages of fluid-cell means of local observables are functions of local conserved densities only. As this must be true in (G)GE's, then the functions are fixed to

(G)GE averages. As this holds in particular for the currents, the Euler equations follow.

The concept of relaxation of fluctuations goes one step further, and makes this assumption for the large-scale fluctuations. It turns out again that the functions $o[\underline{q}]$ are uniquely fixed to the ensemble averages $\mathsf{o}[\underline{q}]$ by this assumption. Indeed, Eqs. (25) at $t = 0$, and (50) at $n = 1$ and $t_1 = 0$, imply

$$\langle \hat{o} \rangle_{\underline{\beta}(x,0)} = \int_{(\mathbb{R})} d\mathbb{P}_{\text{ini}}[\underline{q}(\cdot,0)] \, o[\underline{q}(x,0)], \tag{58}$$

hence by the saddle-point (45) we obtain (48). Of course, this is nothing else but the equivalence of macrocanonical and microcanonical ensembles: the fast relaxation over the fixed-$\underline{q}$ shell gives (microcanonical) values of observables that are functions of the $q_i$'s as per their (macrocanonical) GGE averages.

Local relaxation of fluctuations has appeared in various forms in the literature. Most importantly, the Boltzmann-Gibbs principle is often stated as a projection of fluctuating fields onto fluctuating conserved fields in stochastic particle systems, see, e.g., [26, 27].

With local relaxation of fluctuations, and in particular Eq. (48), it is a simple matter to obtain the BMFT formula. This again parallels the derivation of the Euler hydrodynamic equations. As, by the equations of motion, the conservation laws must be satisfied, we must have

$$\langle \bullet \rangle_\ell = \frac{\left\langle e^{\ell \int_{\mathbb{R}} dx \int_0^T dt \, H^i(x,t)(\partial_t \bar{q}_i + \partial_x \bar{j}_i)} \bullet \right\rangle_\ell}{\left\langle e^{\ell \int_{\mathbb{R}} dx \int_0^T dt \, H^i(x,t)(\partial_t \bar{q}_i + \partial_x \bar{j}_i)} \right\rangle_\ell}, \tag{59}$$

for all $\ell$, and any values of the fields $H^i(x,t)$. Note that variations of densities and currents occur on scales $\ell$, hence with a factor $\ell^2$ from the space-time integral and $\ell^{-1}$ from the derivatives, we indeed have a factor $\ell$ in the exponential; it is made explicit here as integrals are on macroscopic variables $x, t$. The principle of local relaxation of fluctuations at all times $t$ implies that there is some measure $d\mathbb{P}[\underline{q}(\cdot, \cdot)]$ representing averages $\langle \bullet \rangle_\ell$ at large $\ell$,

$$\langle \bullet \rangle_\ell \sim \int_{(\mathbb{S})} d\mathbb{P}[\underline{q}(\cdot, \cdot)] \bullet, \tag{60}$$

and this can be used on (59). With the result (48) for the observables $\hat{j}_i$, we deduce that

$$d\mathbb{P}[\underline{q}(\cdot, \cdot)] \propto d\mathbb{P}[\underline{q}(\cdot, \cdot)] e^{\ell \int_{\mathbb{R}} dx \int_0^T dt \, H^i(x,t)(\partial_t q_i + \partial_x \mathsf{j}_i[\underline{q}])}. \tag{61}$$

Assuming that the hyperbolic system (11) has a unique solution – or imposing appropriate conditions on possible weak solutions to make it unique – the arbitrariness of $H^i(x,t)$ implies that $d\mathbb{P}[\underline{q}(\cdot, \cdot)]$ is supported on this solution. Thus it is sufficient to put a measure on the initial condition, hence to know the marginal at time $t = 0$. As the initial measure $d\mathbb{P}_{\text{ini}}[\underline{q}(\cdot, 0)]$ must be this marginal, we obtain (46) (equivalently (49)), where the flat integration measure over all times with the delta functional guarantee both the right initial-time marginal and the support on the hydrodynamic solution. This shows (46) - (51).

## 3.3 Relation to diffusive MFT and fluctuating hydrodynamics

The goal of this section is to provide further justifications for the BMFT, by proposing how it relates to the well-known and well-established theory of diffusive MFT [7], and by making a parallel with (nonlinear) fluctuating hydrodynamics (NLFHD) [15]. The conventional MFT

gives a probability distribution for mesoscopic variables in purely diffusive systems, and of course BMFT is largely inspired by it.

On the one hand, as explained above, the underlying idea of the BMFT, the local relaxation of fluctuations, implies that the mesoscopic currents $\underline{j}$ do not fluctuate independently, but rather are random variables given, as functions of fluctuating conserved densities, by the stationary (GGE) values, $\underline{j} = \underline{\mathsf{j}}[\underline{q}]$. On the other hand, in diffusive MFT, the currents fluctuate, via a stochastic, Gaussian noise determined by microscopic diffusion via the Einstein relation (see [7]). Both are, nevertheless, "macroscopic fluctuation theories", describing large deviations of fluid trajectories obtained by saddle-point analysis. How are these two theories related?

In order to better understand this, we propose here a multi-scale hydrodynamic fluctuation theory that covers both ballistic and diffusive effects. We will see that BMFT is obtained *under ballistic scaling, in the limit of zero noise and zero diffusion*, while diffusive MFT is obtained *under diffusive scaling, with ballistic currents set to zero*. The resulting saddle-point analysis, in BMFT and diffusive MFT, is therefore done under *different choices of scaling*.

The general idea of combining ballistic and diffusive effects is not new: it was used in [8] to analyse the weakly asymmetric simple exclusion process, and, by a zero-noise limit, to obtain a ballistic fluctuation theory for the totally asymmetric simple exclusion process. But as far as we are aware, a universal theory has not been constructed yet.

Formulated in terms of the ballistic scale $\ell$, our ansatz is a theory for fluctuating classical variables that reproduce not only the leading large-$\ell$ order of correlation functions such as (21) (the Euler scale), but also the *next-to-leading order*. The measure $d\mathbb{P}[\underline{q}(\cdot, \cdot)]$ must be modified in order to reproduce correctly both these leading and next-to-leading orders. This multi-scale hydrodynamic fluctuation theory is no longer a large-deviation theory, as its analysis would require going beyond saddle-point equations and considering "loop corrections", which we reserve for future works. In a sense, it is an action re-formulation for NLFHD, as it combines both ballistic and diffusive scales (recall that their combination, in NLFHD, helps explaining superdiffusive behaviours of two-point correlation functions.) The ideas proposed here, besides being applicable to general multi-component hydrodynamic systems, are also slightly different from those of [8].

Recall that the main assertions of the BMFT is the measure on the mesoscopic densities (46). The macroscopic fluctuation theory that covers both the ballistic and diffusive orders, instead, is a measure on space-time configurations $(\underline{q}(x,t), \underline{j}(x,t)) : (x,t) \in \mathbb{R} \times [0,T]$ of both *the mesoscopic charge densities and currents*. The measure accounts for fluctuations of the currents that, although suppressed in the limit $\ell \to \infty$, give small contributions for $\ell$ large that describe the next-to-leading order corrections to the Euler-scale correlation functions. It is natural to introduce independent fluctuations of fluid-cell means of observables at that order. Indeed, the argument above concerning the separation of scales, with fluctuations of order $\Delta_{\text{in-cell}}\overline{o} \propto 1/L$ and $\Delta_{\text{inter-cell}}\overline{o} \sim 1/\sqrt{L}$, indicates that at the next order, in-cell fluctuations, which affect non-conserved observables, and not just inter-cell fluctuations, by which non-conserved observables follow conserved ones, must be taken into account. As discussed above, corrections to large-scale correlations induced by such additional fluctuations may be diffusive, superdiffusive, etc. Here, we will not develop the full phenomenology, but only express the measure, emphasising how it extends BMFT to including microscopic noise, and how it specialises in the case without ballistic currents to the standard equations of the MFT under the right scaling.

The measure encoding ballistic and diffusive behaviours is obtained by *adding Gaussian*

*noise contributions to the currents.* As explained, this represents the small, rapid fluctuations of quickly relaxing modes that are not protected by conservation laws, occurring within microcanonical shells, on top of the large fluctuations due to fluctuating mesoscopic charges. That is,

$$d\mathbb{P}[(\underline{q},\underline{j})(\cdot,\cdot)] = d\mu[\underline{q}(\cdot,\cdot)]d\mathbb{P}[\underline{j}|\underline{q}]\,e^{-\ell\mathcal{F}[\underline{q}(\cdot,0)]}\delta[\partial_t\underline{q}+\partial_x\underline{j}], \tag{62}$$

where $d\mathbb{P}[\underline{j}|\underline{q}] = d\mathbb{P}_{\text{noise}}[\ell\mathbf{j}.-\ell\mathbf{j}.[\underline{q}]+\mathfrak{D}.^j[\underline{q}]\partial_x q_j|\underline{q}]$ is obtained from adding noise to the diffusive-order constitutive relation for the currents,

$$j_i = \mathsf{j}_i[\underline{q}] + \ell^{-1}\big(-\mathfrak{D}_i{}^j[\underline{q}]\partial_x q_j + \eta_i\big). \tag{63}$$

The measure $d\mathbb{P}_{\text{noise}}[\underline{\eta}|\underline{q}]$ on the zero-mean Gaussian noise $\eta_i$ is fully determined by the correlation matrix

$$\big\langle\eta_i(x,t)\eta_j(x',t')\big\rangle_{\text{noise}} = \mathfrak{L}_{ij}[\underline{q}]\delta(x-x')\delta(t-t'). \tag{64}$$

The factors of $\ell$ in (63) are obtained from the same scaling as above,

$$\overline{q}_i(\ell x,\ell t) \equiv q_i(x,t), \quad \overline{j}_i(\ell x,\ell t) \equiv j_i(x,t). \tag{65}$$

The fluid-cell means $\overline{q}_i$, $\overline{j}_i$ on the mesoscopic scale $L$ must be defined carefully, so as not to "wash out" diffusive effects, hence we propose

$$\ell_{\text{micro}} \ll L \ll \ell_{\text{diff}} \ll \ell, \quad \ell_{\text{diff}} := \sqrt{\ell}. \tag{66}$$

Note that now $j_i(x,t)$ fluctuates independently, through its associated noise (of course, the full measure is still supported on space-time configurations that satisfy the continuity equation $\partial_t q_i + \partial_x j_i = 0$). Above, $\mathfrak{L}_{ij}$ is the "microscopic Onsager matrix", a positive definite matrix representing local Gaussian fluctuations of the microscopic currents. The diffusion matrix $\mathfrak{D}_i{}^j$ is related to the Onsager matrix via the Einstein relation $\mathfrak{D} = \mathfrak{L}\mathsf{C}^{-1}$.

When corrections to the Euler scale of correlation functions are indeed diffusive, and not superdiffusive, the microscopic Onsager matrix can be evaluated from correlation functions in stationary states via the Green-Kubo formula $\mathfrak{L}_{ij} = \int_0^\infty dt\int_{\mathbb{R}} dx\langle\hat{j}_i^-(x,t)\hat{j}_j^-(0,0)\rangle_{\underline{\beta}}^c$ (at $q_i = \mathsf{q}_i[\underline{\beta}]$), where $\hat{j}_i^- = \hat{j}_i - \mathsf{A}_i{}^j\hat{q}_j$ is the current minus its projection onto the conserved densities. The microscopic Onsager matrix is no longer simply given by the Green-Kubo formula when the correction to ballistic transport is superdiffusive. This is in fact the most typical case in the Hamiltonian dynamics of non-integrable translation invariant systems; see [82] for a recent proposal on the microscopic Onsager matrix that should be used in such situations in the fluctuating hydrodynamics. In integrable systems, typically corrections to ballistic transport are indeed diffusive [64, 83].

Naturally, one may ask why not introduce noise to other observables $o(x,t)$ than the currents themselves, with noise correlations of the form (64) and Onsager-like coefficients involving these other observables, $\mathfrak{L}_{\hat{j}_i,\hat{o}}$ and $\mathfrak{L}_{\hat{o},\hat{o}'}$. Generic observables are not directly involved in the measure (62), however if noise correlations exist between generic observables and currents, these will affect the currents and must be included. Such effects appear not to have been discussed in the literature, neither in the context of fluctuating hydrodynamics nor of the MFT. A possible resolution is that, according to the Hilbert space theory for diffusive and thermalising processes developed in [84, 85], it is possible to separate generic observables into a linear combination of currents $\hat{j}_i$ and a part $\hat{o}$ that is orthogonal to currents within the "second order hydrodynamic space" (meaning $\mathfrak{L}_{\hat{j}_i,\hat{o}} = 0$). This would mean that their noises

do not correlate to those of currents, and hence it is safe to only consider fluctuating currents in the measure (62). A complete understanding is certainly missing.

As mentioned, the proposal (62) with (63) is to describe not only the ballistic scale of correlation functions, but also their leading correction. If these are diffusive, a simple example is the correlator $\lim_{\ell\to\infty} \ell^{-\frac{1}{2}} \left\langle\!\!\left\langle q_i(x+\ell^{-\frac{1}{2}}\delta x,t)q_j(0,0) \right\rangle\!\!\right\rangle_\ell^{\mathrm{c}}$ around a ballistic ray of velocity $v = x/t$ in a stationary, homogeneous initial state. The result is a function of $v$ and $\delta x$ that describes the diffusive profile of the correlator around this ray; of course, it is nontrivial if and only if $v$ is one of the hydrodynamic velocities of the model in that state, $v \in \mathrm{spec}(\mathsf{A})$. More generally, the theory (62) with (63) leads to the basic noisy-current stochastic equations for two-point correlation functions (thus, by the NLFHD analysis [15], under the appropriate conditions these equations explain superdiffusion). We expect the theory to also predict the leading corrections to long-time saturation of the transport cumulants $c_n$ in (15) (see also the discussion in Sec. 5).

It is clear from Eq. (62) with (63) that the leading order at large $\ell$ reduces to the BMFT. Effectively, the BMFT is the zero-noise, zero-diffusion limit (of course, physical diffusion does not need to vanish for the BMFT to hold, its effects are simply at a smaller scale than that observed by the BMFT). But also, under appropriate rescaling and with vanishing ballistic currents, one recovers from Eqs. (62) and (63) the standard diffusive MFT [7].

To see this, recall that in diffusive MFT, one concentrates on the diffusive scale $\ell_{\mathrm{diff}} = \sqrt{\ell}$. Let us thus define new scaled variables as

$$\bar{q}_i(\ell_{\mathrm{diff}}x, \ell_{\mathrm{diff}}^2 t) \equiv \check{q}_i(x,t), \quad \ell_{\mathrm{diff}}\bar{\bar{j}}_i(\ell_{\mathrm{diff}}x, \ell_{\mathrm{diff}}^2 t) \equiv \check{j}_i(x,t), \tag{67}$$

where the extra factor on the current guarantees that the continuity equation stays unchanged. Further, in purely diffusive systems, the hydrodynamic velocities vanish. Let us thus take $\mathsf{j}_i = 0$[6]. In fact, with external fields $E^i$, constant currents develop at the diffusive scale, determined by Onsager's coefficients. Accounting for these, one may instead make the replacement in Eq. (63)

$$\mathsf{j}_i[\underline{q}] \to \ell_{\mathrm{diff}}^{-1}\mathfrak{L}_{ij}[\check{q}]E^j. \tag{68}$$

In order to get the theory in the diffusive scaling, we start from the fundamental equation for the ballistically scaled quantities (63) (with this replacement), we rewrite them in terms of the fundamental microscopic quantities as per (65), and we further rewrite these in terms of diffusively scaled quatities as per (67). Using the scaling property $\eta_i(x/\ell_{\mathrm{diff}}, t) = \sqrt{\ell_{\mathrm{diff}}}\eta(x,t)$, the diffusively scaled fluctuating currents then take the form

$$\check{j}_i = \mathfrak{L}_{ij}[\check{\underline{q}}]E^j - \mathfrak{D}_i^{\ j}[\check{\underline{q}}]\partial_x\check{q}_j + \ell_{\mathrm{diff}}^{-\frac{1}{2}}\eta_i. \tag{69}$$

Writing explicitly the Gaussian measure for the noise, one then obtains

$$\mathrm{d}\mathbb{P}[(\check{\underline{q}}, \check{\underline{j}})(\cdot, \cdot)] = \mathrm{d}\mu[\check{\underline{q}}(\cdot, \cdot)]\mathrm{d}\mu[\check{\underline{j}}(\cdot, \cdot)]\, e^{-\ell_{\mathrm{diff}}(\mathcal{F}[\check{\underline{q}}(\cdot, 0)] + \mathcal{I}[\check{\underline{q}}, \check{\underline{j}}])}\delta[\partial_t\check{\underline{q}} + \partial_x\check{\underline{j}}], \tag{70}$$

where the functional $\mathcal{I}[\check{\underline{q}}, \check{\underline{j}}]$ is given by

$$\mathcal{I}[\check{\underline{q}}, \check{\underline{j}}] = \int_0^T \mathrm{d}t \int_{\mathbb{R}} \mathrm{d}x\, \mathfrak{L}^{ij}[\check{\underline{q}}](\check{j}_i - \check{j}_{\mathrm{diff},i}[\check{\underline{q}}, \partial_x\check{\underline{q}}])(\check{j}_j - \check{j}_{\mathrm{diff},j}[\check{\underline{q}}, \partial_x\check{\underline{q}}]), \tag{71}$$

---

[6]It is also possible, by a simple ballistic shift, to concentrate on diffusive effects around other velocities than 0.

with

$$\check{j}_{\text{diff},i} = \mathfrak{L}_{ij}[\underline{\check{q}}]E^j - \mathfrak{D}_i^{\ j}[\underline{\check{q}}]\partial_x\check{q}_j. \tag{72}$$

This is exactly the diffusive MFT probability distribution on space-time configurations $(\check{q}(x,t),\check{j}(x,t)) : (x,t) \in \mathbb{R} \times [0,T]$ of Ref. [7], written here in its most general, multi-component form.

## 4 BMFT for current fluctuations and Euler-scale correlations

With a firm foundation for the BMFT, we move on to its actual implementation to generic quantum or classical many-body systems. In Subsec. 4.1, we focus on the SCGF $F(\lambda,T)$ defined in Eq. (14). In Subsec. 4.2, we discuss how the ballistic MFT allows to recover the result of the aforementioned BFT theory for homogeneous and stationary states. In Subsec. 4.3, the derivation of the Gallavotti-Cohen fluctuation theorem (19) within the BMFT formalism is detailed. In Subsec. 4.4, we discuss Euler-scale correlation functions $S_{\hat{q}_{i_1},\hat{q}_{i_2}}(x_1,t_1;x_2,t_2)$, defined in Eqs. (21)-(23). In Subsec. 4.5, we show how the BMFT formalism naturally embodies the (non)linear hydrodynamic projection result (see Eq. (26)). In Subsec. 4.6, we show how the BMFT predicts long-range correlations.

### 4.1 BMFT of current fluctuations

The BMFT offers us an efficient way of evaluating the SCGF $F(\lambda,T)$, as $\left\langle\!\left\langle e^{\lambda\ell J(T)}\right\rangle\!\right\rangle_\ell \asymp e^{\ell T F(\lambda,T)}$. According to the discussion in Subsec. 3.1, cf. Eqs. (51)-(54), we have

$$\left\langle\!\left\langle e^{\lambda\ell J(T)}\right\rangle\!\right\rangle_\ell \asymp e^{-\ell\mathcal{F}_{\text{curr}}[q^*]}, \tag{73}$$

where $q^*$ minimise

$$S_{\text{curr}}[\underline{q},\underline{H}] := \mathcal{F}_{\text{curr}}[q] + \int_{\mathbb{S}}\mathrm{d}x\mathrm{d}t\, H^i(\partial_t q_i + \partial_x\mathsf{j}_i[\underline{q}]), \tag{74}$$

and $\mathcal{F}_{\text{curr}}[q(\cdot,\cdot)] = \mathcal{F}[\underline{q}(\cdot,0)] - \lambda J(T)$. The saddle-point equations (57) yield the following BMFT equations (dropping the star symbol $^*$ for lightness of notation)

$$H^i(x,0) = \beta^i_{\text{ini}}(x) - \beta^i(x,0), \tag{75a}$$

$$H^i(x,T) = 0, \tag{75b}$$

$$\partial_t\beta^i + \mathsf{A}_j^{\ i}[\underline{\beta}]\partial_x\beta^j = 0, \tag{75c}$$

$$\partial_t H^i + \mathsf{A}_j^{\ i}[\underline{\beta}]\partial_x H^j = -\lambda\delta(x)\mathsf{A}_{i_*}^{\ i}[\underline{\beta}], \tag{75d}$$

where we used $\mathsf{AC} = \mathsf{CA}^{\mathrm{T}}$ [28] and in (75c) and (75d) we have removed the explicit $x,t$ dependence of $\beta^i(x,t)$ and $H^i(x,t)$. Note that for convenience the BMFT equations are written in terms of $\beta^i$, which are related to the densities via the usual mapping to GGE averages, $q_i(x,t) = \mathsf{q}_i[\underline{\beta}(x,t)]$. As remarked after Eq. (57), we also drop the dependence of $H$ and $\beta^i$ on $\lambda$ to lighten the notation. One can recast these equations into a more useful form by redefining the auxiliary field $H^i$ by $H^i(x,t) \mapsto H^i(x,t) - \lambda\delta^i_{\ i_*}\Theta(x)$ with the step function

$\Theta(x)$, yielding

$$\lambda \delta_{i_*}{}^i \Theta(x) - \beta^i(x,0) + \beta^i_{\text{ini}}(x) - H^i(x,0) = 0, \tag{76a}$$

$$\lambda \delta_{i_*}{}^i \Theta(x) - H^i(x,T) = 0, \tag{76b}$$

$$\partial_t \beta^i(x,t) + \mathsf{A}_j{}^i[\underline{\beta}(x,t)] \partial_x \beta^j(x,t) = 0, \tag{76c}$$

$$\partial_t H^i(x,t) + \mathsf{A}_j{}^i[\underline{\beta}(x,t)] \partial_x H^j(x,t) = 0, \tag{76d}$$

which are the main equations we shall deal with. The boundary conditions associated to Eq. (76) are $\beta^i(x \to \pm\infty, t) = \beta^i_{\text{ini}}(x \to \pm\infty)$, $H^i(x \to +\infty, t) = \lambda \delta_{i_*}{}^i$ and $H^i(x \to -\infty, t) = 0$ for any value of $t$. The re-writing in Eq. (76) is equivalent to expressing the time-integrated current in the BMFT action in terms of the density $J(T) = \int_0^\infty \mathrm{d}x\, (q_{i_*}(x,T) - q_{i_*}(x,0))$ (see also after Eq. (14)) from the beginning.

The SCGF may be written in various ways. Clearly using (43) and (55), $TF(\lambda, T) = \mathcal{F}[q(\cdot,0)] - \lambda \int_0^T \mathrm{d}t\, \mathsf{j}_{i_*}[\underline{q}(0,t)]$ evaluated on the solution to (76a)-(76d). But also, note that (in a self-explanatory notation)

$$\frac{\mathrm{d}}{\mathrm{d}\lambda} TF(\lambda, T) = -\frac{\mathrm{d}\mathcal{F}_{\text{curr}}[\underline{q}]}{\mathrm{d}\lambda} = -\frac{\mathrm{d}S_{\text{curr}}[\underline{q}, \underline{H}]}{\mathrm{d}\lambda} = J(T), \tag{77}$$

as $S_{\text{curr}}$ is stationary under changes of $H$ and $q$ on the saddle point. Therefore, integrating over $\lambda$,

$$F(\lambda, T) = \frac{1}{T} \int_0^\lambda \mathrm{d}\lambda' \int_0^T \mathrm{d}t\, \mathsf{j}_{i_*}[\underline{q}^{(\lambda')}(0,t)], \tag{78}$$

where $q_i^{(\lambda')}(0,t) = \mathsf{q}_i[\underline{\beta}^{(\lambda')}(0,t)]$ is the solution to (76a)-(76d) at $x = 0$, for $\lambda$ replaced by $\lambda'$ (with the notation introduced after Eq. (57)). Expression (78) has a similar form to what was obtained in the BFT [20, 21, 23] (see also the discussion in Appendix D about the inhomogeneous BFT).

We note that $\underline{\beta}(x,t)$ (or equivalently $\underline{q}(x,t)$) time-evolve independently from $\underline{H}(x,t)$, and the only effect of having nontrivial $\underline{H}(x,t)$ enters into the boundary conditions. This is in stark contrast with the diffusive cases (see, e.g., [10]).

As is usual in solving Euler equations, weak solutions [86] may appear from the BMFT equations, for both $\underline{\beta}(x,t)$ and $\underline{H}(x,t)$; therefore, ambiguities may arise. Much like for Euler hydrodynamic equations, one may need to re-introduce diffusive effects in order to obtain the equivalent of Lax conditions for the BMFT equations, and thus determine the correct solution for $\underline{\beta}(x,t)$ and $\underline{H}(x,t)$. Such considerations will be relevant in the application of the BMFT, for instance, to the anharmonic chain, whose hydrodynamics is composed of three conservation laws [15], and to the TASEP, whose hydrodynamics is the inviscid Burgers equation. Of importance may be the fact that, in the Jensen-Varadhan theory [37,38], the contributions to the rate function stem only from hydrodynamic configurations that give positive Kruzhkov entropy productions (that is, negative physical entropy production, contrary to what is required for usual hydrodynamic solutions). We leave this for future studies.

## 4.2 Relation with the ballistic fluctuation theory

The BFT [20, 21], briefly recalled in the introduction, gives the time-independent SCGF $F(\lambda, T) = F(\lambda)$ in the case where the initial state $\beta_{\text{ini}}(x) = \beta_{\text{ini}}$ is homogeneous and stationary.

In this theory, the SCGF is written in a form similar to (78), $F(\lambda) = \int_0^\lambda d\lambda'\, \mathsf{j}_{i_*}[\underline{q}(\lambda')]$, but now $q_i(\lambda) = \mathsf{q}_i[\underline{\beta}(\lambda)]$ describes a stationary, homogeneous state satisfying the *BFT flow equation*

$$\partial_\lambda \beta^i(\lambda) = -\mathrm{sgn}(\mathsf{A}[\underline{\beta}(\lambda)])_{i_*}^{\ i}, \quad \beta^i(0) = \beta^i_{\mathrm{ini}}. \tag{79}$$

We note that Eq. (79) is proved solely on the basis of hydrodynamic projection techniques and therefore applies generally, both to integrable and non-integrable systems.

As a verification of the ballistic MFT developed here, we argue that it reproduces the above BFT result. In order to do so, we would only need to establish that $\underline{\beta}^{(\lambda)}(0, t) = \underline{\beta}(\lambda)$ for all $t \in (0, T)$. As $\underline{\beta}^{(0)}(x, t) = \underline{\beta}_{\mathrm{ini}}$, we would only need to show that $\underline{\beta}^{(\lambda)}(0, t)$ is independent of $t$ and satisfies the BFT flow equation (79). This would require a precise analysis of the solution to the BMFT equations, and is related to the assumption, made in [20], that connected time correlation functions of all orders vanish sufficiently fast at large times. We keep this precise analysis for future works, and instead provide a consistent argument.

Under the assumption that $\underline{\beta}^{(\lambda)}(0, t)$ is independent of $t \in (0, T)$, it is clear from (76c) that at $x = 0$, the state is homogeneous, including at the leading order beyond the Euler scale, as $\partial_x \beta^{(\lambda)}(x, t)|_{x=0} = 0$. Let us concentrate on a region $t \in [s - r, s + r]$, $x \in [-y, y]$ where $r > 0$ and $y > 0$ are finite but small (in macroscopic units). Let us further assume that, for a given $\lambda$, the homogeneous, stationary state there (as we argued it must be), is in fact also clustering, thus it is a GGE. As a consequence, we may apply the BMFT on this region, where as initial state in (43) we take $\underline{\beta}_{\mathrm{ini}} = \underline{\beta}^{(\lambda)}(0, s)$. Now let us perturb $\lambda \to \lambda + \delta\lambda$, and consider the BMFT for the insertion of $\overline{\delta\lambda J(T)}$. The leading-order BMFT equations are as in (76a)-(76d), but in terms of the leading-order quantities in $\delta\beta^i(x, t) = \beta^i(x, t) - \beta^i(0, s)$ (which are assumed small as $\delta\lambda$ is taken small), for $x \in [-y, y]$ and $t \in [s - r, s + r]$,

$$\delta\lambda\, \delta_{i_*}^{\ i} \Theta(x) - \delta\beta^i(x, s - r) - \delta H^i(x, s - r) = 0, \tag{80a}$$

$$\delta\lambda\, \delta_{i_*}^{\ i} \Theta(x) - \delta H^i(x, s + r) = 0, \tag{80b}$$

$$\partial_t \delta\beta^i(x, t) + \mathsf{A}_j^{\ i} \partial_x \delta\beta^j(x, t) = 0, \tag{80c}$$

$$\partial_t \delta H^i(x, t) + \mathsf{A}_j^{\ i} \partial_x \delta H^j(x, t) = 0, \tag{80d}$$

where $\mathsf{A} = \mathsf{A}[\underline{\beta}^{(\lambda)}(0, s)]$. This is a linear system, whose solution is simple:

$$\delta H^i(x, t) = \delta\lambda\, \Theta(x - (t - s - r)\mathsf{A})_{i_*}^{\ i}, \tag{81}$$

and then, using $\delta H^i(x, s - r) = \delta\lambda\, \Theta(x + 2r\mathsf{A})_{i_*}^{\ i}$

$$\delta\beta^i(x, t) = \delta\lambda\, [\Theta(x - (t - s + r)\mathsf{A})_{i_*}^{\ i} - \Theta(x - (t - s - r)\mathsf{A})_{i_*}^{\ i}]. \tag{82}$$

Thus $\delta\beta^i(0, s) = -\delta\lambda\, \mathrm{sgn}(\mathsf{A})_{i_*}^{\ i}$, which indeed reproduces (79).

In view of the long-range correlations that appear, as mentioned, generically in inhomogeneous states, the assumption that the state at $(0, s)$ is indeed clustering is the most nontrivial. However, explicit results for the cumulants in Sec. 5 show that the BMFT indeed agrees with the BFT in integrable systems.

The BFT was also extended to inhomogeneous states (12) for integrable systems in [23]. We discuss this in light of the present understanding in Appendix D.

### 4.3 Fluctuation theorem

We now show that the BMFT equations (75a)-(75d) provide an intuitively clear understanding as to how the Gallavotti-Cohen fluctuation theorem (GCFT) is realised in many-body systems [4,5]. To unveil the full symmetry stemming from a time-reversal symmetry of Euler hydrodynamics, we consider in this Subsection the generalised SCGF (18). Accordingly the equation for $H^i$ of the BMFT equations (75d) is altered to

$$\partial_t H^i + \mathsf{A}_j^{\ i}[\underline{\beta}]\partial_x H^j = -\delta(x)\sum_{j\in\mathcal{C}}\lambda^j\mathsf{A}_j^{\ i}[\underline{\beta}]. \tag{83}$$

In the following, we assume that the solution to (75a)-(75d) are continuous solutions, and not weak solutions under certain entropy or Lax condition (see the discussion in Subsec 4.1). We will then discuss briefly what steps may fail for weak BMFT solutions. We recall that for integrable systems, the BMFT solutions are continuous and well behaved, hence the proof below applies.

  Recall that the GCFT manifests itself as a particular symmetry of the SCGF (14), in the partitioning protocol, where by scale invariance, we can set $T = 1$ and $F(\underline{\lambda}, T) = F(\underline{\lambda})$. The symmetry relation is Eq. (19). This is normally seen to originate from a relation between the probability associated to time-forward and time-reversed trajectories. Our argument from the BMFT shows that in fact the theorem holds from a weaker version of time-reversal invariance, and for any many-body systems which admits an Euler hydrodynamic description, independently from the details of the microscopic dynamics (be it classical or quantum, stochastic or deterministic). The time-invariance requirement is the existence of a "time-reversal" involution $\mathcal{T}$ of the algebra of observables, $\mathcal{T}(o_1 o_2) = \mathcal{T}(o_1)\mathcal{T}(o_1)$, $\mathcal{T}\circ\mathcal{T} = 1$, with the only requirements that the *a priori* measure $\mu$ be invariant $\mu\circ\mathcal{T} = \mu$, that it acts on charge densities and currents as follows:

$$\mathcal{T}(\hat{q}_i(x,t)) := \mathcal{S}_i\hat{q}_i(x,1-t), \tag{84a}$$
$$\mathcal{T}(\hat{j}_i(x,t)) := -\mathcal{S}_i\hat{j}_i(x,1-t), \tag{84b}$$

for $\mathcal{S}_i \in \{+1,-1\}$, and that the charges $\hat{Q}_i$ whose (time-integrated) currents are put in the generalised SCGF (18) be invariant, $\mathcal{S}_i = 1$ for all $i \in \mathcal{C}$. The sign $\mathcal{S}_i$ is the parity of the charge $\hat{Q}_i$, taking values 1 (resp. $-1$) if it is time-reversal even (resp. odd). Note that the relation for the current (84b) (where the extra minus sign appears) is not an additional constraint: it is a consequence of the relation for the densities along with the continuity equation.

  The argument is as follows. In the partitioning protocol, the initial state (17) is

$$\beta_{\mathrm{ini}}^i(x) = \begin{cases} \Theta(x)\beta_R^i + \Theta(-x)\beta_L^i & (i\in\mathcal{C}) \\ \beta_0^i & \text{(otherwise)}. \end{cases} \tag{85}$$

First, we make two variable changes. In (83) we write

$$\lambda^i = \beta_L^i - \beta_R^i - \tilde{\lambda}^i \tag{86}$$

for all $i \in \mathcal{C}$. We then note that the extra delta-function terms that this brings can be cancelled by a shift of the functions $H^i(x)$ by $\beta_{\mathrm{ini}}^i(x)$. In order to also exchange the initial and final conditions (75a), (75b), we further shift by $-\beta^i(x,t)$ and change the sign,

$$H^i(x,t) \to \beta_{\mathrm{ini}}^i(x) - \beta^i(x,t) - H^i(x,t). \tag{87}$$

Combining with (75c), the shift does not affect the left-hand side of (83), and hence the equations stay of the same form. Thus

$$H^i(x, 0) = 0, \tag{88a}$$

$$H^i(x, 1) = \beta^i_{\text{ini}}(x) - \beta^i(x, 1), \tag{88b}$$

$$\partial_t \beta^i + \mathsf{A}_j{}^i[\underline{\beta}]\partial_x \beta^j = 0, \tag{88c}$$

$$\partial_t H^i + \mathsf{A}_j{}^i[\underline{\beta}]\partial_x H^j = -\delta(x) \sum_{j \in \mathcal{C}} \tilde{\lambda}^j \mathsf{A}_j{}^i[\underline{\beta}]. \tag{88d}$$

Second, we use the dynamical input of time-reversal invariance, (84). This implies the following identities involving GGE averages

$$\mathsf{q}_i = \mathcal{S}_i \tilde{\mathsf{q}}_i, \quad \mathsf{j}_i = -\mathcal{S}_i \tilde{\mathsf{j}}_i, \tag{89}$$

where $\tilde{\mathsf{o}} := \langle \hat{o} \rangle_{\tilde{\underline{\beta}}}$ with $\tilde{\beta}_i = \mathcal{S}_i \beta^i$ (no sum over repeated indices). These identities give rise in particular to a relation for the flux Jacobian,

$$\mathsf{A}_i{}^j[\underline{\beta}] = -\mathcal{S}_i \mathcal{S}_j \mathsf{A}_i{}^j[\tilde{\underline{\beta}}]. \tag{90}$$

Relation (90), as well as

$$\mathcal{S}_i = 1, \quad \mathsf{j}_i = -\tilde{\mathsf{j}}_i, \tag{91}$$

for all $i \in \mathcal{C}$ are in fact the only dynamical relations that are needed.

With these relations, we can now make the final, time-reversal change of variable

$$\beta^i(x, t) = \mathcal{S}_i \tilde{\beta}^i(x, 1 - t), \quad H^i(x, t) = \mathcal{S}_i \tilde{H}^i(x, 1 - t). \tag{92}$$

It is a simple matter to see, using (90) and $\mathcal{S}_i = 1$ for any $i \in \mathcal{C}$, that (88c), (88d) are invariant under this change, and that the initial and final conditions (88a) and (88b) are again exchanged. Thus we recover the original equations (75a)-(75c) and (83), but in terms of $\tilde{\beta}^i(x, t)$, $\tilde{H}^i(x, t)$ and $\tilde{\lambda}^i$. Assuming that the solution is unique, and re-introducing the explicit $\lambda$-dependence for clarity, we conclude that

$$\tilde{\underline{\beta}}^{(\lambda)}(x, t) = \underline{\beta}^{(\tilde{\lambda})}(x, t), \tag{93}$$

or equivalently $\tilde{\underline{\beta}}^{(\tilde{\lambda})}(x, t) = \underline{\beta}^{(\lambda)}(x, t)$. Finally, in order to obtain the symmetry of the SCGF, we use the expression (78). We have for every $i \in \mathcal{C}$, using $q^{(\lambda)} = \mathsf{q}[\underline{\beta}^{(\lambda)}]$ and the relation $\mathsf{j}_i = -\tilde{\mathsf{j}}_i$, that

$$
\begin{aligned}
\frac{\partial F(\underline{\lambda})}{\partial \lambda^i} &= \int_0^1 \mathrm{d}t\, \mathsf{j}_i[q^{(\lambda)}(0, t)] = \int_0^1 \mathrm{d}t\, \mathsf{j}_i[\tilde{q}^{(\tilde{\lambda})}(0, t)] = -\int_0^1 \mathrm{d}t\, \mathsf{j}_i[q^{(\tilde{\lambda})}(0, 1 - t)] \\
&= -\int_0^1 \mathrm{d}t\, \mathsf{j}_i[q^{(\tilde{\lambda})}(0, t)] = \frac{\partial F(\underline{\beta}_L - \underline{\beta}_R - \underline{\lambda})}{\partial \lambda^i}.
\end{aligned}
\tag{94}
$$

Integrating in $\lambda^i$ from the mid-point $\lambda^i = (\beta^i_L - \beta^i_R)/2$, we obtain Eq. (19).

Let us finally comment on the potential pitfalls if weak solutions are involved. The first is the shift made in (87). This step works only because we use (75c) to keep the left-hand side of (75d) invariant. However, if $\beta^i(x, t)$ and $H^i(x, t)$ admit weak solutions of *different entropy type*, for instance with positive, respectively negative, physical entropy production, then this does not work, and the derivation fails. Another is the time-reversal change of variable itself, Eq. (92), which, on weak solutions, would simply reverse the entropy production type. As mentioned, in integrable systems these pitfalls are avoided.

## 4.4 BMFT of dynamical correlation functions

While in previous studies the MFT has been mainly applied to study density and current fluctuations, it also provides us a powerful way of evaluating dynamical correlation functions in both homogeneous and inhomogeneous states. In diffusive systems, correlation functions in a NESS have been studied using the conventional MFT in [7, 77]. To our knowledge, however, we provide here the first instance where the MFT is applied to compute *dynamical* correlation functions on the Euler scale in arbitrary slowly modulating initial states.

In order to illustrate how it works, let us evaluate $S_{\hat{q}_{i_1}, \hat{q}_{i_2}}(x_1, t_1; x_2, t_2)$, which is the Euler scaling limit (22) for the charge densities $\hat{q}_{i_1}$, $\hat{q}_{i_2}$. We repeat the arguments made in Subsec 4.1 for the SCGF. The BMFT gives the saddle-point result

$$S_{\hat{q}_{i_1}, \hat{q}_{i_2}}(x_1, t_1; x_2, t_2) = -\frac{\mathrm{d}^2}{\mathrm{d}\lambda_1 \mathrm{d}\lambda_2} \mathcal{F}_{\mathrm{corr}}[\underline{q}^*]\Big|_{\lambda_1 = \lambda_2 = 0}, \tag{95}$$

where $\underline{q}^*$ minimises the BMFT action $S_{\mathrm{corr}}[\underline{q}, \underline{H}]$ associated to the dynamical correlation function,

$$S_{\mathrm{corr}}[\underline{q}, \underline{H}] := \mathcal{F}_{\mathrm{corr}}[q] + \int_{\mathbb{S}} \mathrm{d}x \mathrm{d}t \, H^i (\partial_t q_i + \partial_x \mathsf{j}_i[\underline{q}]), \tag{96}$$

with $\mathcal{F}_{\mathrm{corr}}[q(\cdot, \cdot)] = \mathcal{F}[\underline{q}(\cdot, 0)] - (\lambda_1 q_{i_1}(x_1, t_1) + \lambda_2 q_{i_2}(x_2, t_2))$. Here the final time $T$ in $\mathbb{S} = \mathbb{R} \times [0, T]$ could take an arbitrary value so long as $T > t_1, t_2$; we will see that the result is indeed independent of $T$. Again, we drop the star symbol $^*$ for lightness of notation. By the saddle-point equation, the total $\lambda_2$ derivative equals the partial derivative, and since $\partial_{\lambda_2} \mathcal{F}_{\mathrm{corr}} = -\mathsf{q}_{i_2}(x_2, t_2) = \mathsf{q}_{i_2}[\underline{\beta}(x_2, t_2)]$, we have

$$S_{\hat{q}_{i_1}, \hat{q}_{i_2}}(x_1, t_1; x_2, t_2) = \frac{\mathrm{d}}{\mathrm{d}\lambda} \mathsf{q}_{i_2}(x_2, t_2)\Big|_{\lambda = 0}, \tag{97}$$

where we redefined $\lambda := \lambda_1$, with the associated BMFT equations

$$H^i(x, 0) = \beta^i_{\mathrm{ini}}(x) - \beta^i(x, 0), \tag{98a}$$

$$H^i(x, T) = 0, \tag{98b}$$

$$\partial_t \beta^i + \mathsf{A}_j^{\,i}[\underline{\beta}] \partial_x \beta^j = 0, \tag{98c}$$

$$\partial_t H^i + \mathsf{A}_j^{\,i}[\underline{\beta}] \partial_x H^j = -\lambda \delta_{i_1}^{\,i} \delta(x - x_1) \delta(t - t_1). \tag{98d}$$

Here, the boundary conditions are $\beta^i(x \to \pm\infty, t) = \beta^i_{\mathrm{ini}}(x \to \pm\infty)$ and $H^i(x \to \pm\infty, t) = 0$ for any value of $t$.

Similarly, higher-point functions are accessed by multiple derivatives. In fact, one may repeat the argument for arbitrary observables, using

$$\mathcal{F}_{\mathrm{corr}}[\underline{q}] = \mathcal{F}[\underline{q}(\cdot, 0)] - \sum_{a=1}^{n} \lambda_a \mathsf{o}_a[\underline{q}(x_a, t_a)], \tag{99}$$

as per the theory of Subsec. 3.1 (see Eqs. (55)-(57)). This gives

$$S_{\hat{o}_1, \dots, \hat{o}_n}(x_1, t_1; \cdots; x_n, t_n) = \frac{\mathrm{d}^{n-1} \mathsf{o}_n[\underline{\mathsf{q}}(x_n, t_n)]}{\mathrm{d}\lambda_1 \cdots \mathrm{d}\lambda_{n-1}}\Big|_{\lambda_1, \dots, \lambda_{n-1} = 0}, \tag{100}$$

with the BMFT equations

$$H^i(x,0) = \beta^i_{\text{ini}}(x) - \beta^i(x,0), \tag{101a}$$

$$H^i(x,T) = 0, \tag{101b}$$

$$\partial_t \beta^i + \mathsf{A}_j^{\;i}[\underline{\beta}]\partial_x \beta^j = 0, \tag{101c}$$

$$\partial_t H^i + \mathsf{A}_j^{\;i}[\underline{\beta}]\partial_x H^j = -\sum_{a=1}^{n-1} \lambda_a \frac{\partial \mathsf{o}_a}{\partial \mathsf{q}_i} \delta(x-x_a)\delta(t-t_a). \tag{101d}$$

## 4.5 (Non)linear hydrodynamic projections and (non)linear response

We first note that two applications of Eq. (100) with $n = 2$ give

$$S_{\hat{o}_1,\hat{o}_2}(x_1,t_1;x_2,t_2) = \left.\frac{\partial \mathsf{o}_2}{\partial \mathsf{q}_i}\right|_{\underline{\mathsf{q}}(x_2,t_2)} S_{\hat{o}_1,\hat{q}_i}(x_1,t_1;x_2,t_2), \tag{102}$$

where $\underline{\mathsf{q}}(x_2,t_2)$ is the Euler hydrodynamic solution; by recursion this implies the hydrodynamic projection principle of Eq. (26). Further, as the BMFT equation (98c) implies the Euler equation (11) relating $\mathsf{q}_{i_2}(x_2,t_2)$ with $\mathsf{j}_{i_2}(x_2,t_2)$ in (97), we may apply the $t_2$ derivative, use this Euler equation, take the $x_2$ derivative out, and apply the $\lambda$ derivative on $\mathsf{j}_{i_2}(x_2,t_2)$ using (102) with $\hat{o}_1 = \hat{q}_{i_1}$ and $\hat{o}_2 = \hat{j}_{i_2}$. The result is the Euler equation (28) for Euler-scale two-point functions of conserved densities, which is expanded around the inhomogeneous background. Thus, these two pillars of the study of ballistic-scale correlation functions follow simply from the BMFT.

By multiple applications of the $\lambda_a$-derivatives, the BMFT gives rise to a nonlinear hydrodynamic projection principle, for higher-point functions. For instance, for three-point functions, dropping the explicit space-time dependence,

$$S_{\hat{o}_1,\hat{o}_2,\hat{o}_3} = \frac{\mathrm{d}}{\mathrm{d}\lambda_1}\frac{\partial \mathsf{o}_3}{\partial \mathsf{q}_i}\frac{\mathrm{d}\mathsf{q}_i}{\mathrm{d}\lambda_2} = \frac{\partial \mathsf{o}_3}{\partial \mathsf{q}_i}S_{\hat{o}_1,\hat{o}_2,\hat{q}_i} + \frac{\partial^2 \mathsf{o}_3}{\partial \mathsf{q}_i \partial \mathsf{q}_j}S_{\hat{o}_1,\hat{q}_j}S_{\hat{o}_2,\hat{q}_i}, \tag{103}$$

which by recursive applications give rise to (29),

$$S_{\hat{o}_1,\hat{o}_2,\hat{o}_3} = \frac{\partial \mathsf{o}_1}{\partial \mathsf{q}_i}\frac{\partial \mathsf{o}_2}{\partial \mathsf{q}_j}\frac{\partial \mathsf{o}_3}{\partial \mathsf{q}_k}S_{\hat{q}_i,\hat{q}_j,\hat{q}_k} + \frac{\partial^2 \mathsf{o}_1}{\partial \mathsf{q}_i \partial \mathsf{q}_j}\frac{\partial \mathsf{o}_2}{\partial \mathsf{q}_k}\frac{\partial \mathsf{o}_3}{\partial \mathsf{q}_l}S_{\hat{q}_i,\hat{q}_k}S_{\hat{q}_j,\hat{q}_l} + \text{cyclic perm. of } 1,2,3. \tag{104}$$

Thus, for the evaluation of general $n$-point correlation functions in the Euler scaling limit, it stays true that it is sufficient to know the dynamical correlation functions of conserved densities, along with the stationary, homogeneous averages of the local fields involved. We note that formula (29) gives an immediate explanation for certain 3-point function projection formulae obtained earlier by linear response from the Euler equations [84, Eqs 187, 189]. Note that the BMFT techniques described here recast the problem of nonlinear hydrodynamic projections into simple applications of differential operators, and higher-point formulae are straightforward to work out.

The transport equation for higher-point functions can be deduced from the nonlinear hydrodynamic projection principle, as usual by using the conservation laws. For the three-point function, for instance, from Eq. (29) one has

$$\partial_{t_1} S_{\hat{q}_i,\hat{q}_j,\hat{q}_k} + \partial_{x_1}\left(\mathsf{A}_i^{\;l}S_{\hat{q}_l,\hat{q}_j,\hat{q}_k} + \mathsf{H}_i^{\;lr}S_{\hat{q}_l,\hat{q}_j}S_{\hat{q}_r,\hat{q}_k}\right) = 0, \tag{105}$$

where $H_i^{lr} = \partial^2 j_i / \partial q_l \partial q_r$. This is in agreement with the results of nonlinear response arguments from the Euler equations [22].

As a remark on the overall consistency of our theory, we note that the BMFT (non)linear hydrodynamic projection formulae are in fact entirely a consequence of the Euler scaling $\ell^{1-n}$, implied by Eq. (21), of $n$-point connected correlation functions at space-time points scaled with $\ell$, along with the principle of local relaxation of fluctuations. Indeed, one expresses the observable $o[q]$ as a series in powers of $q_i$'s, rewrites this in terms of connected correlation functions, and uses the Euler scaling. The terms that remain nonzero in the Euler scaling limit give the projection formula. For illustration, consider the scaled two-point function $\ell \langle \overline{o_1}\, \overline{q_2} \rangle_\ell^c$ (where the scaled space-time positions are kept implicit), with an observable of the (purely theoretical) form $o_1[q] = q_1^2$. We evaluate, as $\ell \to \infty$

$$
\begin{aligned}
S_{\hat{o}_1, \hat{q}_2} &\sim \ell \langle \overline{o_1}\, \overline{q_2} \rangle_\ell^c \\
&\sim \ell \langle\!\langle q_1^2 q_2 \rangle\!\rangle_\ell - \ell \langle\!\langle q_1^2 \rangle\!\rangle_\ell \langle\!\langle q_2 \rangle\!\rangle_\ell \\
&= \ell \langle\!\langle q_1 q_1 q_2 \rangle\!\rangle_\ell^c + 2\ell \langle\!\langle q_1 \rangle\!\rangle_\ell \langle\!\langle q_1 q_2 \rangle\!\rangle_\ell^c \\
&\sim 2q_1 S_{\hat{q}_1, \hat{q}_2},
\end{aligned}
\tag{106}
$$

where in the last step we used the fact that, by Euler scaling, $\langle\!\langle q_1 q_1 q_2 \rangle\!\rangle_\ell^c = O(\ell^{-2})$ and $\langle\!\langle q_1 q_2 \rangle\!\rangle_\ell^c = O(\ell^{-1})$ as $\ell \to \infty$, and thus only the second term remains finite. This gives (26) for this particular observable.

## 4.6  Long-range correlations

As discussed in Sec. 2, the BMFT generically predicts that the equal-time correlator $S_{\hat{q}_{i_1}, \hat{q}_{i_2}}(x_1, t; x_2, t)$ possesses *long-range correlations*, provided that three elements are present: interactions, initial inohomogeneity, and multiple conservation laws. By the BMFT, it is possible to see this from rather general arguments.

Consider again the BMFT equations (98). Let us denote formally the result of the linear evolution by the space-time dependent propagator $A_j^i$,

$$
\partial_t \gamma^i + A_j^i[\underline{\beta}]\partial_x \gamma^j = 0,
\tag{107}
$$

from time $t = t_1$ to time $t = t_2$, by using the operator $U_\lambda(t_2, t_1)$:

$$
\underline{\gamma}(t_2) = U_\lambda(t_2, t_1)\underline{\gamma}(t_1),
\tag{108}
$$

where here and below, for lightness of notation, when not writing the spatial argument, the result is seen as a function of space. We make explicit the $\lambda$-dependence of the operator; recall that the propagator $A_j^i[\underline{\beta}]$ depends in general on $\beta^i$'s (unless the hydrodynamic system is non-interacting), which are the $\lambda$-dependent solution to Eqs. (98).

Clearly,

$$
\underline{\beta}(t) = U_\lambda(t, t')\underline{\beta}(t'),
\tag{109}
$$

Upon integrating both sides of the equation (98d) for $H$ over $[t_1 - \varepsilon, t_1 + \varepsilon]$ ($\varepsilon > 0$ infinitesimal), we have $H^i(x, t_1 + \varepsilon) - H^i(x, t_1 - \varepsilon) = -\lambda \delta_{i_1}^i \delta(x - x_1)$. Since, by the boundary condition (98b), $0 = \underline{H}(T) = U_\lambda(T, t_1 + \varepsilon)\underline{H}(t_1 + \varepsilon)$, invertibility of the evolution operator implies $H^i(x, t_1 + \varepsilon) = 0$, which yields $H^i(x, t_1 - \varepsilon) = \lambda \delta_{i_1}^i \delta(x - x_1)$. Writing $\underline{H}(t_1 - \varepsilon) = U_\lambda(t_1, 0)\underline{H}(0) =$

$U_\lambda(t_1, 0)\underline{\beta}_{\text{ini}} - U_\lambda(t_1, 0)\underline{\beta}(0)$ from the other boundary condition (98a), and using (109), we then obtain

$$\beta^i(x, t_1) = \left(U_\lambda(t_1, 0)\underline{\beta}_{\text{ini}}\right)^i(x) - \lambda\delta_{i_1}{}^i\delta(x - x_1). \tag{110}$$

Eq. (110) is a crucial result concerning the structure of BMFT. Note that if the evolution operator $U_\lambda(t_1, 0)$ was simply the Euler hydrodynamic (nonlinear) evolution, independent of $\lambda$, then $U_\lambda(t_1, 0)\underline{\beta}_{\text{ini}}$ would give $\beta^i(x, t_1)$ evaluated at $\lambda = 0$. In this case, (110) would simply state that the insertion of the observable at $t = t_1$ can be implemented by evolving the fluid from the initial state up to time $t_1$, and then by perturbing the fluid state and measuring its linear response. However, $U_\lambda(t_1, 0)$, representing time evolution for times $t < t_1$, is not the fluid evolution, and depends on $\lambda$ itself, even though the inserted observable is at time $t_1$. In (110), the $\lambda$ dependence comes both from the explicit $\delta$-function, and from the evolution operator itself. The former represents the linear response contribution, while the latter, we interpret as coming from nonlinear correlated wave production and scattering occurring in the times before $t_1$; an interpretation that is made clearer in our studies of integrable systems.

In Appendix C, we review hydrodynamic response theory. In particular, we show on the basis of the result in Eq. (110) that hydrodynamic response theory for two-point function is correct only if the model is non-interacting, the earlier time in the correlator is zero $t_1 = 0$, or the state is homogeneous. For higher-point function, it is correct only if the model is non-interacting, or all but one of the times are zero.

From (110) one evaluates the correlation function (97) at $t_2 = t_1$ as follows

$$
\begin{aligned}
S_{\hat{q}_{i_1}, \hat{q}_{i_2}}(x_1, t_1; x_2, t_1) &= \left.\frac{\mathrm{d}}{\mathrm{d}\lambda}\mathsf{q}_{i_2}[\underline{\beta}(x_2, t_1)]\right|_{\lambda=0} \\
&= \mathsf{C}_{i_1 i_2}(x_1, t_1)\delta(x_2 - x_1) - \partial_\lambda\left.\left(U_\lambda(t_1, 0)\underline{\beta}_{\text{ini}}\right)^i(x_2)\right|_{\lambda=0}\mathsf{C}_{i\, i_2}(x_2, t_1).
\end{aligned}
\tag{111}
$$

The first term in the resulting expression is the linear response (see Appendix C), which at equal times is simply the thermodynamic response of the fluid cell and thus supported at equal positions. The second term, on the other hand, clearly demonstrates the potential presence of long-range correlations – recall that the derivative with respect to $\lambda$ brings a dependence on $x_1$, and in general the result is nonzero for $x_1 \neq x_2$. It does not appear to be possible in general to obtain a more explicit expression of the resulting correlations than (111), however we will see that in integrable systems, explicit integral equations are found, which give indeed nonzero results for $x_1 \neq x_2$.

As we mentioned, in certain situations no long-range correlations are expected even at $t_1 > 0$ (for $t_1 = 0$ no long range correlation appear as the initial state is by assumption not correlated; in this case $U_\lambda(0, 0) = 1$). Two of the conditions can be seen immediately from the above result. Indeed, the second term in (111) vanishes either if the system is noninteracting (in which case $U_\lambda(t_1, 0)$ is $\lambda$-independent) or the initial condition is homogeneous (in which case $U_\lambda(t_1, 0)\underline{\beta}_{\text{ini}} = \underline{\beta}_{\text{ini}}$).

For the third condition, that no long range correlation appear if there is only one conservation law, a different general argument is required. For this purpose, consider the evolution equation (34) for equal-time Euler-scale correlations, which follows from hydrodynamic projections (shown in the previous subsection from BMFT). In the single component case, it reads

$$\partial_t S_{\hat{q}, \hat{q}}(x, t; x', t) + \partial_x[\mathsf{A}(x, t)S_{\hat{q}, \hat{q}}(x, t; x', t)] + \partial_{x'}[\mathsf{A}(x', t)S_{\hat{q}, \hat{q}}(x, t; x', t)] = 0. \tag{112}$$

It is a simple matter to show that the form $S_{\hat{q},\hat{q}}(x,t;x',t) = \overline{\mathsf{C}}(x,t)\delta(x-x')$ is invariant under time evolution. As the initial condition satisfies it, with $\overline{\mathsf{C}}(x,0) = \mathsf{C}(x)$, then indeed the solution preserved the delta-function structure, and no long-range correlation appear. In particular, one finds for $\overline{\mathsf{C}}(x,t)$

$$\partial_t \overline{\mathsf{C}}(x,t) + \partial_x[\mathsf{A}(x,t)\overline{\mathsf{C}}(x,t)] = 0. \tag{113}$$

The argument using the evolution equation makes it less evident how no long-range correlation may appear if the flux Jacobian is independent of the state, as must be true from the BMFT result (111). Here we simply note that, in this case, by diagonalising the resulting evolution equation, and using conditions on the covariance matrix of normal modes [15] one recovers the lack of long-range correlations.

It should be emphasised that various types of long-range correlations have been observed in both driven-diffusive NESS [87, 88], boundary-driven NESS in ballistic quantum spin chains [89] and in ballistic NESS of integrable systems [60]. In diffusive systems, a NESS, which is usually induced by an external field and hence has nonzero gradients, presents $1/x$ spatial correlations due to Fourier-space discontinuities of diffusive transport coefficients. Indeed, a discontinuity in the Fourier space generically points to $1/x$ decay because of the representation $\Theta(k) \sim \int_{\mathbb{R}} \mathrm{d}x\, e^{\mathrm{i}kx}/x$ of the step function. The existence of such long-range correlations was also microscopically established for the SSEP, which was attributed to non-locality of the density large deviation function in the NESS [90]. In integrable systems, a ballistic NESS also develops at long times from unbalanced initial conditions, e.g., in the partitioning protocol; it itself homogeneous (it has no gradient) with constant fluxes, as permitted by ballistic transport. For instance, in the case of the free massive scalar field theory, it was shown that correlators show long-range correlations with varying exponents ($1/x$ decay for certain correlations involving the "fundamental fields", and $1/x^2$ for correlations involving conserved densities) [59]. In the same vein, it was also demonstrated that density-density correlations in the Lieb-Liniger model in a NESS have long range, $1/x^2$, due to discontinuities in the quasi-particle distributions [60]. Importantly, these long-range correlations are purely due to discontinuities in the NESS density matrix, in its representation in terms of asymptotic particles, and do not necessitate interactions. In the case of the ballistic boundary-driven XX quantum spin chain [89], long-range correlations in the NESS, similarly, do not need interactions and they can only be detected in the atypical-biased dynamics describing quantum trajectories supporting large currents (being absent, instead, in the typical-unbiased dynamics).

Long-range correlations in all these cases are in sharp contrast with what the BMFT captures: long-range correlations that take *generic shape in space*, but that decay as $1/\ell$ and are supported on regions that grow with $\ell$ as time grows like $\ell$ (and the initial inhomogeneity is of length scale $\ell$). These are controlled entirely by the Euler-scale hydrodynamics, and do not necessitate singularities in the Fourier or asymptotic-particle space representation of any transport coefficient. They represent nontrivial correlations between separate fluid cells, because integrals of the resulting correlation functions over macroscopic regions that are at macroscopic separations have finite values. This is in contrast, in particular, to the $1/x^2$ contributions in ballistic NESS, that pertain to single fluid cells.

## 4.7 Fluctuations inside fluid cells

We now argue that the BMFT gives information not only about the Euler-scale correlations and fluctuations, which occur at macroscopic scales, but also about fluctuations *within the fluid cells*. More precisely, we argue that the fluid cells' thermodynamics can be accessed – that is, the susceptibilities and in general all the cumulants of total charges in the cell, scaled by the size of the cell. As the free energy generates such cumulants, this amounts to the specific free energy of the fluid cell, i.e., its total free energy divided by its size.

Most importantly we also argue that, in general in interacting, non-integrable models, *the specific free energy of the fluid cell at macroscopic coordinates $x, t$, is not that of the state described by the solution to the Euler equation $\beta^i(x, t)$*. But in integrable systems, the fluid cell free energy is indeed that of this GGE. The physics is similar to that behind long-range correlation, but instead of being that of interactions between different fluid mode, the nontrivial fluctuations within fluid cells is due to modes *self-interaction*. Here self-interaction refers to the nontrivial dependence of normal modes on their own propagation (effective) velocities. Thus, we conjecture that this effect is absent whenever the fluid is linearly degenerate (such as in integrable systems).

The question is about the large-deviation theory for the mesoscopically extensive conserved quantities $\overline{Q}_i$ within any given fluid cell. At the macroscopic coordinates $x, t$, these may be taken as

$$\overline{Q}_i = L\overline{q}_i(\ell x, \ell t). \tag{114}$$

In order to illustrate the ideas, we concentrate on the second cumulants, but similar calculations can be done for higher cumulants.

In analogy with usual thermodynamics, by extensivity of the charges the cumulants are expected to scale like $L$, which is trivial for the first cumulant, $\langle \overline{Q}_i \rangle_\ell \sim L\mathsf{q}_i(x, t)$. The picture is that within the fluid cell $\{(\ell x + y, \ell t) : y \in [-L/2, L/2]\}$, correlation functions $\langle \hat{q}_i(\ell x + y, \ell t)\hat{q}_j(\ell x, \ell t)\rangle^{\mathrm{c}}_\ell$ decay quickly as $|y|$ grows, up to values of order $\ell^{-1}$, where the long-range correlations start. Note that it has to be of this order so that it is smoothly connected to long-range correlations outside the fluid cell, which would amount to the contribution to $\langle \overline{Q}_{i_1}\overline{Q}_{i_2} \rangle^{\mathrm{c}}_\ell$ with magnitude $O(L^2/\ell)$. Thus, for instance for the second cumulants, one expects $\langle \overline{Q}_{i_1}\overline{Q}_{i_2} \rangle^{\mathrm{c}}_\ell \sim L(\overline{\mathsf{C}}_{i_1,i_2} + O(L/\ell))$ for some $\overline{\mathsf{C}}_{i_1,i_2}$ to be determined. Here $O(L/\ell)$ is a sub-leading term with respect to the first one because usually $L$ is taken as $L = \ell^\alpha$ for some $0 < \alpha < 1$ by the mesoscopic scaling, hence $L/\ell = L^{1-1/\alpha} < 1$. We note that under the picture of fast decay within the fluid cell, one may also take

$$\overline{Q}_i := \int_{-\epsilon\ell/2}^{\epsilon\ell/2} \mathrm{d}y \, \overline{q}(\ell x + y, \ell t). \tag{115}$$

for $\epsilon > 0$ small, which is useful for the calculation below.

It turns out that the covariance matrix $\langle \overline{Q}_{i_1}\overline{Q}_{i_2} \rangle^{\mathrm{c}}_\ell$ is determined by the coefficient $\overline{C}_{i_1,i_2}(x, t)$ of the delta-function contribution,

$$S_{\hat{q}_{i_1}, \hat{q}_{i_2}}(x, t; x'; t) = \overline{\mathsf{C}}_{i_1,i_2}(x, t)\delta(x - x') + \text{regular}. \tag{116}$$

Indeed,

$$
\begin{aligned}
\langle \overline{Q}_{i_1} \overline{Q}_{i_2} \rangle_\ell^{\mathrm{c}} &= \int_{-\epsilon\ell/2}^{\epsilon\ell/2} \mathrm{d}y \, \langle \overline{q}_{i_1}(\ell x + y, t) \, L\overline{q}_{i_2}(\ell x, t) \rangle_\ell^{\mathrm{c}} \\
&= L\ell \int_{-\epsilon/2}^{\epsilon/2} \mathrm{d}y \, \langle \overline{q}_{i_1}(\ell(x + y), t)\overline{q}_{i_2}(\ell x, t) \rangle_\ell^{\mathrm{c}} \\
&= L \int_{-\epsilon/2}^{\epsilon/2} \mathrm{d}y \, S_{\hat{q}_{i_1}, \hat{q}_{i_2}}(x + y, t; x, t) \\
&= L\overline{C}_{i_1, i_2}(x, t) \quad \text{(for } \epsilon \text{ small).}
\end{aligned}
\tag{117}
$$

The question is therefore about the coefficient of this delta-function.

In the BMFT result (111), the first term gives for coefficient the C-matrix of the GGE represented by $\beta^i(x, t)$; this term gives the GGE covariances. But what about the second term?

In integrable models, the BMFT predicts that the covariace matrix is correctly described by the GGE $e^{-\sum_i \beta^i(x,t)\overline{Q}_i}$ associated to the Lagrange multipliers $\beta^i(x, t)$ at that point. In particular, the second term in (111) does not have delta-function contributions. Thus, extending this to all cumulants, although the Euler-hydrodynamics time evolution of (12) does not correctly describe Euler-scale correlations at macroscopic times, in integrable models, it still correctly describes all fluctuations of extensive quantities within fluid cells.

By contrast, a simple analysis for non-integrable models with a single fluid mode suggests that, in such cases, the fluctuations within fluid cells are *not* given by the Gibbs states $e^{-\sum_i \beta^i(x,t)\overline{Q}_i}$. That is, although all averages of local observables agree with this state, mesoscopically extensive quantities fluctuate *according to a different distribution*. In this case, the covariance matrix satisfies (113), and thus it evolves nontrivially.

Of course, local averages do not probe the full distribution: in the limit of large fluid cells, from the viewpoint of local averages, the distribution concentrates on the micro-canonical shell specified by the local conserved densities $\mathsf{q}_i(x, t)$. The scaled cumulants of mesoscopic quantities $\overline{Q}_i$ probe more subtle, rare fluctuations (in the sense of large deviation theory). For instance, it is well known that the macrocanonical and microcanonical ensembles are equivalent for local averages, but give different scaled cumulants of extensive observables (extensive observables are, in fact, non-fluctuating in the microcanonical ensemble). In the BMFT, there is no reason for the fluid cell to be distributed according to $e^{-\sum_i \beta^i(x,t)\overline{Q}_i}$; what we find is that inhomogeneous long-wavelength initial states generate in general, over time, different, non-canonical distributions within fluid cells.

We elucidate this aspect in Appendix E by considering the TASEP as a paradigmatic example.

## 5  BMFT for integrable systems

There are many good reasons to study the out-of-equilibrium physics of integrable systems, as it is well established that integrability qualitatively affects thermalisation and hydrodynamic behaviours (see, e.g., the special issues [65,91]). However, here the main purpose is to provide explicit examples of applications of the BMFT, and demonstrate that the machinery of GHD

allows us to obtain the exact expressions of the two main objects we have been focusing on, the SCGF $F(\lambda, T)$ and the Euler-scale dynamical correlation function $S_{\hat{q}_{i_1}, \hat{q}_{i_2}}(x_1, t_1; x_2, t_2)$.

## 5.1 The BMFT formulation using GHD

Integrable many-body systems possess a large number, that grows with the system's size, of extensive conserved charges [92]. Probably the most important consequence of this is that many-body scattering processes factorise into two-body processes and preserve all momenta. This in turn amounts to the existence of stable excitations called quasi-particles: a quasi-particle is identified with an asymptotic particle of the model (or an "asymptotic object", be it a particle, bound state, soliton, radiation mode, etc.), and by elastic and factorised scattering, its trajectory can be "traced" within space-time throughout the full scattering process, from the in-state to the out-state. This is true at least at the level of precision required for Euler hydrodynamics. Below we will refer to quasi-particles simply as "particles", and we will parametrise their associated asymptotic momenta as $p_\theta$ in terms of a "rapidity" $\theta$. In general, the rapidity $\theta$ may be a multiple index, encoding both the asymptotic momentum and the type of asymptotic object, if the model admits many types; for simplicity we will assume it is a single continuous index take values in $\mathbb{R}$, as is the case for the Lieb-Liniger and hard-rod models, where one may use simply $p_\theta = \theta$.

The density of particles per unit rapidity $\hat{Q}_\theta = \int \mathrm{d}x\, \hat{\rho}_\theta(x)$ (i.e., $\hat{\rho}_\theta(x)\mathrm{d}\theta\mathrm{d}x$ counts the number of particles with rapidity within $[\theta, \theta + \mathrm{d}\theta)$ and position within $[x, x + \mathrm{d}x)$) is an extensive conserved charge, and these together form a complete basis of extensive charges[7], see the review [25]. Thus $\theta$ parametrises the hydrodynamic modes. The standard formulation of GHD is as a hydrodynamic theory in terms of such modes. Thus it is a hydrodynamic equation for the average densities $\rho_\theta(x) := \mathsf{q}_\theta(x)$. The corresponding average currents take the form $\mathsf{j}_\theta = v_\theta^{\mathrm{eff}} \rho_\theta$ (see the reviews [93,94]) where the effective velocity satisfies

$$p'_\theta v_\theta^{\mathrm{eff}} = E'_\theta + 2\pi \int \mathrm{d}\phi\, T^\phi_\theta \rho_\phi (v_\phi^{\mathrm{eff}} - v_\theta^{\mathrm{eff}}); \tag{118}$$

here $E_\theta$ is the asymptotic energy of the particle $\theta$, and primes are derivatives, e.g., $p'_\theta = \mathrm{d}p_\theta/\mathrm{d}\theta$. The effective velocity depends on the specifics of the interactions in the model via the differential scattering phase $T^\theta_\phi$, which is, in quantum systems, related to the two-body scattering phase $S_{\theta\phi}$ as $T^\theta_\phi = -\mathrm{i}\, \mathrm{d}\log S_{\theta\phi}/(2\pi \mathrm{d}\theta)$. Here for simplicity we assume that $T^\theta_\phi = T^\phi_\theta$ is symmetric, which is the case for many integrable systems (seeing $T$ as a matrix, the column index is the leftmost, superscript, and the row index is the rightmost, subsript index). The GHD equation is then

$$\partial_t \rho_\theta + \partial_x (v_\theta^{\mathrm{eff}} \rho_\theta) = 0. \tag{119}$$

See the lecture notes [3] for details.

We will use Greek indices for labelling rapidities, and Roman indices, as done in previous sections, for labelling a generic basis of conserved charges. Lagrange multipliers associated to the charges $\hat{Q}_\theta$ will be denoted $\beta^\theta$. These are related to the Lagrange multipliers $\beta^i$, for a generic basis of conserved charges, as

$$\beta^\theta = \beta^i h^\theta_i, \tag{120}$$

---

[7]More precisely, $\hat{Q}_\theta$ is not quite extensive, but together for $\theta \in \mathbb{R}$ they form a "scattering basis" for the space of extensive conserved charges, out of which any charge can be written as a $\theta$ integral.

where $h_i^\theta$ is the one-particle eigenvalue of $\hat{Q}_i$ in quantum systems, or the quantity of that charge carried by the asymptotic object $\theta$ in classical systems. Charge densities and current averages are given by $q_i = \int_\mathbb{R} d\theta\, h_i^\theta \rho_\theta$ and $j_i = \int_\mathbb{R} d\theta\, h_i^\theta j_\theta$.

In applying the BMFT to GHD, it is convenient to re-write the path-integral (49) in terms of $\rho_\theta$. We now explain how this work; this will also allow us to introduce some of the main objects of the thermodynamics of integrable many body systems.

The BMFT expectation values are given by

$$\langle \bullet \rangle = \int_{(\mathbb{S}\times\mathbb{R})} d\mu[\rho(\cdot,\cdot)] e^{-\ell\mathcal{F}[\rho(\cdot,0)]} \delta(\partial_t\rho + \partial_x j[\rho]) \bullet, \tag{121}$$

where the path-integral is performed over all the possible configurations of $(x,t,\theta) \mapsto \rho_\theta(x,t)$, with $(x,t,\theta) \in \mathbb{S}\times\mathbb{R} = \mathbb{R}\times[0,T]\times\mathbb{R}$. Again, the delta function is best understood via its integral representation

$$\delta(\partial_t\rho + \partial_x j[\rho]) = \int_{(\mathbb{S}\times\mathbb{R})} d\mu[H(\cdot,\cdot)] \exp\left[-\int_{\mathbb{S}\times\mathbb{R}} dt dx d\theta\, H^\theta(\partial_t\rho_\theta + \partial_x(v_\theta^{\text{eff}}\rho_\theta))\right]. \tag{122}$$

The probability distribution for the initial fluctuation in Eq. (43) reads

$$\mathcal{F}[\rho(\cdot,0)] = \int_\mathbb{R} dx \left( \int_\mathbb{R} d\theta\, \beta_{\text{ini}}^\theta(x)\rho_\theta(x,0) - f[\beta_{\text{ini}}(x)] - s[\rho(x,0)] \right), \tag{123}$$

where the free energy density $f[\beta]$ and the (Yang-Yang) entropy density $s[\rho]$ [95,96] are given by $f[\beta] = \int_\mathbb{R} d\theta\, p_\theta' F^\theta/(2\pi)$ and $s[\rho] = \int_\mathbb{R} d\theta\, \rho_\theta^{\text{tot}}(n_\theta\epsilon^\theta - F^\theta)$, respectively, with $F^\theta = F(\epsilon^\theta)$ the free energy function [3]. The latter encodes the statistics of the quasi-particles; e.g., for fermions it is given by $F(\epsilon) = -\log(1 + e^{-\epsilon})$. The quantities $n_\theta$ and $\epsilon^\theta$ are the occupation function and pseudo-energy, respectively, and are related to each other by $n_\theta = dF(\epsilon)/d\epsilon|_{\epsilon=\epsilon^\theta}$. The pseudo-energy is in turn related to the Lagrange multipliers by the non-linear integral equation

$$\epsilon^\theta = \beta^\theta + \int_\mathbb{R} d\phi\, T^\theta_{\ \phi} F^\phi. \tag{124}$$

The total density of states $\rho_\theta^{\text{tot}}$ is

$$\rho_\theta^{\text{tot}} = \frac{p_\theta'}{2\pi} + \int_\mathbb{R} d\phi\, T^\phi_{\ \theta}\rho_\phi, \tag{125}$$

and one can check that the above definitions imply the relation

$$n_\theta = \frac{\rho_\theta}{\rho_\theta^{\text{tot}}}. \tag{126}$$

Finally, let us comment on how the Lieb-Liniger model and the hard rods are characterised by the quantities we introduced above. First, the differential scattering phase $T^\theta_{\ \phi}$ is given, respectively, as follows:

$$(T_{\text{LL}})^\theta_{\ \phi} = \frac{2c}{(\theta-\phi)^2 + c^2}, \quad (T_{\text{HR}})^\theta_{\ \phi} = -a, \tag{127}$$

where $c$ is the coupling constant of the Lieb-Liniger model (5). Second, another quantity that distinguishes them is the statistics factor $F(\epsilon)$, which reads $F_{\text{LL}}(\epsilon) = -\log(1 + e^{-\epsilon})$ and $F_{\text{HR}}(\epsilon) = -e^{-\epsilon}$. Note that, since both of them are Galilean invariant, the dispersion relation is given in the same way: $E_\theta = \theta^2/2$ and $p_\theta = \theta$.

## 5.2  Main predictions

Before jumping into the formulation of the BMFT for integrable systems, we collect in Subsecs. 5.2.1 and 5.2.2 the main results obtained from the BMFT for the cumulants and the correlation functions, respectively, so that readers who are interested in only results can simply consult. The details of the calculations for current fluctuations and cumulants are reported in Subsecs. 5.3 and 5.4 and in Appendix G, while correlation functions are discussed in Subsec. 5.5 and in Appendix H.

### 5.2.1  Cumulants

The evaluation of the SCGF $F(\lambda, T)$ is equivalent to computing all the cumulants $c_n(T) = \mathrm{d}^n F(\lambda, T)/\mathrm{d}\lambda^n|_{\lambda=0}$, see the definitions in Eqs. (14) and (15). The BMFT allows us to compute an arbitrary $c_n(T)$ by knowing how thermodynamic quantities change as $\lambda$ varies. The equation that turns out to be instrumental in computing $c_n(T)$ is the one for $\partial_\lambda \epsilon^\theta$, which we call the *flow equation* (see Eq. (147)), for the pseudo-energy $\epsilon^\theta$.

Using the flow equation, we can compute the first few cumulants for homogeneous initial conditions. The final formulas for the second and the third cumulants are given by

$$c_2^{\mathrm{hom}} = \int_{\mathbb{R}} \mathrm{d}\theta \, \chi_\theta |v_\theta^{\mathrm{eff}}| (h_{i_*}^{\mathrm{dr};\theta})^2, \tag{128}$$

$$c_3^{\mathrm{hom}} = \int_{\mathbb{R}} \mathrm{d}\theta \, \chi_\theta |v_\theta^{\mathrm{eff}}| h_{i_*}^{\mathrm{dr};\theta} \Big( s_\theta \tilde{f}_\theta \, (h_{i_*}^{\mathrm{dr};\theta})^2 + 3[sf \, (h_{i_*}^{\mathrm{dr}})^2]^{\mathrm{dr};\theta} \Big). \tag{129}$$

Let us explain the quantities that appear in Eqs. (128) and (129). First, the dressing operation is defined for any function of $\theta$ as $\mathsf{a}^{\mathrm{dr};\theta} = \int_{\mathbb{R}} \mathrm{d}\phi \, (R^{-\mathrm{T}})^\theta{}_\phi \mathsf{a}^\phi$, where the transformation matrix $R$ is defined by $R = 1 - nT$; the superscript T means the transpose, and the superscript $-\mathrm{T}$ is the transpose of the inverse. Note that the matrix $R$ diagonalises the flux Jacobian $\mathsf{A}_\theta{}^\phi = \partial(v_\theta^{\mathrm{eff}} \rho_\theta)/\partial \rho_\phi$, i.e., $RAR^{-1} = \mathrm{diag} \, v^{\mathrm{eff}}$. The quasi-particle susceptibility $\chi_\theta := \rho_\theta f_\theta$ is a statistics-dependent quantity where $f_\theta = f(\epsilon^\theta) = -(\mathrm{d}^2\mathsf{F}(\epsilon)/\mathrm{d}\epsilon^2)/(\mathrm{d}\mathsf{F}(\epsilon)/\mathrm{d}\epsilon)|_{\epsilon=\epsilon^\theta}$ (for example, $f_\theta = 1 - n_\theta$ for fermionic statistics). Further, we define $\tilde{f}_\theta := -(\mathrm{d}\log f(\epsilon)/\mathrm{d}\epsilon|_{\epsilon=\epsilon^\theta} + 2f_\theta)$, and $s_\theta$ is the sign of the effective velocity $s_\theta := \mathrm{sgn} \, v_\theta^{\mathrm{eff}}$.

Expressions (128) and (129) precisely coincide with the cumulants obtained previously using the BFT in Ref. [21]. Since the homogeneous BFT is built solely upon the tenet of Euler hydrodynamics (with the assumption of strong clustering in both space and time), the agreement gives an important consistency check for the BMFT.

The real advantage of the BMFT, however, is that it also allows us to calculate the SCGF $F(\lambda, T)$ and the cumulants for inhomogeneous initial conditions in a unified way. We emphasize that results for the SCGF $F(\lambda, T)$ and the cumulants in inhomogeneous initial states are scarce. So far, this problem has been, indeed, addressed only with the inhomogeneous BFT approach of Ref. [23]. Importantly, since it does not account for long-range correlation among fluid cells, which in general are present in ballistic many-body systems by our results as discussed in Sec. 1 and 3, this approach is at present not solidly founded. The BMFT approach accounts for all ballistic effects, including potential long-range correlations, to large scale fluctuations in interacting and inhomogeneous fluids.

We evaluated the second cumulant $c_2^{\mathrm{part}}$ in the partitioning protocol (recall that the cumulants $c_n^{\mathrm{part}}$ do not depend on $T$ for the partitioning protocol by scale invariance, see the discussion after Eqs. (14) and (17) in Subsec. 1).

In integrable systems, the partitioning protocol can be solved explicitly; the result is a space-time profile of states which depends only on the ray $\xi = x/t$, by scale invariance. The space-time profile is determined by the occupation function $n_\theta(\xi)$, which satisfies the following self-consistency equation,

$$n_\theta(\xi) = n_{R,\theta}\Theta(\xi - v_\theta^{\rm eff}(\xi)) + n_{L,\theta}\Theta(v_\theta^{\rm eff}(\xi) - \xi), \tag{130}$$

where the $\xi$-dependence of $v_\theta^{\rm eff}(\xi)$ is determined by $n_\theta(\xi)$. The occupation functions $n_{R,\theta}$ and $n_{L,\theta}$ are those corresponding to the left and right states, respectively, in (17). A remarkable feature of the solution is that, since $\theta$ is a continuous parameter, it naturally gives rise to a smooth profile of the fluid in space-time, where each fluid mode $\theta$ presents a single contact discontinuity [3, 16].

We find from the BMFT that $c_2^{\rm part}$ is *completely given by the thermodynamic quantities evaluated with respect to* $n_\theta(\xi = 0)$:

$$c_2^{\rm part} = \int_{\mathbb{R}} \mathrm{d}\theta\, \chi_\theta(0)\, |v_\theta^{\rm eff}(0)|\, (h_{i_*}^{{\rm dr};\theta}(0))^2, \tag{131}$$

Eq. (131) explicitly shows that $c_2^{\rm part}$, computed over the inhomogeneous and non-stationary partitioning protocol state, reduces to the cumulant (128) evaluated on the homogeneous NESS, the state at $\xi = 0$ emerging at long times in the partitioning protocol. This is a highly non-trivial statement since in inhomogeneous and non-stationary fluids [19,31], indirect effects, present if the model is interacting and not directly caused by the propagation of normal modes, means that correlations depend on the full inhomogeneous fluid profile and are not only determined by the fluid characteristics connecting the space-time points of interest. It turns out that these additional effects cancel out in the partitioning protocol when we count the statistics at $x = 0$; in general, however, these effects are present if we change the ray on which the statistics is evaluated or use other initial conditions.

In Subsec. 5.4, we present the detailed derivation of Eq. (131) for $c_2^{\rm part}$ with the BMFT formalism and we compare it with numerical simulations of the hard-rod model observing an excellent agreement. The derivation of Eq. (131) with the inhomogeneous version of BFT is reported in Appendix D. The numerical analysis we perform thereby validates our new BMFT.

We also checked that the inhomogeneous BFT yields Eq. (131). Hence for this cumulant in the partitioning protocol on the ray $\xi = 0$, the inhomogeneous BFT is correct, despite not taking into account long-range correlations. Within the BMFT, this can be understood technically by the fact that the contribution to $c_2^{\rm part}$ coming from the long-range part of the current-current correlator vanishes because the fluid velocity of the normal mode that propagates on the ray $\xi = 0$ is zero.

In principle the higher cumulants $c_{n \geq 4}$ can be computed by evaluating $\partial_\lambda^{n-1} \epsilon^\theta\big|_{\lambda=0}$, but the task gets increasingly convoluted. Although the exact expression is harder to write down, one can also calculate the SCGF $F(\lambda, T) = \int_{\mathbb{S}} \mathrm{d}t \mathrm{d}\theta \int_0^\lambda \mathrm{d}\lambda'\, h_{i_*}^\theta\, \mathsf{j}_\theta^{(\lambda')}(0, t)/T$ by solving the MFT equation with a $\lambda$-dependent initial condition (144), using the method of characteristics. We hope to look into this in a future work.

### 5.2.2  Dynamical correlation functions

As explained in the previous sections, the BMFT predicts the existence of long-range correlations in generic ballistic many-body systems. Since such long-range correlations had not been

predicted by any other means, it is of paramount importance to quantitatively evaluate them in a concrete model.

To this end, we computed the dynamical correlation function $S_{q_{i_1}, q_{i_2}}(x_1, t; x_2, t)$ for integrable systems using the BMFT, and we obtained the following formula, which specialises (111):

$$S_{\hat{q}_{i_1}, \hat{q}_{i_2}}(x_1, t; x_2, t) = \mathsf{C}_{i_1 i_2}(x_1, t)\delta(x_1 - x_2) + E_{i_1 i_2}(x_1, x_2; t), \tag{132}$$

where $\mathsf{C}_{i_1 i_2}(x_1, t) := \int_{\mathbb{R}} \mathrm{d}\theta \, [\chi_\theta h_{i_1}^{\mathrm{dr};\theta} h_{i_2}^{\mathrm{dr};\theta}](x_1, t)$ is the local covariance matrix, and $E_{i_1 i_2}(x_1, x_2; t) := -\int_{\mathbb{R}} \mathrm{d}\theta \, [\chi_\theta h_{i_2}^{\mathrm{dr};\theta} \mathcal{E}^\theta](x_2, t)$ is the term that represents long-range correlations. The symbol $[\bullet](x, t)$ means that all the quantities inside the bracket are evaluated at the space-time point $(x, t)$. The function $\mathcal{E}^\theta(x, t)$ satisfies the integral equation

$$\mathcal{E}^\theta(x, t) = \mathcal{E}_0^\theta(x, t) + w^\theta(x, t) \int_{-\infty}^x \mathrm{d}y \, [\chi\mathcal{E}]^{\mathrm{dr};\theta}(y, t), \tag{133}$$

where we defined (here and below, $\partial$ is the derivative with respect to the spatial argument)

$$w^\theta(x, t) := \frac{\partial \epsilon_{\mathrm{ini}}^\theta(u_\theta(x, t))}{\rho_\theta^{\mathrm{tot}}(u_\theta(x, t), 0)}, \tag{134}$$

with the initial pseudo-energy $\epsilon_{\mathrm{ini}}^\theta(x)$ related to $\beta_{\mathrm{ini}}^i(x)$ via (124). We set $u_\theta(x, t) := \mathcal{U}_\theta(x, t; 0)$, where $\mathcal{U}_\theta(x, t; s)$ defines the fluid characteristics for the mode $\theta$: it is the spatial coordinate of the characteristic line, at time $s$, that passes through $(x, t)$. It satisfies [75]

$$\int_{-\infty}^{\mathcal{U}_\theta(x,t;s)} \mathrm{d}y \, \rho_\theta^{\mathrm{tot}}(y, s) + v_\theta^- (t - s) = \int_{-\infty}^x \mathrm{d}y \, \rho_\theta^{\mathrm{tot}}(y, t), \tag{135}$$

with $v_\theta^- := \lim_{x=-\infty} v_\theta^{\mathrm{eff}} \rho_\theta^{\mathrm{tot}}$.

Finally, the source term of the integral equation (133), i.e., $\mathcal{E}_0^\theta(x, t)$, is written as a sum of two terms $\mathcal{E}_0^\theta(x, t) = \mathcal{D}_1^\theta(x, t) + \mathcal{D}_2^\theta(x, t)$. The first term reads

$$\mathcal{D}_1^\theta(x, t) := \int_{\mathbb{R}^2} \mathrm{d}\phi \, \mathrm{d}\alpha \, (R^{-\mathrm{T}})^\theta_{\phi}(u_\theta(x, t), 0)\partial \left[ (R^{\mathrm{T}})^\phi_{\alpha} h^{\mathrm{dr};\alpha} \right](x_1, t)\Theta(u_\theta(x, t) - u_\alpha(x_1, t))$$
$$- w^\theta(x, t)[\chi h^{\mathrm{dr}}]^{\mathrm{dr};\theta}(x_1, t)\Theta(x_2 - x_1). \tag{136}$$

The second term comes from the initial condition

$$\mathcal{D}_2^\theta(x, t) := -w^\theta(x, t) \int_{-\infty}^{u_\theta(x,t)} \mathrm{d}y \, [\chi\mathcal{D}_3]^{\mathrm{dr};\theta}(y, 0), \tag{137}$$

where $\mathcal{D}_3^\theta(x, 0)$ is given by

$$\mathcal{D}_3^\theta(x, 0) := -h^{\mathrm{dr};\theta}(x_1, t)\frac{\delta(x - u_\theta(x_1, t))}{\partial \mathcal{U}_\theta(u_\theta(x_1, t), 0; t)}$$
$$+ \int_{\mathbb{R}^2} \mathrm{d}\phi \, \mathrm{d}\alpha \, (R^{-\mathrm{T}})^\theta_{\phi}(x, 0)\partial \left[ (R^{\mathrm{T}})^\phi_{\alpha} h^{\mathrm{dr};\alpha} \right](x_1, t)\Theta(x - u_\alpha(x_1, t)). \tag{138}$$

Despite the tedious expression of the long-range correlation term $E_{i_1 i_2}(x_1, x_2; t)$, the numerical evaluation of it turns out to agree well with its value obtained from hard-rod simulations. This confirms that not only do the long-range correlations exist at least in integrable systems but

also they can be accurately computed by the BMFT. The comparison between the predictions by the BMFT for the correlator $S_{\hat{q}_0, \hat{q}_0}(x, t; 0, t)$, from Eqs. (132)-(138) specialized to the hard-rod model (cf. Eqs. (38) and (39)), and numerics is reported in Fig. (1) of the companion manuscript [24]. For the correlator $S_{\hat{q}_0, \hat{q}_0}(x, t; -x, t)$, the comparison is reported in Sec. 5.5 in Fig. 4.

## 5.3  Current fluctuations in integrable systems

In what follows, for brevity, much like for romain indices, we will also use Einstein's "summation convention" for rapidities: whenever expressions contain factors with repeated *upper and lower* rapidity indices, integrals over $\mathbb{R}$ are understood, for instance

$$a^\theta b_\theta \equiv \int_{\mathbb{R}} \mathrm{d}\theta \, a^\theta b_\theta. \tag{139}$$

As is required, we will also use $v^{\mathrm{eff};\theta} = v_\theta^{\mathrm{eff}}$.

Let us start with current fluctuations, for which we follow the procedure of Subsec. 4.1. The action to be minimised (cf. Eq. (74)) is

$$S_{\mathrm{curr}}[\rho, H] = \mathcal{F}_{\mathrm{curr}}[\rho] + \int_{\mathbb{S}} \mathrm{d}t \mathrm{d}x \, H^\theta(x, t)(\partial_t \rho_\theta + \partial_x j_\theta[\rho]), \tag{140}$$

where $\mathcal{F}_{\mathrm{curr}}[\rho] = \mathcal{F}[\rho(\cdot, 0)] - \lambda \int_0^T \mathrm{d}t \, h_{i_*}^\theta j_\theta[\rho]$. Let us redefine $h^\theta := h_{i_*}^\theta$ for brevity. Accordingly the set of MFT equations, from Eq. (76), for integrable models are

$$\lambda h^\theta \Theta(x) - \beta^\theta(x, 0) + \beta_{\mathrm{ini}}^\theta(x) - H^\theta(x, 0) = 0, \tag{141a}$$

$$\lambda h^\theta \Theta(x) - H^\theta(x, T) = 0, \tag{141b}$$

$$\partial_t \beta^\theta(x, t) + \mathsf{A}_\phi^\theta[\beta(x, t)] \partial_x \beta^\phi(x, t) = 0, \tag{141c}$$

$$\partial_t H^\theta(x, t) + \mathsf{A}_\phi^\theta[\beta(x, t)] \partial_x H^\phi(x, t) = 0. \tag{141d}$$

Note that the third equation is equivalent to the usual GHD equation in terms of the density of particle $\partial_t \rho_\theta + \partial_x(v_\theta^{\mathrm{eff}} \rho_\theta) = 0$ via $\mathsf{AC} = \mathsf{CA}^{\mathrm{T}}$. To solve these equations self-consistently, we shall first recall how one can solve the initial-value problems in GHD. For simplicity, we treat an integrable many-body system where quasi-particles have a single species and scatter diagonally (e.g., Lieb-Liniger model). Let us start with introducing normal modes. One usually introduces the normal modes by diagonalising the *linearised* Euler equation, but in integrable systems it has been known that normal modes exists even in the fully non-linear GHD equation, which are actually not unique [16, 17]. We choose and define our normal mode by $\partial_{t,x} \epsilon^\theta := (R^{-\mathrm{T}}(x, t))^\theta{}_\phi \partial_{t,x} \beta^\phi$, where the pseudo-energy $\epsilon^\theta$ was already defined in (124). Accordingly we rewrite (141c) as $\partial_t \epsilon^\theta + v^{\mathrm{eff};\theta} \partial_x \epsilon^\theta = 0$ [16, 17]. It is clear that any function of $\epsilon^\theta$, e.g., the occupation function $n^\theta$, is also transported in a convective fashion as $\epsilon^\theta$, hence is eligible for being a normal mode. Motivated by this we also introduce a normal mode $G^\theta$ for the auxiliary field $H^\theta$ in the same way:

$$\partial_{t,x} G^\theta := (R^{-\mathrm{T}}(x, t))^\theta{}_\phi \partial_{t,x} H^\phi. \tag{142}$$

While a priori it is not obvious if such a normal mode could exist, it turns out in integrable systems that it does. This is because a compatibility condition $\partial_t \partial_x G^\theta = \partial_x \partial_t G^\theta$ holds due

to the fact that the row of the (transposed) transformation matrix $R^{\mathrm{T}}$ is a normal mode: $\partial_t(R^{\mathrm{T}})^\theta{}_\phi + v^{\mathrm{eff}}_\phi \partial_x (R^{\mathrm{T}})^\theta{}_\phi = 0$, which is obvious from the definition of $R$.

In terms of the normal modes, one readily obtains the solution of the MFT equations. We first solve the equation for $G^\theta(x,t)$ given by

$$\lambda h^{\mathrm{dr};\theta}(0,T)\Theta(x) - G^\theta(x,T) = 0, \tag{143a}$$

$$\partial_t G^\theta(x,t) + v^{\mathrm{eff};\theta}(x,t)\partial_x G^\theta(x,t) = 0. \tag{143b}$$

Note that in principle there is an additional constant term in (143a), which is $G^\theta(-\infty, T)$. This however can always be set to zero, as this goes away when transforming back to $H^\theta$, whose boundary conditions are specified after Eq. (76).

Since $G^\theta(x,t)$ is a normal mode, it admits the solution $G^\theta(x,t) = G^\theta(r_\theta(x,t), T) = h^{\mathrm{dr};\theta}(0,T)\Theta(r_\theta(x,t))$, where we defined $r_\theta(x,t) := \mathcal{U}_\theta(x,t;T)$ from Eq. (135). The full spacetime profile of $G^\theta$ is however of no importance, and we merely use it to write down a self-consistent initial condition for $\beta^\theta(x,0)$:

$$\beta^\theta(x,0) = \beta^\theta_{\mathrm{ini}}(x) + \lambda h^\theta \Theta(x) - \lambda(R^{\mathrm{T}})^\theta{}_\phi(0,T)\Theta(x - u_\phi(0,T))h^{\mathrm{dr};\phi}(0,T). \tag{144}$$

Therefore the MFT dynamics is now recast into GHD with the $\lambda$-dependent initial condition given in the above self-consistent way, which again can be solved by the method of characteristics. A somewhat special initial condition that complicates these considerations is the partitioning protocol in Eq. (85), where one starts with a step initial condition $\beta^\theta_{\mathrm{ini}}(x) = \beta^\theta_L \Theta(-x) + \beta^\theta_R \Theta(x)$. This situation calls for a more careful treatment, as the flow equation $\partial_\lambda \epsilon^\theta(x,t)$ generically contains $h^{\mathrm{dr}}(0,0)$, which depends on the regularisation chosen in the partitioning protocol. Such dependence on the regularisation at $x = 0$ is also reflected in the fact that the transformation matrix $R(x,t)$ defined as above becomes ill-defined at $x = t = 0$ due to $\delta(0)$ that stems from $\partial_x \beta^\theta(x,0)$ at $\lambda = 0$. Fortunately in integrable systems one can directly define $R$ using the integral equation that defines $\epsilon^\theta$, which yields $\partial_\lambda \epsilon^\theta(x,t) = (R^{-\mathrm{T}}(x,t))^\theta{}_\phi \partial_\lambda \beta^\phi(x,t)$. Note that which regularisation to use does not affect the definition of $G^\theta$, as $H^\theta$ and its derivatives are zero at $\lambda = 0$ anyway.

## 5.4 Flow equation and cumulants

To evaluate the cumulants, one needs to know how the fluid variables change as $\lambda$ varies, i.e., $\partial_\lambda \epsilon^\theta(x,t)$ (from which one recovers the derivatives of other variables). In general, the flow equation for $\partial_\lambda \epsilon^\theta(x,t)$ takes a cumbersome form, but in the homogeneous case $\beta^\theta_{\mathrm{ini}}(x) = \beta^\theta_{\mathrm{ini}}$, where $\beta^\theta_{\mathrm{ini}}$ does not depend on the space coordinate $x$, it is given in a simple way. Note from (144) that, using $\partial_x \beta^\theta := (R^{\mathrm{T}}(x,t))^\theta{}_\phi \partial_x \epsilon^\phi$, we have

$$\partial_x \epsilon^\theta(x,0) = \lambda h^{\mathrm{dr};\theta}(0,0)\delta(x) - \lambda\delta(x - u_\theta(0,T))h^{\mathrm{dr};\theta}(0,T), \tag{145}$$

where we used $(R^{\mathrm{T}})^\phi{}_\theta(u_\theta(x,t), 0) = (R^{\mathrm{T}})^\phi{}_\theta(x,t)$ and note that we do not take summations over rapidities when the only quantities with lower indices are $u_\theta$ (or $r_\theta$). Upon integrating over $x$ and invoking $\epsilon^\theta(x,t) = \epsilon^\theta(u_\theta(x,t), 0)$, the full profile of the normal mode is readily obtained as

$$\epsilon^\theta(x,t) = \epsilon^\theta_{\mathrm{ini}} + \lambda\big(h^{\mathrm{dr};\theta}(0,0)\Theta(u_\theta) - h^{\mathrm{dr};\theta}(0,T)\Theta(r_\theta)\big), \tag{146}$$

where the identity $\Theta(u_\theta(x,t) - u_\theta(0,T)) = \Theta(r_\theta(u_\theta(x,t),0)) = \Theta(r_\theta(x,t))$ was invoked and we introduced the shorthanded notation $u_\theta := u_\theta(x,t)$ and $r_\theta := r_\theta(x,t)$. What we are after however is not the explicit $\epsilon_\theta(x,t)$ but rather its derivative by $\lambda$, which is obviously given by

$$\partial_\lambda \epsilon^\theta(x,t) = \partial_\lambda \left[ \lambda(h^{\mathrm{dr};\theta}(0,0)\Theta(u_\theta) - h^{\mathrm{dr};\theta}(0,T)\Theta(r_\theta)) \right]. \tag{147}$$

This is the sought flow equation in the homogeneous case. The flow equation with a generic inhomogeneous initial condition (barring the partitioning protocol, which will be treated separately later) is reported in Appendix F. The flow equation (147) essentially encodes all the information needed to compute the cumulants that are given by $c_n = \int_0^T dt\, \partial_\lambda^{n-1} h^\theta j_\theta(0,t)\big|_{\lambda=0}$, where we note that the cumulants do not depend on $T$ in the homogeneous case. Notice also that alternatively they can also be written as $c_n = \partial_\lambda^{n-1} \int_0^\infty dx\, h^\theta (\rho_\theta(x,T) - \rho_\theta(x,0))\big|_{\lambda=0}$, which we shall use for evaluating the cumulants.

As explained in Subsec. 4.2, $\beta(0,t)$ in the BMFT ($\theta$ and $\lambda$ dependence are suppressed for brevity) can be identified with $\beta(\lambda)$ in the BFT if $\beta(0,t)$ is time-independent and satisfies the BFT flow equation (79). It turns out that verifying these is still challenging even for integrable systems, and only thing we can immediately notice is that at $\lambda = 0$ the MFT flow equation (147) becomes that of $\epsilon^\theta(\lambda)$ in the BFT, which indicates that the identification is true at least up to the first order in $\lambda$. Further analysis on the structure of the BMFT flow equation (147) is left for future studies.

With the flow equation at our disposal, let us see how it can be used to compute $c_2$ for the homogeneous initial condition. We first note

$$\partial_\lambda \rho_\theta(x,t) = -(R^{-\mathrm{T}})^\phi_{\ \theta}(x,t)\chi_\phi(x,t)\partial_\lambda \epsilon^\phi(x,t), \tag{148}$$

where we used $\partial\rho_\theta/\partial\epsilon^\phi = -(R^{-\mathrm{T}})^\phi_{\ \theta}\chi_\phi$, from which we get

$$\lim_{\lambda\to 0} \partial_\lambda \rho_\theta(x,t) = (R^{-\mathrm{T}})^\phi_{\ \theta} h^{\mathrm{dr};\phi}\chi_\phi(\Theta(r_\phi) - \Theta(u_\phi)), \tag{149}$$

where the thermodynamic quantities without arguments are meant to be evaluated with respect to the initial homogeneous state. Since $u_\theta(x,t) = x - v_\theta^{\mathrm{eff}} t$ and $r_\theta(x,t) = x - v_\theta^{\mathrm{eff}}(t-T)$ when $\lambda = 0$, it is a simple matter to observe that the homogeneous cumulant $c_2^{\mathrm{hom}}$ reads

$$c_2^{\mathrm{hom}} = \int_\mathbb{R} d\theta\, \chi_\theta |v_\theta^{\mathrm{eff}}|(h^{\mathrm{dr};\theta})^2, \tag{150}$$

where and we restored the integral for clarity. This is precisely the second cumulant that has also been obtained using different methods previously [21,28]. Following the same logic, albeit growing complexity as $n$ increases, one can in principle compute arbitrary higher cumulants $c_n$. See Appendix G for the computation of $c_3$ (129).

One of the virtues of the MFT is that it allows us to compute cumulants for arbitrary initial conditions that are not homogeneous following the same procedures. That being said, as mentioned before, the partitioning protocol requires a separate consideration due to the singularity of the rotation matrix $R(0,0)$ at $x = t = 0$. Firstly the flow equation for $\partial_\lambda \epsilon^\theta(x,t)$ takes a tedious and not so informative form (see Appendix F). As it turns out, it is more convenient to evaluate the alternative form of $c_2$, which is

$$c_2 = \partial_\lambda \int_0^T dt\, h^\theta \rho_\theta(0,t) v_\theta^{\mathrm{eff}}(0,t)\bigg|_{\lambda=0} = -h^{\mathrm{dr};\theta}(0)\chi_\theta(0)v_\theta^{\mathrm{eff}}(0) \int_0^T dt\, \partial_\lambda \epsilon^\theta(0,t)\bigg|_{\lambda=0}, \tag{151}$$

where the thermodynamic quantities with arguments 0 are evaluated at $\xi = x/t = 0$. Therefore we have to compute $\partial_\lambda \epsilon^\theta(0,t)\big|_{\lambda=0}$, which can be written as

$$\partial_\lambda \epsilon^\theta(0,t) = \partial_\lambda u_\theta \, \partial_x \epsilon^\theta(x,0)\Big|_{x=u_\theta(0,t)} + (\partial_\lambda \epsilon^\theta)(u_\theta(0,t),0), \tag{152}$$

where $\lambda = 0$ is taken in the right hand side. We hereafter assume that $\lambda = 0$ is always taken in each equation at the end of manipulations unless otherwise stated. Note that in fact the first term does not contribute. To see this, we recall that, at $\lambda = 0$, $\partial_x \epsilon(x,0) = (\epsilon_R^\theta - \epsilon_L^\theta)\delta(x)$ ($\epsilon_{R/L}^\theta$ are the pseudo-energies of the initial right/left subsystems), hence $\partial_x \epsilon^\theta(x,0)\big|_{x=u_\theta(0,t)}$ is proportional to $\delta(u_\theta(0,t))$. Since in the partitioning protocol we have $\delta(u_\theta(0,t)) = \delta(-v_\theta^{\rm eff}(0)t) = \delta(v_\theta^{\rm eff}(0))/t$ when $t > 0$, this gives zero when multiplied by $v_\theta^{\rm eff}(0)$ in (151). The task therefore boils down to calculate $(\partial_\lambda \epsilon^\theta)(u_\theta(0,t),0)$.

Using (144), $(\partial_\lambda \epsilon^\theta)(u_\theta(0,t),0)$ is given by

$$(\partial_\lambda \epsilon^\theta)(u_\theta(0,t),0) = h^{{\rm dr};\theta}(u_\theta(0,t),0)\Theta(u_\theta(0,t))$$
$$- (R^{-{\rm T}})^\theta_{\phantom{\theta}\phi}(u_\theta(0,t),0)(R^{\rm T})^\phi_{\phantom{\phi}\gamma}(0,T)\Theta(u_\theta(0,t) - u_\gamma(0,T))h^{{\rm dr};\gamma}(0,T). \tag{153}$$

To proceed, we need to invoke a few important relations. First, it is a simple matter to see

$$\partial_t u_\theta(0,t) = -\frac{v_\theta(0,t)}{\rho_\theta^{\rm tot}(u_\theta(0,t),0)}, \tag{154}$$

with $v_\theta(x,t) = v_\theta^{\rm eff}(x,t)\rho_\theta^{\rm tot}(x,t)$ from Eq. (135). In the partitioning protocol, the solutions are self-similar (i.e., they depend only on $\xi = x/t$), hence we have $v_\theta(0,t) = v_\theta(\xi = 0)$ when $t > 0$. This implies that $u_\theta(0,t)$ is either monotonically increasing or decreasing depending on the sign of $v_\theta(0)$. Since $u_\theta(0,0) = 0$, we conclude that

$$\Theta(u_\theta(0,t)) = \Theta(-v_\theta^{\rm eff}(0)). \tag{155}$$

A similar observation can be made to obtain $\Theta(r_\theta(0,t)) = \Theta(v_\theta^{\rm eff}(0))$. Another relation is that $(R^{\rm T})^\theta_{\phantom{\theta}\phi}(x,t)$ is a normal mode with respect to $\phi$, i.e., $(R^{\rm T})^\theta_{\phantom{\theta}\phi}(x,t) = (R^{\rm T})^\theta_{\phantom{\theta}\phi}(u_\phi(x,t),0)$. This means that $(R^{\rm T})^\phi_{\phantom{\phi}\gamma}(0,T) = (R^{\rm T})^\phi_{\phantom{\phi}\gamma}(u_\gamma(0,T),0)$, which in turn allow us to compute the building block $(R^{-{\rm T}})^\theta_{\phantom{\theta}\phi}(u_\theta(0,t),0)(R^{\rm T})^\phi_{\phantom{\phi}\gamma}(u_\gamma(0,T),0)\Theta(u_\theta(0,t) - u_\gamma(0,T))$. Importantly, this quantity depends only on the sign of $v_\theta^{\rm eff}(0)$ and $v^{{\rm eff};\gamma}(0)$. Thus when the signs of both velocities are the same, it simply gives $\delta^\theta_{\phantom{\theta}\gamma}\Theta(v_\theta^{\rm eff})$, where we used $\Theta(u_\theta(0,t) - u_\theta(0,T)) = \Theta(v_\theta^{\rm eff})$ because $t < T$. When the signs differ, only situation when it has a nonzero contribution is when $v_\theta^{\rm eff}(0) < 0 < v_\gamma^{\rm eff}(0)$. Combining these, we obtain

$$(R^{-{\rm T}})^\theta_{\phantom{\theta}\phi}(u_\theta,0)(R^{\rm T})^\phi_{\phantom{\phi}\gamma}(u_\gamma,0)\Theta(u_\theta - u_\gamma) = \delta^\theta_{\phantom{\theta}\gamma}\Theta(v_\theta^{\rm eff}) + \Theta(-v_\theta^{\rm eff})\Theta(v_\gamma^{\rm eff})(R_R^{-{\rm T}})^\theta_{\phantom{\theta}\phi}(R_L^{\rm T})^\phi_{\phantom{\phi}\gamma}, \tag{156}$$

where temporarily we suppressed the argument of $u_\theta(0,t)$, $u_\gamma(0,T)$ and $v_\theta^{\rm eff}(0)$. Therefore we end up with

$$(\partial_\lambda \epsilon^\theta)(u_\theta(0,t),0) = \left[h_R^{{\rm dr};\theta} - \Theta(v_\gamma^{\rm eff})(R_R^{-{\rm T}})^\theta_{\phantom{\theta}\phi}(R_L^{\rm T})^\phi_{\phantom{\phi}\gamma}h^{{\rm dr};\gamma}(0)\right]\Theta(-v_\theta^{\rm eff}) - h^{{\rm dr};\theta}(0)\Theta(v_\theta^{\rm eff})$$
$$= (R_R^{-{\rm T}})^\theta_{\phantom{\theta}\phi}(R^{\rm T})^\phi_{\phantom{\phi}\gamma}(0)\left[h^{{\rm dr};\gamma}(0) - \Theta(v_\gamma^{\rm eff})h^{{\rm dr};\gamma}(0)\right]\Theta(-v_\theta^{\rm eff}) - h^{{\rm dr};\theta}(0)\Theta(v_\theta^{\rm eff})$$
$$= -h^{{\rm dr};\theta}(0){\rm sgn}(v_\theta^{\rm eff}(0)), \tag{157}$$

where we used $\Theta(v_\gamma^{\mathrm{eff}})(R_L^{\mathrm{T}})^\phi{}_\gamma = \Theta(v_\gamma^{\mathrm{eff}})(R^{\mathrm{T}})^\phi{}_\gamma(0)$ and $\Theta(-v_\gamma^{\mathrm{eff}})(R_R^{\mathrm{T}})^\phi{}_\gamma = \Theta(-v_\gamma^{\mathrm{eff}})(R^{\mathrm{T}})^\phi{}_\gamma(0)$ when passing from the second to the third, and from the third to the fourth line, respectively. Plugging this back into (151), we finally obtain the cumulant in the partitioning protocol $c_2^{\mathrm{part}}$:

$$c_2^{\mathrm{part}} = \int_{\mathbb{R}} \mathrm{d}\theta \, \chi_\theta(0) |v_\theta^{\mathrm{eff}}(0)| (h_{i_*}^{\mathrm{dr};\theta}(0))^2. \tag{158}$$

This is the result anticipated in Eq. (131) of Subsec. 5.2. We again emphasize that in the previous equation, where the counting statistics is performed at the point $x = 0$, the terms caused by the interactions among normal modes cancel out. This determines the simple expression in Eq. (158), where all the quantities are evaluated on the single ray $\xi = 0$, corresponding to the homogeneous NESS of the partitioning protocol.

In order to test the non-trivial prediction of Eq. (158), we perform simulations of the interacting hard-rod model, which we already introduced in Sec. 2.

The rods are initialized in the inhomogeneous partitioning protocol state in Eq. (85) with only the inverse temperature Lagrange multiplier being non zero, i.e, the set $\mathcal{C}$ contains only the energy conserved charge (and $\beta_0^i = 0$ otherwise). The initial state has therefore the form in Eq. (17). The inverse temperatures $\beta_{L,R}$ and the rod length $a$ therefore fix the rod densities $\rho(\beta_{L,R})$ of the left ($x < 0$) and the right ($x > 0$) half. The rods' positions are initially distributed in a symmetric interval $[-L_{\mathrm{size}}/2, L_{\mathrm{size}}/2]$ around the origin according to the aforementioned thermal densities $\rho(\beta_{L,R})$, with $N_L = \rho(\beta_L)L_{\mathrm{size}}/2$ and $N_R = \rho(\beta_R)L_{\mathrm{size}}/2$ rods initially in the left and the right half, respectively. The rods' velocities are sampled from the thermal velocity distribution, which is a Gaussian with variance given by the corresponding inverse temperature $1/\beta_{L,R}$. Statistical fluctuations are thereby solely determined by the initial sample of the positions and velocities, while the dynamics is fully deterministic. In the numerical analysis, we focus on particle transport, with single particle eigenvalue $h_{i_*}^\theta = 1$, for the sake of simplicity. We count numerically the number of rods transferred from the left to the right half over a time interval $T$ since the start of the dynamics. The cumulants are eventually computed by rescaling by the time duration $T$ and by averaging over a large number $M$ of independent samples of the initial rods' distribution. The comparison between the prediction in Eq. (158) and the value of $c_2^{\mathrm{part}}$ from the numerical simulations of the hard-rod gas is shown in Fig. 3.

From the figure we can see that the numerical data deviate from the Euler-scale prediction for short times. This is caused by the fact, cf. the discussion after Eq. (17) in Subsec. 2.1, that the initial state does not show large wavelengths variations. Corrections coming from the microscopic nature of the initial state, on the one hand, cause the numerical results to deviate from the Euler-scale prediction (158) at short times. At long times, however, the partitioning protocol initial state quickly relaxes to a smoothly varying state exhibiting large scale variations and the Euler-scale prediction is recovered with an excellent precision. The discrepancy between Eq. (158) and the numerical data, at long times $T \in [10, 15]$ in the inset of Fig. 3, is, indeed, observable only on the fourth decimal digit and it is well within the statistical uncertainty bars. The latter are computed by propagating the statistical uncertainty of the computed mean transferred particle number (and powers thereof) as detailed in Appendix I.

We emphasize that the prediction in Eq. (158), with the comparison with the numerical simulations in Fig. 3, represents, to our knowledge, the first result in integrable models for cumulants evaluated over inhomogeneous and non-stationary states, such as the partitioning protocol state (17). For integrable systems, as a matter of fact, results for the cumulants were so far limited to the simpler case of homogeneous and stationary states, such as Eq. (128)

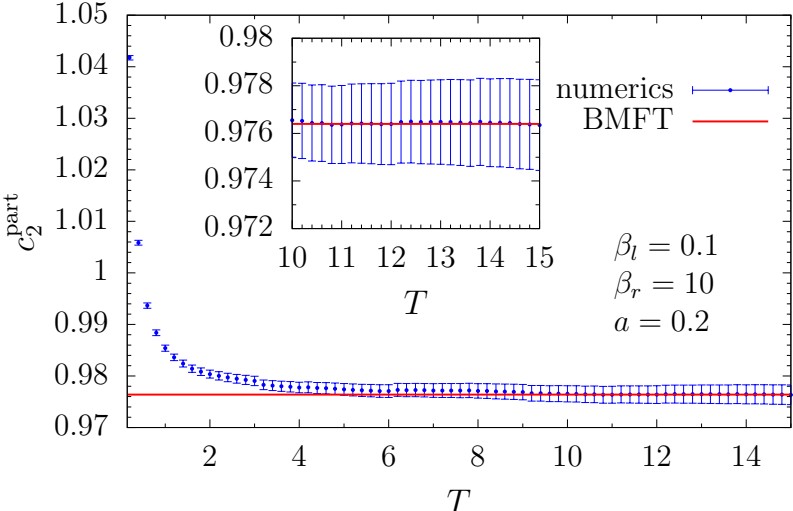

Figure 3: **Second cumulant in the partitioning protocol of the hard-rod gas.** The figure displays an excellent agreement between the numerical simulations and the BMFT prediction. The latter for the second cumulant $c_2^{\text{part}}$ of the particle current is given by Eq. (158) with $h_{i*}^\theta = 1$ (red solid line). The numerical data are obtained by computing the particle number second cumulant rescaled by the time $T$ (blue points with the corresponding statistical uncertainties' bars). In the figure $T \in [0.2, 15]$. In the inset, the data at long times, $T \in [10, 15]$, are zoomed in to highlight the excellent agreement therein between the BMFT prediction and the numerical simulations. The initial rods' distribution present a step at $x = 0$ in the inverse temperature $\beta_l = 0.1$ ($x < 0$) and $\beta_r = 10$ ($x > 0$). The rod length is $a = 0.2$, the initial length $L_{\text{size}}$ where the rods are distributed is $L_{\text{size}} = 10^5$, the number of rods initially on the left and right sides are $N_L = 42610$ and $N_R = 6007$, respectively. A number $M = 8.8 \times 10^7$ of independent statistical samples has been used.

and (129) for the NESS emerging at long times from the partitioning protocol, as shown in Ref. [21]. We also checked that Eq. (158) is obtained within the inhomogeneous version of the BFT theory. The numerical analysis of Fig. 3 thus also provides the first numerical confirmation of the inhomogeneous BFT, at least for the cumulants on the ray $\xi = 0$ in the partitioning protocol. In more general cases the inhomogeneous BFT requires additional checks since it does not assume any long-range correlation on equal-time correlation functions. In passing, we mention that at intermediate times, $T \in [4, 10]$ in the Figure, the numerical data are consistent with a power-law relaxation to the Euler-scale prediction as a function time. This behavior might be related to diffusive sub-leading, i.e., $\mathcal{O}(\sqrt{\ell})$, corrections to the BMFT action in Eq. (46) (see also the discussion in Sec. 3.3). This effect goes therefore beyond the scope of the present manuscript and its analysis is left for future investigations.

## 5.5 Dynamical correlation functions and universal long-range correlations

We follow here the analysis of Subsec. 4.4. The action $S_{\hat{q}_{i_1}, \hat{q}_{i_2}}(x_1, t_1; x_2, t_2)$ whose saddle point characterises the Euler scale dynamical correlation function is (cf. Eq. (96))

$$S_{\text{corr}}[\rho, H] := \mathcal{F}_{\text{corr}}[\rho] + \int_{\mathbb{S}} \mathrm{d}t \mathrm{d}x \, H^\theta(x, t)(\partial_t \rho_\theta + \partial_x \mathsf{j}_\theta[\rho]), \qquad (159)$$

where $\mathcal{F}_{\mathrm{corr}}[\rho] = \mathcal{F}[\rho(\cdot, 0)] - (\lambda_1 h^\theta_{i_1} \rho_\theta(x_1, t_1) + \lambda_2 h^\theta_{i_2} \rho_\theta(x_2, t_2))$. Following the same reasoning in the generic case, the set of MFT equations are given in Eq. (98), which for integrable systems read as

$$\beta^\theta(x, 0) - \beta^\theta_{\mathrm{ini}}(x) + H^\theta(x, 0) = 0, \tag{160a}$$

$$H^\theta(x, T) = 0, \tag{160b}$$

$$\partial_t \beta^\theta + \mathsf{A}^\theta_\phi \partial_x \beta^\phi = 0, \tag{160c}$$

$$\partial_t H^\theta + \mathsf{A}^\theta_\phi \partial_x H^\phi + \lambda h^\theta_{i_1} \delta(x - x_1) \delta(t - t_1) = 0. \tag{160d}$$

The correlator is then given from Eqs. (97) and (148) by

$$S_{\hat{q}_{i_1}, \hat{q}_{i_2}}(x_1, t_1; x_2, t_2) = - \left. [(h_{i_2})^{\mathrm{dr};\theta} \chi_\theta \partial_\lambda \epsilon^\theta](x_2, t_2) \right|_{\lambda=0}. \tag{161}$$

As in the case of the current fluctuations, let us derive the initial condition given in a self-consistent way. Integrating both sides of the equation for $H^\theta(x, t)$ over $[t_1 - \varepsilon, t_1 + \varepsilon]$, we get

$$H^\theta(x, t_1 + \varepsilon) - H^\theta(x, t_1 - \varepsilon) = -\lambda h^\theta_{i_1} \delta(x - x_1). \tag{162}$$

From the boundary condition $H^\theta(x, T) = 0$ it is clear that $H^\theta(x, t) = 0$ for $t > t_1$, hence we obtain $H^\theta(x, t_1 - \varepsilon) = \lambda h^\theta_{i_1} \delta(x - x_1)$. In order to obtain the full time profile of $H^\theta(x, t)$, one has to first move to the normal modes and invoke $G^\theta(x, t) = G^\theta(\mathcal{U}_\theta(x, t; t_1), t_1)$, obtaining

$$G^\theta(x, t) = \lambda \Big( h^{\mathrm{dr};\theta}(x_1, t_1) \delta(\mathcal{U}_\theta(x, t; t_1) - x_1) - \partial_{x_1} h^{\mathrm{dr};\theta}(x_1, t_1) \Theta(\mathcal{U}_\theta(x, t; t_1) - x_1) \Big) \Theta(t_1 - t), \tag{163}$$

where we defined $h := h_{i_1}$. In particular, at time $t = 0$ we get

$$G^\theta(x, 0) = \lambda \Big( h^{\mathrm{dr};\theta}(x_1, t_1) \frac{\delta(x - u_\theta(x_1, t_1))}{\partial \mathcal{U}_\theta(u_\theta(x_1, t_1), 0; t_1)} - \partial_{x_1} h^{\mathrm{dr};\theta}(x_1, t_1) \Theta(x - u_\theta(x_1, t_1)) \Big), \tag{164}$$

where we used $t_1 > 0$ and $x_1 = \mathcal{U}^\theta(u_\theta(x_1, t_1), 0; t_1)$. Notice that in Eqs. (163) and (164) we have set the additive constant $G^\theta(-\infty, t)$ to zero without loss of generality for the same reason as after Eq. (143). In the previous equation, the derivative $\partial \mathcal{U}_\theta(u_\theta(x_1, t_1), 0; t_1)$ is taken with respect to $u_\theta(x_1, t_1)$. Transforming back to $H^\theta$, one observes that the terms can be reorganised nicely, which yields the following $\beta^\theta(x, 0)$:

$$\beta^\theta(x, 0) = \beta^\theta_{\mathrm{ini}}(x) + \lambda \partial_{x_1} \left( (R^{\mathrm{T}})^\theta{}_\phi(x_1, t_1) h^{\mathrm{dr};\phi}(x_1, t_1) \Theta(x - u_\phi) \right), \tag{165}$$

where $u_\phi := u_\phi(x_1, t_1)$. See the Appendix H.1 for the full profile of $\partial_\lambda \epsilon^\theta(x, t)$ at $\lambda = 0$ in Eq. (229).

Before making a crucial observation in the inhomogeneous case, let us compute the correlator for the homogeneous initial condition. Using (165), we have at $\lambda = 0$

$$\partial_\lambda \epsilon^\theta(x_2, t_2) = \partial_\lambda u_\theta \partial_y \left. \epsilon^\theta(y, 0) \right|_{y=u_\theta} + (\partial_\lambda \epsilon)(u_\theta, 0) = -(h_{i_1})^{\mathrm{dr};\theta} \delta(x_2 - x_1 - v^{\mathrm{eff}}_\theta(t_2 - t_1)), \tag{166}$$

with $u_\theta = u_\theta(x_2, t_2)$ in the previous equation. From Eq. (166), one immediately has

$$S_{\hat{q}_{i_1}, \hat{q}_{i_2}}(x_1, t_1; x_2, t_2) = (h_{i_1})^{\mathrm{dr};\theta} \chi_\theta \delta(x_2 - x_1 - v_\theta^{\mathrm{eff}}(t_2 - t_1))(h_{i_2})^{\mathrm{dr};\theta}. \tag{167}$$

This is precisely what we expect. In particular note that on the same time slice $t_1 = t_2 = t$, the correlator is simply given by the local covariance matrix, i.e., $S_{\hat{q}_{i_1}, \hat{q}_{i_2}}(x_1, t_1; x_2, t_2) = \mathsf{C}_{i_1 i_2} \delta(x_1 - x_2)$ where $\mathsf{C}_{i_1 i_2} := (h_{i_1})^{\mathrm{dr};\theta} \chi_\theta (h_{i_2})^{\mathrm{dr};\theta}$.

Next let us evaluate $S_{\hat{q}_{i_1}, \hat{q}_{i_2}}(x_1, t; x_2, t)$ with respect to an inhomogeneous initial condition. To be more precise, it takes the following form:

$$S_{\hat{q}_{i_1}, \hat{q}_{i_2}}(x_1, t; x_2, t) = \mathsf{C}_{i_1 i_2}(x_1, t)\delta(x_1 - x_2) + E_{i_1 i_2}(x_1, x_2; t), \tag{168}$$

where $\mathsf{C}_{i_1 i_2}(x_1, t)$ is the local covariance matrix, while $E_{i_1 i_2}(x_1, x_2; t)$ is a function that goes to zero when $|x_1 - x_2| \gg 1$ and it accounts for Euler-scaled long-range correlations. The exact expression of $E_{i_1 i_2}$ (see again Appendix H.1 for the details of the calculation) can be obtained by first solving Eq. (229) self-consistently to obtain $\partial_\lambda \epsilon^\theta(x_2, t)\big|_{\lambda=0}$ and plugging it into (161).

The existence of the long-range correlations can be in fact already inferred in the initial condition (164). Namely, it is readily seen that the first term in the bracket amounts to the local equilibrium correlator, while the second term gives long-range correlations. A physical interpretation of such long-range correlations is clear: two normal modes, which can be identified with two particle-hole excitations [83], retain memories of their scattering, as their trajectories would be severely affected by the density landscape around the position where the scattering took place. The scattering picture also makes it evident that the following three conditions are needed for long-range correlations to be supported: interaction, inhomogeneity, and multiple conservation laws. Indeed, were the system to have just a single conservation law, the rotation matrix $R$ would be trivialised, making the long-range contribution in (164) vanish. The first two conditions also ensure that $\partial_{x_1} h^{\mathrm{dr};\theta}(x_1, t_1)$ has a non-zero value at $\lambda = 0$, which implies that it contributes in $\partial_\lambda \epsilon^\theta(x_2, t_1)\big|_{\lambda=0}$.

In order to verify our predictions, we computed the dynamical correlation functions $S_{\hat{q}_0, \hat{q}_0}(x, t; 0, t)$ and $S_{\hat{q}_0, \hat{q}_0}(x, t; -x, t)$ for the hard-rod model and we compared the results with molecular dynamics simulations. In particular, we looked into two initial conditions: two-modes bump-release and the partitioning protocol. In the latter, we implement the very same inverse temperature partitioning initial condition that we used in Subsec. 5.4 for the calculation of the second cumulant $c_2^{\mathrm{part}}$. It is worth to emphasize that in this case, one has a continuum of normal modes since the velocity distribution is a Gaussian with variance given by the inverse temperature. In the former, instead, the rods are released at time $t = 0$ from the Gaussian density bump profile (sketched in the lower band of Fig. 1(a) with $\hat{q}_0$ the particle density)

$$\langle \hat{q}_0(x, 0) \rangle_\ell = \frac{1 + 3e^{-(x/\ell)^2}}{3 + 3e^{-(x/\ell)^2}} \in [1/3, 2/3]. \tag{169}$$

We consider the case where rods' velocities can only take two values $v = \pm 1$ with the same probability. In this case, therefore, the velocity distribution is supported on two delta functions and one has a discrete set of velocities and, consequently, normal modes. This choice of the velocity distribution is not only a drastic simplification for analytic computations, but it also makes the existence of long-range correlations more evident. According to the aforementioned scattering interpretation of long-range correlations, the presence of a continuum set of normal modes would, indeed, cause correlations to spread among all the normal modes

through scattering events among all the rods' velocities. This is expected to make long-range correlations small and barely numerically detectable. In the presence of two normal modes only, on the contrary, scattering events necessarily concern rods with the opposite velocities $+v$ and $-v$ and the long-range correlations between the two associated modes are enhanced. This makes also convenient to compute correlations numerically by reducing the source of statistical error.

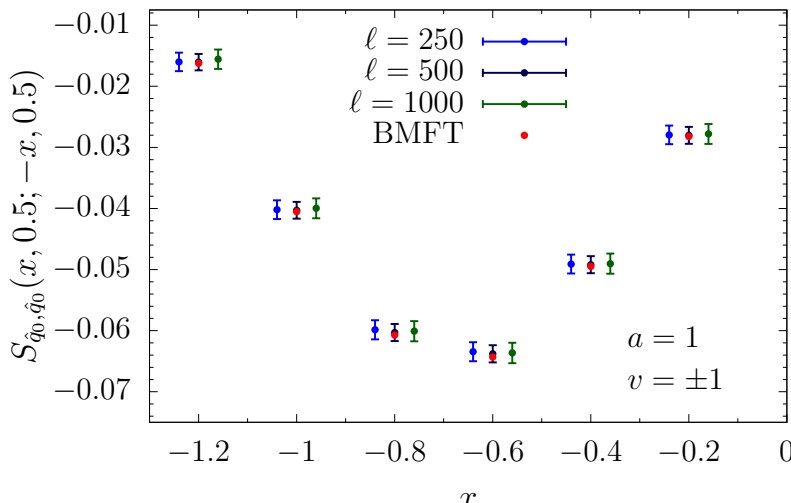

Figure 4: **Connected correlations from a bump release of the hard-rod gas.** The figure shows the rod-density equal-time connected correlation function $S_{\hat{q}_0,\hat{q}_0}(x,t;-x,t)$ evaluated at the macroscopic time $t = 0.5$ as a function of the macroscopic space coordinate $x$. The initial state is the Gaussian density bump in Eq. (169). Also in this case, the numerical results clearly show the existence of long-range correlations, also in this case of order $10^{-2}$, with an evident collapse of the numerical data with respect to the macroscopic length scale $\ell = 250, 500$ and $1000$. The theoretical prediction from BMFT (red points) excellently predict the long-range correlations values, the discrepancies with the numerical data being within the uncertainties' bars. The parameters used in the numerical simulations are rod length $a = 1$ and two possible and equally likely values of the rods' velocities $v = \pm 1$. One has $M = 9.216 \cdot 10^8, 2.1504 \cdot 10^9$ and $1.536 \cdot 10^9$ independent statistical samples for the simulations at the scales $\ell = 250, 500$ and $1000$, respectively. We shifted the data corresponding to the scales $\ell = 250$ and $1000$ by $\pm 0.04$ for the sake of illustration purposes.

In Fig. 4, we report the comparison between the BMFT prediction in Eqs. (132)-(138) and the hard-rod simulations for the correlator $S_{\hat{q}_0,\hat{q}_0}(x,t;-x,t)$. The analysis for the correlator $S_{\hat{q}_0,\hat{q}_0}(x,t;0,t)$ from the same initial state (169) is reported in Fig. (1) of the companion manuscript [24]. These are the first results showing the existence of long-range Euler-scaled correlations in integrable models. We report numerical data for three different scales $\ell = 250, 500$ and $1000$. The simulations, as in the case discussed in Subsec. 5.4 for $c_2^{\text{part}}$, are done in infinite volume, with the rods initially distributed in a symmetric interval $[-L_{\text{size}}, L_{\text{size}}]$ around the origin, with $L_{\text{size}} = 10\ell$. The number $N$ of rods used in the simulations is therefore fixed by the initial density and $L_{\text{size}}$ as $N = \int_{-L_{\text{size}}}^{L_{\text{size}}} dx \langle \hat{q}_0(x,0) \rangle_\ell$. In particular, we have $N = 2700, 5000$ and $10^4$ rods for the simulations at the scales $\ell = 250, 500$ and $1000$,

respectively. The deterministic hard rod evolution from this fluctuating initial condition is then implemented. Equal-time correlation functions are obtained by performing the fluid-cell averaging as per Eq. (35), with the fluid cell length $L = 0.05\ell$, i.e., upon taking the Euler-scaling limit. The numerical estimate for the correlator is eventually obtained by averaging over a large number $M$ of independent realizations of the initial rods' configuration. A number $M = 1.2288 \cdot 10^9, 2.1504 \cdot 10^9$ and $2.1504 \cdot 10^9$ of independent statistical samples has been taken in Fig. (1) of Ref. [24] for the data at the scales $\ell = 250, 500$ and $1000$, respectively. In the case of Fig. 4, the number of samples taken is reported in the corresponding caption. The collapse of the data as a function of $\ell$ is convincing, with the tiny differences among the different scales fully within the uncertainties' bars. This result remarkably confirms, at the numerical level, the existence of Euler-scale correlations, i.e., correlations developing over times $t$ and space regions $x$ proportional to $\ell$, with an amplitude decaying as $\ell^{-1}$. Moreover, the BMFT result in Eqs. (132)-(138) particularized to the hard-rod model predicts the value of such correlations with a very good agreement, despite of the small $10^{-2}$ order of magnitude of the long-range correlations. The largest difference between the BMFT prediction and the numerical result is, indeed, observable on the third decimal digit and it is always fully within the statistical uncertainties' bars. The latter are computed according to the same method used in Subsec. 5.4 for $c_2^{\text{part}}$ (see the Appendix I for the details). Our numerical analysis thereby firmly corroborate the existence of long-range correlations in interacting inhomogeneous fluids, and at the same time, the power of the BMFT in quantitatively predicting this effect.

In the case of the partitioning protocol, the initial step is chosen in the very same way as for the calculation of $c_2^{\text{part}}$ in Fig. 3. In particular, we consider the case of a step initial inverse temperature profile $\beta_{\text{ini}}^i(x)$ in Eq. (85) (again with the single multiplier associated to the energy conserved charge being non-zero). The initial density profile therefore shows a discontinuity at $x = 0$ as well, with $\rho(\beta_L)$ ($\rho(\beta_R)$) the density for $x < 0$ ($x > 0$). Since the partitioning protocol state is scale invariant, the numerical data for $S_{\hat{q}_0, \hat{q}_0}(x, t; 0, t)$ and $S_{\hat{q}_0, \hat{q}_0}(x, t; -x, t)$ are obtained by performing the fluid-cell averaging in Eq. (35) with the scale $\ell = 300$ taken as the microscopic-simulation time. We have specifically computed both the equal-time correlators $S_{\hat{q}_0, \hat{q}_0}(x, 1; 0, 1)$ and $S_{\hat{q}_0, \hat{q}_0}(x, 1; -x, 1)$ at the macroscopic time $t = 1$ for $x \in [-3, -1]$, for the same set of parameters in Fig. 3. We average, in this case, over $M = 10^9$ independent realizations of the initial condition. Also in this case, we numerically observe the existence of long-range correlations of order $10^{-2}$. This shows that the presence of long-range correlations, caused by the emission of normal modes in the past, is a robust phenomena present in different physically relevant out of equilibrium scenarios. Crucially, the BMFT prediction of the correlator, evaluated from the specialization of Eqs. (132)-(138) to the partitioning initial state (see Sec. H.2 of the Appendix), gives values of order $10^{-6}$. These values are no way compatible with the result from the numerical analysis, whose statistical uncertainties are of order $10^{-3}$. The physical interpretation of this discrepancies is clear as the partitioning protocol state does not show large-scale variations. Microscopic correlations from the initial sharp inhomogeneity are therefore produced at early times in addition to the Euler-scaled hydrodynamic correlations discussed so far. These microscopic contributions to correlations, since are caused by normal modes emitted in the past, do not vanish at long times, despite the partitioning state rapidly relaxing to a large-wavelength state. This is in stark contrast to the cumulant analysis of Subsec. 5.4, where we found that the initial microscopic contributions vanish at long times. Our results further show that the microscopic, short-time, contribution to the correlations is predominant, causing the numerically observed values to be much larger than the BMFT prediction.

In all the cases where the initial state displays, on the contrary, long wavelengths variations (such as in the bump release protocol discussed previously), microscopic early time contributions to the correlations are suppressed. Only Euler-scaled, universal, correlations are therefore present. The latter are quantitatively predicted by our BMFT.

# 6 Conclusion and Outlook

In this manuscript we thoroughly discussed and further elaborated the results of the companion work [24]. We extended the conventional diffusive MFT to describe the physics controlled by rare but significant fluctuations, such as large deviations and the dynamical correlation functions, at the Euler scale of hydrodynamics, in many-body systems supporting ballistic transport. The fundamental principle of the BMFT is the *local relaxation of fluctuations* introduced in Sec. 3: fluctuations of mesoscopic observables (averages over fluid cells) are encoded as classical random variables, functions of the fluctuating mesoscopic conserved densities $\underline{q}(\cdot, \cdot)$ on space-time $\mathbb{S} = \mathbb{R} \times [0, T]$. Their functional form is entirely fixed by the maximal entropy states (Gibbs and generalised Gibbs ensembles) of the model. This is a version of the Boltzmann-Gibbs principle of projection of local observables onto local densities, but expressed in full generality in the context of the Euler-scale physics. Local relaxation of fluctuations, combined with conservation laws of the microscopic model, implies that the fluctuations in space-time originate from those of the initial condition, as described by thermodynamics; initial fluctuations are simply time-evolved in a deterministic way according to the Euler hydrodynamic equation. This is a generalisation of the principle of local relaxation of averages, which is the cornerstone of Euler hydrodynamics, to that of rare fluctuations. From this principle, the measure of the space-time configuration of the fluctuating densities in Eq. (46) is derived. Equation (46) is the basis of all the BMFT predictions.

The BMFT is similar in spirit to the conventional, diffusive MFT, in that it is a large-deviation theory for space-time configurations based on an action formalism on conserved densities. However, one cannot be obtained from the other. Instead, we conjecture in Subsec. 3.3 a theory that describes both the ballistic and diffusive scale, where, in addition to the ballistic scale, noise contributions are included to the currents at smaller scales. The "zero-noise" limit of this theory, the limit where one concentrates on the ballistic scale, recovers the BMFT; and the special case where no ballistic transport is present gives back the conventional MFT.

The two main physical predictions of the BMFT that we focused on in this paper are discussed in Sec. 4: the SCGF of the time-integrated current $F(\lambda, T)$ (14) in Subsecs. 4.1-4.3, and the Euler scale correlation function $S_{\hat{q}_{i_1}, \hat{q}_{i_2}}(x_1, t_1; x_2, t_2)$ (22) in Subsecs. 4.4-4.7, for arbitrary weakly-inhomogeneous initial state of the form (12). There are two main observations.

One is that the existence of a time-reversal symmetry of the Euler hydrodynamics imply, from the natural symmetry of the BMFT equations, the Gallavotti-Cohen fluctuation theorem for transport in the partitioning protocol, as shown in Subsec. 4.3. This is one of the most universal out-of-equilibrium properties of many-body systems.

Another one, explained in Subsec. 4.6 and in the companion manuscript [24], is the existence of certain types of *long-range correlations* out of equilibrium. We show that for the system to develop such long-range correlations, three conditions must be met: the system must be interacting (nonlinear Euler hydrodynamics), it must admit more than one conservation

laws (more than one hydrodynamic velocity), and its initial condition must be inhomogeneous (for instance, presenting large-wavelength variations). In particular, long-range two-point correlations offer a clear way of distinguishing between interacting and non-interacting models without the need for evaluating higher-point correlation or response functions [22] or going to the diffusive scale [97].

The BMFT also gives access to more subtle information, such as the large-deviation theory of fluctuations within fluid cells. It shows that, in certain situations, such fluctuations are *not given* by those of the local maximal entropy state that corresponds to the values of local densities solving the Euler hydrodynamics. We predict the scaled covariance for TASEP, but further investigations would be necessary.

Importantly, only the Euler hydrodynamic data of the model – in particular the flux Jacobian – is needed for the BMFT, and the predictions apply to any many-body systems, quantum or classical, deterministic or stochastic (so long as the hydrodynamics at the Euler scale is nontrivial). We emphasise in particular that *even in stochastic systems, the theory for the dynamics of fluctuations at the ballistic scale does not involve noise.* In these cases, the noise on the microscopic dynamics serves the principle of local relaxation of fluctuations, and it stays true that fluctuations at the ballistic scale are obtained by deterministic evolution (via the emerging Euler equation) of the initial large-scale fluctuations.

Having these general predictions from the BMFT at our disposal, we focused on integrable systems, where a much more elaborate analysis can be carried out than in non-integrable systems. We relied heavily on the machinery of GHD.

We first computed cumulants associated with current fluctuations both in homogeneous and step initial conditions, all of which perfectly agreed with hard-rods simulations.

We next computed the Euler-scale dynamical correlation function from general long-wavelength initial condition. We specialised these to two inhomogeneous initial conditions: the bump release and the partitioning protocol; the former is expected to be described by the BMFT as it has smooth initial spatial variations, while the latter is generically not, as with rough initial spatial variations, additional correlations are created by microscopic processes, which are not universally set by Euler hydrodynamics. This turned out to indeed be the case. For the bump release, we observed a very good agreement in Fig. 4 between the analytic BMFT result and the numerical simulation. In the case of the partitioning protocol, instead, disagreement is seen, with more correlations in the simulation; these are interpreted as additional correlations that build up during the transient dynamics from the initial step condition.

There are numerous avenues that can be pursued by applying the BMFT. One immediate direction is to put the idea of local relaxation of fluctuations on a mathematically more rigorous ground. As mentioned, this idea is closely related to the Boltzmann-Gibbs principle, which in general states that, at an appropriately large scale, any fluctuating fields that are functions of space-time can be replaced by some functional of fluctuating density fields [26]. This statement has been proved for stochastic systems, see e.g., [27], but it would be strongly desired to establish it also in Hamiltonian systems, and in the generality of hydrodynamics with many conservation laws.

Another direction, along the line of further checking the validity the BMFT, is to compute the whole large deviation function $F(\lambda, T)$ and compare it with numerical simulations of classical systems, such as the hard rods. This would, moreover, allow us to verify the equivalence between the BMFT and the homogeneous BFT beyond the perturbative argument in Subsec. 4.2.

It would also be satisfying to reproduce our predictions for integrable systems by focusing on analytically tractable models such as the box-ball system [98,99] (see in particular [43] for the exact computations of the cumulants). Concerning non-integrable systems, such as the TASEP and anhamornic chains, wherein hyperbolicity of the hydrodynamic system generically amounts to shocks, it is of paramount importance to investigate large deviations and dynamical correlations using the BMFT. Within this perspective, it would be illuminating to discern any possible qualitative differences between integrable and non-integrable systems. It would be also interesting to work out a solvable stochastic exclusion process with multiple species, e.g., Arndt-Heinzl-Rittenberg model [100], to see if our predictions on the long-range correlations can be microscopically confirmed.

The BMFT as formulated here can also be extended to describe ballistic large deviations more generally, such as under time evolution with long-wavelength, low-frequency variations of external fields and coupling parameters [101–103], and with initial conditions that already include long-range correlations such as those appearing after quantum quenches [91,104,105]. As suggested in [20], ballistic fluctuations are connected to objects called "twist fields", which have many interesting applications, including for the study of entanglement [106]; this is another area where the BMFT may offer new insight.

It is also tempting to go beyond the BMFT by including the subleading corrections in the measure as in Subsec. 3.3, in particular (63). Importantly, the onset to the stationary value of $c_2^{\text{part}}$ observed in Fig. 3 would be captured by the BMFT with diffusive corrections. More interestingly, we should study the cases where subleading corrections to the ballistic transport are superdiffusive. In such a situation one can try to incorporate these corrections perturbatively, which amounts to the inclusion of fluctuations around the saddle point. Such fluctuations are generally controlled by the determinant of the Hessian of the action, which tantalisingly suggests a connection with determinantal structures that are often found in systems that belong to the KPZ universality class [107]. We also assert that the BMFT with diffusive corrections applied to spin transport in the gapped and isotropic XXZ spin-1/2 chain might shed some light on the recently discovered apparent breakdown of large-deviation principles in the model [108–110].

Another promising direction is to generalise the idea of the (B)MFT to study the dynamics that is strongly influenced by quantum fluctuations. As we have seen, the underlying idea of the BMFT is the propagation of initial fluctuations, which are, in the present case, dominated by the thermal ones; quantum fluctuations are in general controlled by different scales. We would need to combine the ideas of quantum GHD [111] with the (B)MFT path-integral formalism.

Finally, some of the predictions from the BMFT, e.g., the exact cumulant (158), could be observed in the state-of-the-art cold atom experiments. Indeed, in a recent experiment [112], the full counting statistics of spin transport in the isotropic XXZ spin-1/2 chain was experimentally studied using a quantum gas microscope.

# Acknowledgements

We are grateful to Bruno Bertini, Olalla Castro-Alvaredo, Jacopo De Nardis, Fabian Essler, Tony Jin, Pierre Le Doussal, Adam Nahum, Tibor Rakovszky, Paola Ruggiero, and Herbert Spohn for illuminating discussions.

**Funding information** The work of BD was supported by the Engineering and Physical Sciences Research Council (EPSRC) under grants EP/W000458/1 and EP/W010194/1. GP acknowledges support from the Alexander von Humboldt Foundation through a Humboldt research fellowship for postdoctoral researchers. The work of TS has been supported by JSPS KAKENHI Grant Nos. JP16H06338, JP18H03672, JP19K03665, JP21H04432, JP22H01143. BD, TS and TY acknowledge hospitality and support from the Galileo Galilei Institute, and from the scientific program on "Randomness, Integrability, and Universality".

## A    The measure for the initial state

In this section we derive the generating function Eq. (44) for the Euler-scale equal-time correlations of conserved densities for the initial state (12) (generalising (16) to $x$-dependent parameter $\lambda(x)$). We write

$$\left\langle \exp \int_{\mathbb{R}} \mathrm{d}x \, \lambda^i(x/\ell)\hat{q}_i(x) \right\rangle_\ell = \frac{\mu\Big[ \exp \ell \int_{\mathbb{R}} \mathrm{d}x \, (\lambda^i(x) - \beta^i_{\mathrm{ini}}(x))\hat{q}_i(\ell x) \Big]}{\mu\Big[ \exp -\ell \int_{\mathbb{R}} \mathrm{d}x \, \beta^i_{\mathrm{ini}}(x)\hat{q}_i(\ell x) \Big]}, \tag{170}$$

where all charge densities are evaluated at time 0. We then apply $\delta\big(\ell^{-1}\log\bullet\big)/\delta\lambda^i(x)$ on this expression, in the limit $\ell \to \infty$:

$$\frac{\delta}{\delta\lambda^i(x)} \lim_{\ell\to\infty} \ell^{-1} \log \left\langle \exp \int_{\mathbb{R}} \mathrm{d}x \, \lambda^i(x/\ell)\hat{q}_i(x) \right\rangle_\ell = \langle \hat{q}_i(\ell x)\rangle_\ell = \mathsf{q}_i[\underline{\beta}_{\mathrm{ini}}(x) - \underline{\lambda}(x)]. \tag{171}$$

In the quantum case, this last calculation must be argued for more carefully because of non-vanishing commutation relations. We write for instance $\ell \int_{\mathbb{R}} \mathrm{d}x \, \beta^i(x)\hat{q}_i(\ell x) = \ell \int_{\mathbb{R}} \mathrm{d}x \, \beta^i(x)\overline{q_i}(\ell x) = L\sum_{k\in\mathbb{Z}} \beta^i(x_k)\overline{q_i}(\ell x_k)$ where the sum is over the fluid cell at positions $x_k = kL/\ell$ and of size $L/\ell$ (in macroscopic coordinates), and here we take fluid cell averaging in space only. Because the fluid cells are large, by locality of the densities, commutators vanish. For neighbouring cells, the only nontrivial case, the calculation is as follows

$$
\begin{aligned}
[\overline{q_i}(\ell x_k), \overline{q_j}(\ell x_{k+1})] &= \frac{1}{L^2} \int_0^L \mathrm{d}y \int_0^L \mathrm{d}z \, [\hat{q}_i(\ell x_k + y), \hat{q}_j(\ell x_k + z + L)] \\
&= \frac{1}{L^2} \int_{L-\ell_{\mathrm{micro}}}^L \mathrm{d}y \int_0^{\ell_{\mathrm{micro}}} \mathrm{d}z \, [\hat{q}_i(\ell x_k + y), \hat{q}_j(\ell x_k + z + L)],
\end{aligned}
\tag{172}
$$

where we used the fact that commutators of local observables are nonzero only at microscopic distances. Therefore $||[\overline{q_i}(\ell x_k), \overline{q_i}(\ell x_{k+1})]|| \leq 2\ell_{\mathrm{micro}}^2/L^2 \, ||\hat{q}_i|| \, ||\hat{q}_j|| \to 0$. The variables $\overline{q_i}(\ell x_k)$ are commuting macroscopic variables in the sense introduced by von Neumann in the context of his quantum ergodicity theorem (see, e.g., [80]).

From (171) and the condition at $\lambda(x) = 0$, and from the definition of the free energy $f[\underline{\beta}]$, we deduce that

$$\left\langle \exp \int_{\mathbb{R}} \mathrm{d}x \, \lambda^i(x/\ell)\hat{q}_i(x) \right\rangle_\ell \asymp \exp \ell \left( \int_{\mathbb{R}} \mathrm{d}x \, \big(f[\underline{\beta}_{\mathrm{ini}}(x)] - f[\underline{\beta}_{\mathrm{ini}}(x) - \underline{\lambda}(x)]\big) \right). \tag{173}$$

This shows (44).

# B    Thermodynamics of hard-rods system

The hard-rods model is a classical many-body system, describing a gas of identical rods (we set the rods' mass to 1) with length $a$. The rods propagate freely until they experience elastic pairwise collisions, where the velocities get exchanged. The system has an infinite number of conservation laws labelled by the velocities $\theta$ and hence is integrable. The hard-rods system occupies a vital position amongst integrable systems from the viewpoint of GHD. This is partly because one of the key insights offered by GHD is that, on the Euler scale, a fluid of integrable systems can be thought of as a gas of tracer particles of hard-rods (with velocity-dependent jumps) [28,76]. Here, velocity tracers are quasi-particles assigned to each rod, which propagate along straight line trajectories (in the space-time diagram) interspersed with jumps of length $a$ at each collision. The single particle eigenvalue of the energy, $E_\theta$, momentum, $p_\theta$, and particle number, $N_\theta$, are the ones of a classical Galilean particle

$$E_\theta = \frac{\theta^2}{2}, \quad p_\theta = \theta, \quad N_\theta = 1. \tag{174}$$

The scattering phase shift equals (we follow the notation convention of Ref. [28])

$$T^\theta_{\ \phi} = -a, \tag{175}$$

which amounts to the following integral equation for the pseudo-energy (124)

$$\epsilon^\theta = \beta^\theta + a \int_\mathbb{R} \frac{\mathrm{d}\phi}{2\pi} \exp\left(-\epsilon^\phi\right), \tag{176}$$

where we used $n^\theta = e^{-\epsilon^\theta}/(2\pi)$ in hard-rods. Thanks to this particularly simple form of the phase shift, the description of hard-rods is substantially simplified. For instance, the dressing operation simply gives, for any $\mathsf{a}_\theta$,

$$\mathsf{a}^{\mathrm{dr};\theta} = \mathsf{a}^\theta - a(1-a\rho) \int_\mathbb{R} \mathrm{d}\phi\, n_\phi \mathsf{a}^\phi, \tag{177}$$

where the density of rods satisfies $\rho/\rho^{\mathrm{tot}} = \int_\mathbb{R} \mathrm{d}\phi\, n^\phi$ with $\rho^{\mathrm{tot}} := 1 - a\rho$. Accordingly, the effective velocity reads

$$v^{\mathrm{eff}}_\theta = \frac{\theta - a\mathsf{j}}{\rho^{\mathrm{tot}}}, \tag{178}$$

where $\mathsf{j} := \rho^{\mathrm{tot}} \int_\mathbb{R} \mathrm{d}\phi\, \phi n^\phi$. A simplification also occurs for the characteristics $\mathcal{U}^\theta(x,t;s)$, which is now determined by

$$\int_{-\infty}^{\mathcal{U}^\theta(x,t;s)} \mathrm{d}y\, \rho^{\mathrm{tot}}(y,s) + (\theta - a\mathsf{j}_-)(t-s) = \int_{-\infty}^{x} \mathrm{d}y\, \rho^{\mathrm{tot}}(y,t), \tag{179}$$

where $\mathsf{j}_- = \lim_{x\to-\infty} \mathsf{j}(x,0)$.

# C    Hydrodynamic response theory

In this section we shall review the basics of the hydrodynamic response theory, and in particular articulate under which circumstances it is valid. The underlying idea of the hydrodynamic

response theory is that an initial local entropy-maximised state (12) propagates in time while keeping its form intact, and that perturbations of this state generate Euler-scale correlation functions.

Suppose the system is initially in a local entropy-maximised state labelled by Lagrange multipliers $\beta^j(x, 0) = \beta_{\text{ini}}^j(x)$. We then let the system evolve in time. According to Euler hydrodynamics, $\beta^j(x, t)$ satisfy (11). In order to access dynamical correlation functions, the insertion of a mesoscopic (fluid-cell averaged) conserved density $\overline{q_{i_1}}(x_1, t_1)$ is obtained by performing a small perturbation of the state at the space-time $(x_1, t_1)$: $\beta^{i_1}(x_1, t_1) \mapsto \beta^{i_1}(x_1, t_1) + \delta\beta^{i_1}(x_1, t_1)$. The way the Euler hydrodynamic solution changes for a (possibly different) conserved density $\mathsf{q}_{i_2}(x_2, t_2)$, at a later time $t_2 > t_1$, gives the correlation between $\overline{q_{i_1}}(x_1, t_1)$ and $\overline{q_{i_2}}(x_2, t_2)$. A similar principle holds for higher-point functions: for the $n$-point Euler-scale function $S_{\hat{q}_{i_2}, \ldots, \hat{q}_{i_n}}(x_2, t_2; \cdots; x_n, t_n)$, one looks at the way the Euler-scale $(n-1)$-point function changes under the perturbation $\beta^{i_1}(x_1, t_1) \mapsto \beta^{i_1}(x_1, t_1) + \delta\beta^{i_1}(x_1, t_1)$; this is referred to as nonlinear response as, by induction, it requires focusing on the $(n-1)^{\text{th}}$ power of perturbations of the original hydrodynamic solution.

Thus, the two-point function $S_{\hat{q}_{i_1}, \hat{q}_{i_2}}(x_1, t_1; x_2, t_2)$, according to the hydrodynamic response theory, is given by

$$S_{\hat{q}_{i_1}, \hat{q}_{i_2}}(x_1, t_1; x_2, t_2) = -\frac{\delta\mathsf{q}_{i_2}(x_2, t_2)}{\delta\beta^{i_1}(x_1, t_1)}. \tag{180}$$

This means that $\mathsf{q}_{i_2}(x_2, t_2)$ is evaluated in the state which is, at $t_1$, described by

$$\beta^i(x, t_1)\Big|_{\text{linear response}} = \beta^i(x, t_1) - \lambda\delta_{i_1}{}^i\delta(x - x_1), \tag{181}$$

and that the derivative is with respect to $\lambda$, after which $\lambda = 0$ is taken. This idea is generalised to perturbations of the hydrodynamic solution $\mathsf{o}(x_2, t_2)$ for an arbitrary observable $\hat{o}(x_2, t_2)$, by considering $\mathsf{o}(x_2, t_2)$ as a function of $\mathsf{q}_i(x_2, t_2)$'s and using the chain rule for differentiation. The idea is further generalised, somewhat formally, by assuming that for every observable $\hat{o}(x, t)$, there is an associated Lagrange multiplier $\beta^{\hat{o}}$ which we can perturb in order to insert that observable.

These response principles naturally lead to (26) and (28). But in addition, as mentioned, implicit in response theory is that the form (12) stays valid for all times – this is how one can justify modifying the parameter $\beta^{i_1}(x_1, t_1)$, at time $t_1$, in order to insert $\overline{q_{i_1}}(x_1, t_1)$ for instance. And thus, in particular, as a direct consequence of (181), the initial condition for (28) is the delta-function form $S_{\hat{q}_i, \hat{q}_j}(x_1, t_1; x_2, t_1) = \mathsf{C}_{ij}(x_1, t_1)\delta(x_1 - x_2)$, which in general disagrees with the long-range correlations found in the BMFT if $t_1 > 0$.

We will now see how the linear-response principles are in fact in disagreement with the BMFT, even though (26) and (28) are correct.

In order to see this, we recall that in the BMFT, we consider, much like in linear response theory, Lagrange multipliers $\beta^i(x, t)$ that evolve according to the Euler equation. The insertion of a conserved density $\overline{q_{i_1}}(x_1, t_1)$ is also obtained by modifying the state appropriately. However, we point out that the modification is *not given by (181)*. Indeed, the result (110) states that the Lagrange multipliers at time $t_1$ have the form

$$\beta^i(x, t_1)\Big|_{\text{BMFT}} = \big(U_\lambda(t_1, 0)\underline{\beta_{\text{ini}}}\big)^i(x) - \lambda\delta_{i_1}{}^i\delta(x - x_1). \tag{182}$$

Recall that $U_\lambda(t, t')$ is the linear evolution operator that transports a quantity along the $\lambda$-dependent fluid of the BMFT, Eqs. (107) and (108). Note also that at $\lambda = 0$ the BMFT

simply gives for $\beta^i(x,t)$ the Euler hydrodynamics from the initial state $\beta^i_{\mathrm{ini}}(x)$. Thus we can write

$$\beta^i(x,t_1) = \left(U_0(t_1,0)\underline{\beta_{\mathrm{ini}}}\right)^i(x). \tag{183}$$

We can now see clearly the difference between the linear response (181) and the BMFT (182): it lies in the first term, which is $\left(U_0(t_1,0)\underline{\beta_{\mathrm{ini}}}\right)^i(x)$ in the linear response theory, and $\left(U_\lambda(t_1,0)\underline{\beta_{\mathrm{ini}}}\right)^i(x)$ in the BMFT.

Therefore, linear response and the BMFT agree for the Euler-scale two-point functions only in non-interacting models (in which case $U_\lambda(t,t')$ is independent of $\lambda$), at $t_1 = 0$ (as $U_\lambda(0,0) = 1$), or in homogeneous states (as $U_\lambda(t_1,0)\underline{\beta_{\mathrm{ini}}} = \underline{\beta_{\mathrm{ini}}}$). This further suggests that nonlinear response theory is incorrect (higher-point functions are not given by linear response principles) even in homogeneous states, if the model is interacting and at least two of the times are greater than zero. Indeed, as explained above, nonlinear response theory is obtained from considering two-point functions in inhomogeneous, perturbed states.

# D   The inhomogeneous ballistic fluctuation theory

For ease of notation, here and in the rest of the appendices, we will occasionally use "dr" and "eff" as subscripts. We will use upper and lower indices for $u_\theta$, $r_\theta$, $T$, and $v^{\mathrm{eff}}$ liberally too.

The inhomogeneous BFT, developed in Ref. [23], was devised to generalise the BFT to long-wavelength inhomogeneous initial states, and built on the linear response theory developed in [19] for integrable systems, which in particular assumes (180). The inhomogeneous BFT turns out to predict that there are in general two contributions to Euler-scaled correlations $S_{\hat{q}_i,\hat{q}_j}(x,t;y,t')$ (22) and cumulants $c_n$ (24) determined by a direct and an indirect propagator. In particular, focusing on the Euler-scaled second cumulant $c_2$ defined in Eq. (24) one has

$$c_2(T) = \frac{2}{T}\int_0^T \mathrm{d}t_2 \int_0^{t_2} \mathrm{d}t_1 \left(\Gamma_{(0,t_1)\to(0,t_2)}\right)^\theta_\phi v^{\mathrm{eff},\phi}(0,t_1) h^{\mathrm{dr},\phi}_{i_*}(0,t_1)\chi_\theta(0,t_2) v^{\mathrm{eff}}_\theta(0,t_2) h^{\mathrm{dr}}_{i_*,\theta}(0,t_2), \tag{184}$$

where we symmetrized the two-point function as it is invariant under the exchange $t_1 \leftrightarrow t_2$. The propagator $\Gamma$ can be split, as anticipated, into the direct and indirect part as

$$\left(\Gamma_{(y,\tau')\to(x,\tau)}\right)^\theta_\phi = \delta\left(y - \mathcal{U}_\theta(x,\tau,\tau')\right)\delta^\theta_\phi + \left(\Delta_{(y,\tau')\to(x,\tau)}\right)^\theta_\phi. \tag{185}$$

The first term on the right hand side is the direct propagator and it depends on the single rapidity $\theta$ whose associated normal mode propagates between the two space-time points $(y,t')$ and $(x,t)$ correlations refer to, as sketched in Fig.1b. This contribution is present even in the simpler case of homogeneous GGE states. The indirect propagator $\Delta_{(y,\tau')\to(x,\tau)}$, instead, necessarily requires an inhomogeneous fluid background and interactions among normal modes. The indirect propagator encodes the perturbation of the trajectory of the normal mode with rapidity $\theta$ due to the interaction with normal modes with a different rapidity $\phi$, not necessarily connecting the space-time points $(y,t')$ and $(x,t)$. As a consequence, the indirect propagator depends on all the rapidities.

Before giving the expression for the indirect propagator $\Delta_{(y,\tau')\to(x,\tau)}$, we consider the specialization of Eq. (184) to the partitioning protocol, discussed in Sec. 5.3. In this case,

as explained in the main text after Eq. (130), the state depends only the ray and therefore equation (184) can be rewritten as

$$c_2^{\text{part}} = \frac{2}{T} \int_0^T \mathrm{d}t_2 \int_0^{t_2} \mathrm{d}t_1 \left(\Gamma_{(0,t_1)\to(0,t_2)}\right)^\theta{}_\phi v^{\text{eff},\phi} h_{i_*}^{\text{dr},\phi} \chi_\theta v_\theta^{\text{eff}} h_{i_*,\theta}^{\text{dr}}$$
$$= 2 \int_0^1 \mathrm{d}a \, (\Gamma_a)^\theta{}_\phi \, v^{\text{eff},\phi} h_{i_*}^{\text{dr},\phi} \chi_\theta v_\theta^{\text{eff}} h_{i_*,\theta}^{\text{dr}}, \tag{186}$$

where the state-dependent functions reported without space-time arguments are meant henceforth in this Subsection to be evaluated on the ray $\xi = x/t = 0$. In the second equality, we used the scaling property of the propagator $\Gamma$

$$\left(\Gamma_{(0,\alpha\tau')\to(\alpha\tau)}\right)^\theta{}_\phi = \frac{1}{\alpha}\left(\Gamma_{0,\tau'\to 0,\tau}\right)^\theta{}_\phi, \tag{187}$$

which is valid since the partitioning protocol initial state is invariant under space-time rescaling transformations. We further defined $\Gamma_a = \Gamma_{0,a\to 0,1}$ (omitting the space point $x = 0$ for brevity), with $0 < a < t_1/t_2 < 1$. It is immediate to evaluate the contribution $c_2^{\text{part,dir}}$ of the direct propagator in Eq. (185) to the second cumulant

$$c_2^{\text{part,dir}} = 2 \int_0^1 \mathrm{d}a \chi_\theta (v_\theta^{\text{eff}} h_{i_*}^{\text{dr},\theta})^2 \delta\left(\mathcal{U}_\theta(0,1,a)\right) = \chi_\theta \left|v_\theta^{\text{eff}}\right| (h_{i_*}^{\text{dr};\theta})^2, \tag{188}$$

where we used that $\delta\left(\mathcal{U}_\theta(0,1,a)\right) = \delta(a-1)/|v_\theta^{\text{eff}}|$ and the regularization $\Theta(0) = 1/2$ of the Heaviside step function. Equation (188) is readily recognized as (158). In order to conclude to proof of (158), we therefore need to show that the indirect propagator $\Delta$ in Eq. (185) gives zero contribution to $c_2^{\text{part}}$.

This requires more work as the expression of the indirect propagator $\Delta$ is given through an integral equation. We report the latter equation for the specific case of the partitioning protocol (see Refs. [19, 23, 31] for the general discussion) This indirect propagator satisfies

$$(\Delta_{a,\xi})^\theta{}_\phi \, v^{\text{eff},\phi} h_{i_*}^{\text{dr},\phi} =$$
$$2\pi \, a^{\text{eff},\theta} (\mathcal{U}^\theta(\xi,1,a)/a) \left[ [W_{a,\xi} v^{\text{eff},\theta} h_{i_*}^{\text{dr},\theta}] + \int_{-\infty}^{\xi} \mathrm{d}\zeta \left( \rho^{\text{tot},\theta}(\zeta) f^\theta(\zeta) [(\Delta_{a,\zeta})^\theta{}_\phi \, v^{\text{eff},\phi} h_{i_*}^{\text{dr},\phi}] \right)^{*\text{dr}} (\zeta) \right], \tag{189}$$

with the notation $\Delta_{(0,\tau')\to(x,\tau)} = \tau^{-1}\Delta_{(0,a)\to(\xi,1)} \equiv \tau^{-1}\Delta_{(a,\xi)}$ from the scaling property (187). Here and below $\xi, \zeta$ are rays, and $a = t_1/t_2$ as above (more precisely, $a = t_1$ with $t_2 = 1$). In the previous equation, we denoted with $h^{*\text{dr},\theta}(\zeta) = h^{\text{dr},\theta}(\zeta) - h^\theta$, where dressing operation is performed with respect to the state on the ray $\zeta$ for a generic function $h^\theta$ of the rapidity. The operator $W_{a,\xi}$ is defined as

$$[W_{a,\xi} v^{\text{eff},\theta} h_{i_*}^{\text{dr},\theta}] = -\Theta\left(\mathcal{U}^\theta(\xi,1,a)\right)(\rho^{\text{tot},\theta} f^\theta v^{\text{eff},\theta} h_{i_*}^{\text{dr},\theta})^{*\text{dr}}(0)$$
$$+ \int_{-\infty}^{\xi} \mathrm{d}\zeta \left. \frac{\rho^{\text{tot},\gamma}(\zeta) n^\gamma f^\gamma \left(T^{\text{dr}}\right)^{\theta,\gamma}(\zeta) v^{\text{eff},\gamma} h_{i_*}^{\text{dr},\gamma}}{|\partial_\gamma U^\gamma(\zeta,1,a)|} \right|_{\gamma=\theta_*(\zeta,a)}, \tag{190}$$

where $\mathcal{U}^{\theta_*(\zeta,a)}(\zeta,1,a) = 0$ that is $\theta_*(\zeta,a)$ is the rapidity for which the characteristic at time 1 passing by $\zeta$, at time $a$ passes by 0. We assumed monotonicity of the characteristic $\mathcal{U}^\theta$ with

respect to the rapidity variable, but otherwise there is a sum over values of $\theta_*(\zeta, a)$. We have further denoted with $\left(T^{\mathrm{dr}}\right)^{\theta,\phi}(\zeta, \lambda, \gamma)$ (with two upper indices to emphasize that no rapidity integration is performed inside the $\zeta$ integral) the differential scattering kernel dressed with respect to the state $n(\zeta)$. In Eq. (189), we have also introduced the effective acceleration $a_\theta^{\mathrm{eff}}$ [19], which encodes the inhomogeneity of the initial state as

$$
\begin{aligned}
a_\theta^{\mathrm{eff}}(\xi) = \frac{\partial_\xi n_\theta(\xi)}{2\pi\rho_\theta(\xi)f_\theta(\xi)} &= \frac{\delta n_\theta}{2\pi\rho_\theta(\xi)f_\theta(\xi)}\delta(v_\theta^{\mathrm{eff}}(\xi) - \xi)\left(1 - \frac{\partial v_\theta^{\mathrm{eff}}(\xi)}{\partial\xi}\right) \\
&= \frac{\delta n_\theta}{2\pi\rho_\theta(\xi)f_\theta(\xi)}\delta(v_\theta^{\mathrm{eff}}(\xi) - \xi),
\end{aligned}
\tag{191}
$$

in the second step we used Eq. (130) for the state $n_\theta(\xi)$ in the partitioning protocol and $\delta n_\theta = n_{R,\theta} - n_{L,\theta}$. Importantly, in the partitioning protocol, the effective acceleration contains a delta function that enforces the constraint that the ray $\xi$ must be equal to $\xi_\theta^*$ defined as

$$
\xi_\theta^*: \quad v_\theta^{\mathrm{eff}}(\xi_\theta^*) = \xi_\theta^*.
\tag{192}
$$

The ray derivative of the effective velocity evaluated at $\xi^*$ accordingly vanishes

$$
\left.\frac{\partial v_\theta^{\mathrm{eff}}(\xi)}{\partial\xi}\right|_{\xi = \xi_*^*} = 0, \quad \text{since} \quad \partial_\xi v_\theta^{\mathrm{eff}}(\xi)\rho_\theta^{\mathrm{tot}}(\xi) = (\xi - v_\theta^{\mathrm{eff}}(\xi))\partial_\xi\rho_\theta^{\mathrm{tot}}(\xi).
\tag{193}
$$

The last equation follows from the fact that the total density of states $\rho_\theta^{\mathrm{tot}}$ (assumed to be strictly positive) satisfies the same GHD equation as $\rho_\theta$ (written in terms of the ray $\xi = x/t$ coordinate). We used this property in the third equality in Eq. (191).

One can see that the effective acceleration, according to its definition, vanishes when the state $n$ is homogeneous. In the latter case, therefore, the indirect propagator $\Delta$ vanishes and cumulants are solely determined by the direct contribution of Eq. (185). In this case, we, indeed, recover the prediction from the homogeneous BFT theory discussed in the text. For the second cumulant, in particular, one has (188) in agreement with Eq. (128). We now, however, show that for the partitioning protocol initial inhomogeneous state the structure of the effective acceleration in (191) allows to show that the indirect propagator vanishes for $c_2(T)$. Since the effective accelaration $a_\theta^{\mathrm{eff}}$ has to be computed in Eq. (189) along the characteristic curve $\mathcal{U}^\theta(\xi, 1, a)$, we exploit the identity

$$
\begin{aligned}
\frac{\partial\mathcal{U}^\theta(x, \tau, \tau')}{\partial\tau'} &= \frac{\partial(\tau'\mathcal{U}^\theta(x/\tau', \tau/\tau', 1))}{\partial\tau'} \\
&= \mathcal{U}^\theta(x/\tau', \tau/\tau', 1) - \frac{x}{\tau'}\frac{\partial\mathcal{U}^\theta(x/\tau', \tau/\tau', 1)}{\partial(x/\tau')} - \frac{\tau}{\tau'}\frac{\partial\mathcal{U}^\theta(x/\tau', \tau/\tau', 1)}{\partial(\tau/\tau')} \\
&= v_\theta^{\mathrm{eff}}(\mathcal{U}^\theta(x/\tau', \tau/\tau', 1)),
\end{aligned}
\tag{194}
$$

where the first equality follows from the scaling property $\mathcal{U}^\theta(x, \tau, \tau') = \tau'\mathcal{U}^\theta(x/\tau', \tau/\tau', 1)$ valid for the partitioning protocol state, while in the third equality directly follows upon differentiating with respect to $\tau'$ the integral equation (135) for $\mathcal{U}^\theta(x, \tau, \tau')$. Inserting the expression for $\mathcal{U}^\theta(x/\tau', \tau/\tau', 1)$ from (194) into the expression for the effective acceleration in (190), one has that the latter can be rewritten as

$$
a_\theta^{\mathrm{eff}}(\mathcal{U}^\theta(\xi/a, 1/a, 1)) = \frac{1}{2\pi}\delta n_\theta\frac{\delta(v_\theta^{\mathrm{eff}}(\xi) - \xi)}{\rho_\theta(\xi)f_\theta(\xi)} = \frac{1}{2\pi}\delta n_\theta\frac{\delta(\xi - \xi_\theta^*)}{\rho_\theta(\xi)f_\theta(\xi)},
\tag{195}
$$

with $\xi_\theta^*$ defined in (192). In order to eventually compute $c_2(T)$ from (186) one needs to know $\Delta_{a,\xi=0}$ from (189) and, therefore $a_\theta^{\text{eff}}(\mathcal{U}^\theta(0, 1/a, 1))$. The latter readily follows from the previous equation

$$a_\theta^{\text{eff}}(\mathcal{U}^\theta(0, 1/a, 1)) = \frac{1}{2\pi}\delta n_\theta \frac{\delta(v_\theta^{\text{eff}}(\xi))}{\rho_\theta f_\theta} = \frac{1}{2\pi}\delta n_\theta \frac{\delta(\theta - \theta^*)}{|\partial_\theta v_\theta^{\text{eff}}|\rho_\theta f_\theta}, \quad \text{with} \quad v^{\text{eff}}(\theta^*) = 0. \quad (196)$$

The effective acceleration determining the integral equation for $\Delta_{a,\xi=0}$ is therefore supported only on the rapidity $\theta^*$ such that the effective velocity $v_\theta^{\text{eff}}$ on the ray $\xi = 0$ is zero. Inserting the expression (196) into (189) and eventually into Eq. (186), one readily recognizes that the indirect contribution to $c_2$ vanishes because of effective velocity $v_\theta^{\text{eff}}$ factor appearing into the integral (186). Therefore, we conclude that $c_2(T)$ is exactly given by Eq. (188), which remarkably depends solely on thermodynamic quantities dependent on the state $\xi = 0$. This constitutes the derivation of the result in Eq. (158) of the main text within the inhomogeneous BFT formalism.

In order to compute higher cumulants and the whole scaled-cumulant generating function $F(\lambda, T)$ one needs to solve the inhomogeneous BFT flow equation, which describes the flow of space-time dependent Lagrange multipliers $\underline{\beta}(x,t)$, in the manifold of inhomogeneous GGE states $\underline{\beta}(x,t) \mapsto \underline{\beta}(x,t,\lambda)$. This equation generalizes Eq. (79) to long-wavelength initial states (12) by embodying the effect of indirect correlations among normal modes as per the indirect propagator $\Delta$. This equation has been reported in Ref. [23] and it relies on the results of Ref. [19] for integrable systems. As such, the inhomogeneous BFT is at present limited to the latter class of systems, differently from its homogeneous counterpart in Eq. (79). We do not report the equation here as the investigation of higher order cumulants is left for future works. Here it is sufficient to say that from the solution of the flow equation, the SCGF $F(\lambda, T)$ in Eq. (14), is eventually retrieved as $F(\lambda, T) = \int_0^\lambda d\lambda' \int_0^T dt \, j_{i_*}(0, t, \lambda')/T$, with $j_{i_*}(0, t, \lambda') = \langle \hat{j}_{i_*}(0, 0) \rangle_{\underline{\beta}(0,t,\lambda')}$, which is remarkably similar to the result from BMFT in Eq. (78).

It is here fundamental to emphasize that, given our findings about the existence of long-range Euler-scaled correlations, the inhomogeneous version of BFT still requires more checks. As a matter of fact, this theory is based on Eq. (180) and therefore by construction it does not account for the long-range contribution to correlations generated by normal modes coherently emitted at the past from the inhomogeneity (see the discussion in Appendix C). That being said, in the present manuscript, we numerically verified that the inhomogeneous BFT gives the correct second cumulant $c_2^{\text{part}}$ in Eq. (158) for the partitioning protocol on the ray $\xi = x/t = 0$ (see the discussion in Subsec. 5.4 of the main text and Fig. 3). This a consequence of the fact that the contribution to current-current correlators stemming from the correlations between normal modes in both the inhomogeneous BFT and the BMFT vanishes because the fluid velocity of the normal mode that propagates along the ray $\xi = 0$ is zero (see Eq. (196)). The result for $c_2^{\text{part}}$ provides the first confirmation of the validity of the inhomogeneous BFT, at least for cumulants in the partitioning protocol. The effect of the presence of long-range correlations on higher-order cumulants and on the scaled-cumulant generating function $F(\lambda, T)$, requires, however, a more in-depth analysis and numerical checks which would shed light on the relation betwen the BMFT and the inhomogeneous BFT and the limits of validity of the latter. Regarding correlations functions, instead, whenever the contribution from the long-range Euler scaled correlations is present, we expect the BFT and the BMFT to predict different results, the latter of which is deemed to be correct.

# E  Non-canonical mesoscopic fluctuations out of equilibrium: TASEP

The TASEP is a classical stochastic exclusion process defined on a lattice, where particles hop only towards one direction randomly, subject to hard-core exclusion [113]. Its hydrodynamics is well-known and is given by the inviscid Burgers equation

$$\partial_t \rho(x,t) + \partial_x[\rho(x,t)(1-\rho(x,t))] = 0, \tag{197}$$

where $\rho(x,t)$ is the particle density. Its conjugate variable $\beta(x,t)$ is then determined by $\partial\beta/\partial\rho = -1/(\rho(1-\rho))$, which yields $\beta[\rho] = -\log(\rho/(1-\rho))$. The rest of the MFT equations (98) carries over, except having only one component with the flux Jacobian replaced by $v[\rho] := 1 - 2\rho$. Following the same argument as after Eq. (109) in Subsec. 4.6, we have $H(x,t_1) = \lambda\delta(x-x_1)$. Defining the characteristic curve $\mathcal{U}(x,t;t_1)$ by $x = v[\beta(\mathcal{U}(x,t;t_1),t_1)](t-t_1) + \mathcal{U}(x,t;t_1)$ so that $H(x,t) = H(\mathcal{U}(x,t;t_1),t_1)$, the full profile of $H$ can be given by $H(x,t) = \lambda\delta(\mathcal{U}(x,t;t_1) - x_1)$. Further introducing $u(x,t) := \mathcal{U}(x,t;0)$ and invoking $\beta(x_2,t_1) = \beta(u(x_2,t_1),0)$, we therefore have, at $\lambda = 0$,

$$\partial_\lambda\beta(x_2,t_1) = (\partial_\lambda\beta)(u,0) + \partial_\lambda u\partial_u\beta = \frac{(\partial_\lambda\beta)(u,0)}{1+\partial v(u,0)t_1}, \tag{198}$$

where we used $\partial_\lambda u = -(t\,\partial_\beta v(\partial_\lambda\beta)(u,0))/(1+\partial v(u,0)t)$ and we denote by $\partial$ the derivative of the function with respect to its spatial argument. Since $(\partial_\lambda\beta)(u(x_2,t_1),0) = -\delta(x_1-x_2)$, it turns out that the nontrivial interaction as well as the initial homogeneity amounts to a change of the local covariance matrix rather than long-range correlations:

$$S_{\hat{q}_0,\hat{q}_0}(x_1,t_1;x_2,t_2) = \frac{\delta(x_1-x_2)}{1+\partial v(u,0)t_1}\mathsf{C}_{00}(x_1,t_1), \tag{199}$$

where $\mathsf{C}_{00}[\rho] = \rho(1-\rho)$. Of course, it is a well-known fact that the TASEP could generically develop shocks [114], but at least up to the time when it starts developing shocks, the above correlation function is expected to be valid. The inevitable appearance of a shock also prevents the correlator from having the vanishing weight when $t_1 \to \infty$. The change of the local weight suggests an intriguing phenomenon in the TASEP, which is that the fluid cells cannot be thought of as being described by the local Gibbs distribution $e^{-\beta(x,t)\overline{Q}_0}$, as the weight differs from what one obtains from this distribution, i.e., $\mathsf{C}_{00}(x,t)$. Clearly this phenomenon persists even for other one-component systems whose hydrodynamics are controlled by the hyperbolic equation of type $\partial_t\rho + f(\rho)\partial_x\rho = 0$ for an arbitrary function $f(\rho)$. We therefore expect that the lack of long-range correlations in systems that have only one conservation law to be generically true.

# F  The MFT flow equation for current fluctuations in integrable systems

The initial condition (144) allows us to write down the flow equation for $\epsilon^\theta(x,t)$. Since $\epsilon^\theta(x,t) = \epsilon(u^\theta(x,t),0)$, we have

$$
\begin{aligned}
\partial_\lambda \epsilon^\theta(x,t) &= \partial_\lambda u^\theta \, \partial_y \epsilon^\theta(y,0)\Big|_{y=u^\theta(x,t)} + (\partial_\lambda \epsilon^\theta)(u^\theta(x,t),0) \\
&= \lambda h_{\mathrm{dr}}^\theta(u^\theta(x,t),0)\Theta(u^\theta(x,t),0) + \partial_\lambda u^\theta \, \partial_x \epsilon^\theta(x,0)\Big|_{x=u^\theta(x,t)} \\
&\quad - (R^{-\mathrm{T}})^\theta{}_\alpha(u^\theta(x,t),0)\,\partial_\lambda \left[\lambda(R^{\mathrm{T}})^\alpha{}_\phi(0,T)\Theta(x-u^\phi(0,T))h_{\mathrm{dr}}^\phi(0,T)\right].
\end{aligned}
\tag{200}
$$

The treatment of $\partial_x \epsilon(x,0)$ depends on the initial condition we choose. For the most of the cases, we can simply use the boundary condition of the MFT equations and get

$$
\partial_x \epsilon(x,0) = \lambda h_{\mathrm{dr}}^\theta(0,0)\delta(x) + (R^{-\mathrm{T}})^\theta{}_\phi(x,0)\partial_x \beta_{\mathrm{ini}}^\phi(x) - \lambda\delta(x-u^\theta(0,T))h_{\mathrm{dr}}^\theta(0,T).
\tag{201}
$$

Having a quantity whose argument is $x = t = 0$ could however cause problems in some situation, such as the partitioning protocol. In such a case one needs to take a detour by first obtaining $\epsilon^\theta(x,0)$ by differentiating $\beta^\theta(x,0)$ with respect to $\lambda$, obtaining

$$
\begin{aligned}
\epsilon^\theta(x,0) ={}& \epsilon_{\mathrm{ini}}^\theta(x) \\
&+ \int_0^\lambda \mathrm{d}\lambda' \bigg[ h_{\mathrm{dr}}^\theta(x,0)\Theta(x) - (R^{-\mathrm{T}})^\theta{}_\alpha(x,0) \\
&\qquad\qquad\qquad \times \partial_{\lambda'}\left(\lambda'(R^{\mathrm{T}})^\alpha{}_\phi(0,T)\Theta(x-u^\phi(0,T))h_{\mathrm{dr}}^\phi(0,T)\right) \bigg],
\end{aligned}
\tag{202}
$$

from which one obtains $\partial_x \epsilon^\theta(x,0)$. Therefore the flow equation in a generic inhomogeneous case or in the partitioning protocol is given by (200) with (201) or (202), respectively.

# G  Third cumulant in the homogeneous case

In this section we shall explain how one can obtain $c_3$ using the flow equation (147). Recall that what we have to evaluate is

$$
c_3 = \frac{1}{T}\int_0^\infty \mathrm{d}x \int_{\mathbb{R}} \mathrm{d}\theta \, h^\theta \partial_\lambda^2 \left(\rho_\theta(x,T) - \rho_\theta(x,0)\right)|_{\lambda\to 0}.
\tag{203}
$$

For that, we first note

$$
\partial_\lambda^2 \rho_\theta(x,t) = -\partial_\lambda^2 \epsilon^\phi(x,t)(R^{-\mathrm{T}})^\phi{}_\theta(x,t)\chi_\phi(x,t) - \partial_\lambda \epsilon^\phi(x,t)\partial_\lambda((R^{-\mathrm{T}})^\phi{}_\theta(x,t)\chi_\phi(x,t)).
\tag{204}
$$

Let us start with $\partial_\lambda^2 \epsilon^\phi(x,t)$. Using (147) we have

$$
\begin{aligned}
\partial_\lambda^2 \epsilon^\phi(x,t)\Big|_{\lambda\to 0} ={}& 2\left[\partial_\lambda h_{\mathrm{dr}}^\phi(0,0)\Big|_{\lambda\to 0}\Theta(u^\phi(x,t)) - \partial_\lambda h_{\mathrm{dr}}^\phi(0,T)\Big|_{\lambda\to 0}\Theta(r^\phi(x,t))\right] \\
&+ 2h_{\mathrm{dr}}^\phi\left[\partial_\lambda u^\phi(x,t)\Big|_{\lambda\to 0}\delta(u^\phi(x,t)) - \partial_\lambda r^\phi(x,t)\Big|_{\lambda\to 0}\delta(r^\phi(x,t))\right].
\end{aligned}
\tag{205}
$$

Note

$$\partial_\lambda h_{\mathrm{dr}}^\phi(0,0)\Big|_{\lambda\to 0} = \partial_\lambda \epsilon^\gamma(0,0)|_{\lambda\to 0}\,\frac{\partial}{\partial \epsilon^\gamma}h_{\mathrm{dr}}^\phi = -(T^{\mathrm{dr}})^{\phi\gamma}n^\gamma f^\gamma h_{\mathrm{dr}}^\gamma\,\partial_\lambda \epsilon^\gamma(0,0)|_{\lambda\to 0}\,, \tag{206}$$

where we recalled that $h_{\mathrm{dr}}^\phi = (R^{-\mathrm{T}})^\phi_{\ \gamma}h^\gamma$. Defining $\Theta(0) := 1/2$, $\partial_\lambda \epsilon^\gamma(0,0)$ is simply

$$\partial_\lambda \epsilon^\gamma(0,0)|_{\lambda\to 0} = h_{\mathrm{dr}}^\gamma\left(\frac{1}{2}-\Theta(v^\gamma)\right) = -\frac{1}{2}\mathrm{sgn}\,v_{\mathrm{eff}}^\gamma h_{\mathrm{dr}}^\gamma. \tag{207}$$

Likewise

$$\partial_\lambda \epsilon^\gamma(0,T)|_{\lambda\to 0} = h_{\mathrm{dr}}^\gamma\left(\Theta(-v^\gamma)-\frac{1}{2}\right) = -\frac{1}{2}\mathrm{sgn}\,v_{\mathrm{eff}}^\gamma h_{\mathrm{dr}}^\gamma. \tag{208}$$

Combining these we can compute the first line in (205)

$$2\left[\partial_\lambda h_{\mathrm{dr}}^\phi(0,0)\Big|_{\lambda\to 0}\Theta(u^\phi(x,t))-\partial_\lambda h_{\mathrm{dr}}^\phi(0,T)\Big|_{\lambda\to 0}\Theta(r^\phi(x,t))\right]$$
$$= \mathrm{sgn}\,v_{\mathrm{eff}}^\gamma(T^{\mathrm{dr}})^\phi_{\ \gamma}n_\gamma f_\gamma(h_{\mathrm{dr}}^\gamma)^2\left(\Theta(x-v_{\mathrm{eff}}^\phi t)-\Theta(x-v_{\mathrm{eff}}^\phi(t-T))\right). \tag{209}$$

Next we deal with the second line in (205). For that note first

$$\partial_\lambda r^\phi(x,t)\Big|_{\lambda\to 0}(p')_{\mathrm{dr}}^\phi = \int_{-\infty}^x \mathrm{d}y\,\partial_\lambda(p')_{\mathrm{dr}}^\phi[n(y,t)]\Big|_{\lambda\to 0} - \int_{-\infty}^{r^\phi(x,t)}\mathrm{d}y\,\partial_\lambda(p')_{\mathrm{dr}}^\phi[n^T(y)]\Big|_{\lambda\to 0}. \tag{210}$$

Furthermore

$$\int_{-\infty}^x \mathrm{d}y\,\partial_\lambda(p')_{\mathrm{dr}}^\phi[n(y,t)]\Big|_{\lambda\to 0} = \frac{\partial(p')_{\mathrm{dr}}^\phi}{\partial \epsilon^\gamma}\int_{-\infty}^x \mathrm{d}y\,\partial_\lambda \epsilon^\gamma[n(y,t)]|_{\lambda\to 0}$$
$$= -\frac{\partial(p')_{\mathrm{dr}}^\phi}{\partial \epsilon^\gamma}h_{\mathrm{dr}}^\gamma\int_{-\infty}^x \mathrm{d}y\,\left(\Theta(y-v_{\mathrm{eff}}^\gamma(t-T))-\Theta(y-v_{\mathrm{eff}}^\gamma t)\right), \tag{211}$$

and

$$\int_{-\infty}^{r^\phi(x,t)}\mathrm{d}y\,\partial_\lambda(p')_{\mathrm{dr}}^\phi[n^T(y)]\Big|_{\lambda\to 0} = -\frac{\partial(p')_{\mathrm{dr}}^\phi}{\partial \epsilon^\gamma}h_{\mathrm{dr}}^\gamma\int_{-\infty}^{r^\phi(x,t)}\mathrm{d}y\,\left(\Theta(y)-\Theta(y-v_{\mathrm{eff}}^\gamma T)\right). \tag{212}$$

Clearly $\partial_\lambda r^\phi(x,T)\big|_{\lambda\to 0} = 0$ and

$$\partial_\lambda r^\phi(x,0)\Big|_{\lambda\to 0}$$
$$= h_{\mathrm{dr}}^\gamma\frac{\partial}{\partial \epsilon^\gamma}\log(p')_{\mathrm{dr}}^\phi\int_{-\infty}^x \mathrm{d}y\,\left(\Theta(y)+\Theta(y+v_{\mathrm{eff}}^\phi T)-\Theta(y+v_{\mathrm{eff}}^\gamma T)-\Theta(y+(v_{\mathrm{eff}}^\phi-v_{\mathrm{eff}}^\gamma)T)\right), \tag{213}$$

which entails

$$\delta(r^\phi(x,0))\partial_\lambda r^\phi(x,0)\Big|_{\lambda\to 0}$$
$$= h_{\mathrm{dr}}^\gamma\frac{\partial}{\partial \epsilon^\gamma}\log(p')_{\mathrm{dr}}^\phi\left(v_{\mathrm{eff}}^\gamma\Theta(-v_{\mathrm{eff}}^\gamma)+v^{\phi\gamma}\Theta(-v^{\phi\gamma})-v^\phi\Theta(-v_{\mathrm{eff}}^\phi)\right)\delta(x+v_{\mathrm{eff}}^\phi T)$$
$$= -\frac{h_{\mathrm{dr}}^\gamma}{\rho_\phi^{\mathrm{tot}}}(T^{\mathrm{dr}})^{\phi\gamma}\chi_\gamma\left(v_{\mathrm{eff}}^\gamma\Theta(-v_{\mathrm{eff}}^\gamma)+v^{\phi\gamma}\Theta(-v^{\phi\gamma})-v_{\mathrm{eff}}^\phi\Theta(-v_{\mathrm{eff}}^\phi)\right)\delta(x+v_{\mathrm{eff}}^\phi T), \tag{214}$$

where $v^{\phi\gamma} = v^\phi_{\text{eff}} - v^\gamma_{\text{eff}}$. In the same way, one can also show that $\partial_\lambda u^\phi(x,0)\big|_{\lambda\to0} = 0$ and

$$\delta(u^\phi(x,T))\partial_\lambda u^\phi(x,T)\Big|_{\lambda\to0}$$
$$= -\frac{h^\gamma_{\text{dr}}}{\rho^{\text{tot}}_\phi}(T^{\text{dr}})^{\phi\gamma}\chi_\gamma\left(v^\gamma_{\text{eff}}\Theta(v^\gamma_{\text{eff}}) + v^{\phi\gamma}\Theta(v^{\phi\gamma}) - v^\phi_{\text{eff}}\Theta(v^\phi_{\text{eff}})\right)\delta(x - v^\phi_{\text{eff}}T). \qquad (215)$$

Next we turn to the second term in (204). This is easier than the first term because we merely need to evaluate

$$\partial_\lambda\epsilon^\phi(x,t)\partial_\lambda((R^{-\text{T}})^\phi_\theta(x,t)\chi_\phi(x,t))\Big|_{\lambda\to0} = \partial_\lambda\epsilon^\phi(x,t)\Big|_{\lambda\to0}\partial_\lambda\epsilon^\gamma(x,t)|_{\lambda\to0}\frac{\partial}{\partial\epsilon^\gamma}((R^{-\text{T}})^\phi_\theta\chi_\phi)$$
$$= h^\phi_{\text{dr}}h^\gamma_{\text{dr}}\Theta^\phi(x,t)\Theta^\gamma(x,t)\frac{\partial}{\partial\epsilon^\gamma}((R^{-\text{T}})^\phi_\theta\chi_\phi), \qquad (216)$$

where $\Theta^\phi(x,t) := \Theta(x - v^\phi_{\text{eff}}(t-T)) - \Theta(x - v^\phi_{\text{eff}}t)$. Notice

$$\frac{\partial}{\partial\epsilon^\gamma}((R^{-\text{T}})^\phi_\theta\chi_\phi)$$
$$= -\chi_\phi(T^{\text{dr}})^{\phi\gamma}n^\gamma f^\gamma(R^{-\text{T}})^\gamma_\theta - (R^{-\text{T}})^\phi_\theta n_\phi f_\phi(T^{\text{dr}})^{\phi\gamma}\chi^\gamma - (R^{-\text{T}})^\phi_\theta\chi_\phi(1 - 2n_\phi)\delta^\phi_\gamma. \qquad (217)$$

Plugging this into (216), we get

$$\partial_\lambda\epsilon^\phi(x,t)\partial_\lambda((R^{-\text{T}})^\phi_\theta(x,t)\chi_\phi(x,t))\Big|_{\lambda\to0} =$$
$$- 2(R^{-\text{T}})^\gamma_\theta h^\phi_{\text{dr}}h^\gamma_{\text{dr}}\chi_\phi(T^{\text{dr}})^{\phi\gamma}n^\gamma f^\gamma\Theta^\phi(x,t)\Theta^\gamma(x,t) - (R^{-\text{T}})^\phi_\theta(h^\phi_{\text{dr}})^2\chi_\phi(1 - 2n_\phi)(\Theta^\phi(x,t))^2. \qquad (218)$$

Having all the relevant terms at our disposal, we are in the position to compute $c_3$. Let us first compute the nondiagonal contributions, which read

$$(R^{-\text{T}})^\gamma_\theta h^\phi_{\text{dr}}h^\gamma_{\text{dr}}\chi_\phi(T^{\text{dr}})^\phi_\gamma n_\gamma f_\gamma\frac{1}{T}\int_0^\infty \text{d}x$$
$$\times\left(-\text{sgn}\, v^\gamma_{\text{eff}}\left(\Theta(x - v^\phi_{\text{eff}}T) - \Theta(x) - (\Theta(x) - \Theta(x + v^\phi_{\text{eff}}T))\right)\right.$$
$$+ 2\left(v^\phi_{\text{eff}}\Theta(v^\phi_{\text{eff}}) - v^{\phi\gamma}\Theta(-v^{\phi\gamma}) - v^\gamma_{\text{eff}}\Theta(v^\gamma_{\text{eff}})\right)\delta(x - v^\gamma_{\text{eff}}T)$$
$$+ 2\left(v^\phi\Theta(-v^\phi_{\text{eff}}) - v^{\phi\gamma}\Theta(v^{\phi\gamma}) - v^\gamma_{\text{eff}}\Theta(-v^\gamma_{\text{eff}})\right)\delta(x + v^\gamma_{\text{eff}}T)$$
$$+ 2(\Theta(x) - \Theta(x - v^\phi_{\text{eff}}T))(\Theta(x) - \Theta(x - v^\gamma_{\text{eff}}T))$$
$$\left. - 2(\Theta(x + v^\phi_{\text{eff}}T) - \Theta(x))(\Theta(x + v^\gamma_{\text{eff}}T) - \Theta(x))\right). \qquad (219)$$

It's easier to work out each building block. The first one is

$$-\frac{1}{T}\text{sgn}\, v^\gamma_{\text{eff}}\int_0^\infty \text{d}x\left(\Theta(x - v^\phi_{\text{eff}}T) - \Theta(x) - (\Theta(x) - \Theta(x + v^\phi_{\text{eff}}T))\right) = \text{sgn}\, v^\gamma_{\text{eff}}|v^\delta_{\text{eff}}|. \qquad (220)$$

The second one is

$$\frac{1}{T}\int_0^\infty \text{d}x\left(\left(v^\phi_{\text{eff}}\Theta(v^\phi_{\text{eff}}) - v^{\phi\gamma}\Theta(-v^{\phi\gamma}) - v^\gamma_{\text{eff}}\Theta(v^\gamma_{\text{eff}})\right)\delta(x - v^\gamma_{\text{eff}}T)\right.$$
$$\left. + \left(v^\phi_{\text{eff}}\Theta(-v^\phi_{\text{eff}}) - v^{\phi\gamma}\Theta(v^{\phi\gamma}) - v^\gamma_{\text{eff}}\Theta(-v^\gamma_{\text{eff}})\right)\delta(x + v^\gamma_{\text{eff}}T)\right)$$
$$= \left(v^\phi_{\text{eff}}\Theta(v^\phi_{\text{eff}}) - v^{\phi\gamma}\Theta(-v^{\phi\gamma})\right)\Theta(v^\gamma_{\text{eff}}) + \left(v^\phi_{\text{eff}}\Theta(-v^\phi_{\text{eff}}) - v^{\phi\gamma}\Theta(v^{\phi\gamma})\right)\Theta(-v^\gamma_{\text{eff}}) - v^\gamma_{\text{eff}}. \qquad (221)$$

The third one turns out to be the same as the first one, namely

$$
\begin{aligned}
\frac{1}{T} \int_0^\infty & \left( (\Theta(x) - \Theta(x - v_{\text{eff}}^\phi T))(\Theta(x) - \Theta(x - v_{\text{eff}}^\gamma T)) \right. \\
& \left. - (\Theta(x + v_{\text{eff}}^\phi T) - \Theta(x))(\Theta(x + v_{\text{eff}}^\gamma T) - \Theta(x)) \right) \\
& = \Theta(v_{\text{eff}}^\gamma)\Theta(v^\phi)\min(v_{\text{eff}}^\gamma, v_{\text{eff}}^\phi) + \Theta(-v_{\text{eff}}^\gamma)\Theta(-v_{\text{eff}}^\phi)\max(v_{\text{eff}}^\gamma, v_{\text{eff}}^\phi).
\end{aligned} \tag{222}
$$

Hence, the nondiagonal terms add up to

$$
\begin{aligned}
& \operatorname{sgn} v_{\text{eff}}^\gamma |v_{\text{eff}}^\delta| + 2 \left( \Theta(v_{\text{eff}}^\gamma)\Theta(v_{\text{eff}}^\phi)\min(v_{\text{eff}}^\gamma, v_{\text{eff}}^\phi) + \Theta(-v_{\text{eff}}^\gamma)\Theta(-v_{\text{eff}}^\phi)\max(v_{\text{eff}}^\gamma, v_{\text{eff}}^\phi) \right) \\
& + 2 \left( \left( v_{\text{eff}}^\phi \Theta(v_{\text{eff}}^\phi) - v^{\phi\gamma}\Theta(-v^{\phi\gamma}) \right) \Theta(v_{\text{eff}}^\gamma) + \left( v_{\text{eff}}^\phi \Theta(-v_{\text{eff}}^\phi) - v^{\phi\gamma}\Theta(v^{\phi\gamma}) \right) \Theta(-v_{\text{eff}}^\gamma) - v_{\text{eff}}^\gamma \right),
\end{aligned} \tag{223}
$$

which turns out to be $3|v_{\text{eff}}^\delta| \operatorname{sgn} v_{\text{eff}}^\gamma$. We then have

$$
\begin{aligned}
c_3 &= h^\theta (h^{\text{dr};\phi})^2 \left( s_\phi \chi_\phi |v_\phi^{\text{eff}}| (R^{-\text{T}})^\phi{}_\theta + 3(R^{-\text{T}})^\gamma{}_\theta n_\phi f_\phi |v_\gamma^{\text{eff}}| s_\phi \chi_\gamma (T^{\text{dr}})^{\gamma\phi} \right) \\
&= h^\theta (h^{\text{dr};\phi})^2 \left( s_\phi \chi_\phi |v_\phi^{\text{eff}}| (R^{-\text{T}})^\phi{}_\theta + 3(R^{-\text{T}})^\gamma{}_\theta f_\phi |v_\gamma^{\text{eff}}| s_\phi \chi_\gamma (R^{-1})^\phi{}_\gamma - 3 f_\phi |v_\phi^{\text{eff}}| \chi_\phi (R^{-\text{T}})^\phi{}_\theta \right) \\
&= \chi_\phi |v_\phi^{\text{eff}}| h^{\text{dr};\phi} \left( s_\phi \tilde{f}_\phi (h_\phi^{\text{dr}})^2 + 3 [sf(h_{\text{dr}})^2]_\phi^{\text{dr}} \right),
\end{aligned} \tag{224}
$$

which coincides with the known $c_3$ from Ref. [21].

# H Dynamical correlation functions in integrable systems

In this Appendix, we provide details on the calculations of dynamical correlation functions from inhomogeneous states in integrable systems. In Appendix H.1, we detail the derivation of the full profile of $\partial_\lambda \epsilon^\theta(x, t)$, with particular emphasis on the application to equal time correlations and hence to the appearance of Euler-scaled long-range correlations. In the Appendix H.2, we specialize the analysis to the hard-rod model from the partitioning protocol inhomogeneous initial state.

## H.1 General cases

We start with the general formula for $S_{\hat{q}_{i_1}, \hat{q}_{i_2}}(x_1, t_1; x_2, t_2)$ evaluated with respect to an arbitrary initial condition $\beta_{\text{ini}}(x)$. Using (165) and $\partial_\lambda \epsilon^\theta(x_2, t_2) = \partial_\lambda u^\theta \partial_y \epsilon^\theta(y, 0)\big|_{y=u^\theta} + (\partial_\lambda \epsilon)(u^\theta, 0)$, the correlator is given by

$$
S_{\hat{q}_{i_1}, \hat{q}_{i_2}}(x_1, t_1; x_2, t_2) = - \int_{\mathbb{R}} \mathrm{d}\theta \, h_{i_2}^\theta (R^{-\text{T}})^\theta{}_\phi(x_2, t_2) \chi_\phi(x_2, t_2) \partial_\lambda \epsilon^\phi(x_2, t_2) \bigg|_{\lambda=0}. \tag{225}
$$

The task therefore is to compute $\partial_\lambda \epsilon_\theta(x_2, t_2)$. To this end recall first that

$$
\begin{aligned}
\beta^\theta(x, 0) = \beta_{\text{ini}}^\theta(x) - \lambda \Big( & (R^{\text{T}})^\theta{}_\phi(x_1, t_1) h_{\text{dr}}^\phi(x_1, t_1) \frac{\delta(x - u^\phi(x_1, t_1))}{\partial \mathcal{U}^\phi(u^\phi(x_1, t_1), 0; t_1)} \\
& + \partial \left( (R^{\text{T}})^\theta{}_\phi(x_1, t_1) h_{\text{dr}}^\phi(x_1, t_1) \right) \Theta(x - u^\phi(x_1, t_1)) \Big).
\end{aligned} \tag{226}
$$

Recall that here $\partial$ without subscript implies that it acts as the spacial derivative with respect to the space argument of a function. The previous equation implies that $\partial_\lambda \epsilon_\theta(x_2, t_2)$ satisfies

$$\partial_\lambda \epsilon^\theta(x_2, t_2)$$
$$= -\frac{h_{\mathrm{dr}}^\theta(x_1, t_1)}{\partial \mathcal{U}^\theta(u^\theta(x_1, t_1), 0; t_1)} \delta(u^\theta(x_2, t_2) - u^\theta(x_1, t_1)) + \partial_\lambda u^\theta(x_2, t_2) \partial \epsilon_{\mathrm{ini}}(u^\theta(x_2, t_2))$$
$$+ (R^{-\mathrm{T}})^\theta{}_\phi(u^\theta(x_2, t_2), 0) \partial \left( (R^{\mathrm{T}})^\phi{}_\alpha(x_1, t_1) h_{\mathrm{dr}}^\alpha(x_1, t_1) \right) \Theta(u^\theta(x_2, t_2) - u^\alpha(x_1, t_1)). \quad (227)$$

Note that $\partial_\lambda u^\theta(x, t)$ satisfies

$$\partial_\lambda u^\theta(x, t) \rho_{\mathrm{tot}}^\theta(u^\theta(x, t), 0)$$
$$= -\int_{-\infty}^{u^\theta(x,t)} \mathrm{d}y \, \partial_\lambda \rho_{\mathrm{tot}}^\theta(y, 0) + \int_{-\infty}^x \mathrm{d}y \, \partial_\lambda \rho_{\mathrm{tot}}^\theta(y, t)$$
$$= -\int_{-\infty}^{u^\theta(x,t)} \mathrm{d}y \, (R^{-\mathrm{T}})^\theta{}_\phi(y, 0) \chi_\phi(y, 0) \partial_\lambda \epsilon^\phi(y, 0) + \int_{-\infty}^x \mathrm{d}y \, (R^{-\mathrm{T}})^\theta{}_\phi(y, t) \chi_\phi(y, t) \partial_\lambda \epsilon^\phi(y, t).$$
$$(228)$$

The case of equal-time correlations, i.e., $t_1 = t_2 = t$, is of particular interest. In this case the flow equation becomes

$$\partial_\lambda \epsilon^\theta(x_2, t) = -h_{\mathrm{dr}}^\theta(x_1, t) \delta(x_1 - x_2) + \partial_\lambda u^\theta(x_2, t) \partial \epsilon_{\mathrm{ini}}(u^\theta(x_2, t))$$
$$+ (R^{-\mathrm{T}})^\theta{}_\phi(u^\theta(x_2, t), 0) \partial \left( (R^{\mathrm{T}})^\phi{}_\alpha(x_1, t) h_{\mathrm{dr}}^\alpha(x_1, t) \right) \Theta(u^\theta(x_2, t) - u^\alpha(x_1, t)).$$
$$(229)$$

Clearly the first term accounts for the local covariance matrix weight $C_{i_1 i_2}(x_1, t)$ that depends on the local state, whereas the rest of the terms contribute to long-range correlations, which amounts to $E_{i_1 i_2}(x_1, t)$ given by Eqs. (132) and (133). To be more precise, let us assume that $\partial_\lambda \epsilon^\theta(x_2, t)$ takes the following form: $\partial_\lambda \epsilon^\theta(x_2, t) = \mathcal{E}_0(x_1, t) \delta(x_1 - x) + \mathcal{E}(x, t)$, where $\mathcal{E}(x, t)$ is a regular function that contains no delta-function. Then plugging this into (229) with (228), it is readily seen that $\mathcal{E}_0(x_1, t)$ has to be $\mathcal{E}_0(x_1, t) = -h_{\mathrm{dr}}^\theta(x_1, t)$, and $\mathcal{E}(x, t)$ satisfies the integral equation (133) (together with Eqs. (134)-(138)). Since the integral equation has the unique solution, which can also be verified numerically, the above ansatz is justified.

## H.2 Partitioning protocol in the hard-rod gas

As mentioned in the main text in Subsec. 5.5, in the partitioning protocol, the long-range correlations have two origins: one is the same as other protocols, i.e., correlations between normal modes, and another is due to early time dynamics controlled by non-universal micsroscopic physics. While the latter, on the basis of the numerical analysis we carried out, gives the dominant contribution, it is interesting to see how the former contribution can be exactly computed in this case. To better illustrate it, let us focus on the system of hard rods (see Appendix B) and choose the particle density for both densities in the correlator: $i_1 = i_2 = 0$. Moreover we set $x_2 = 0$. The flow equation (229) then reads

$$\partial_\lambda \epsilon^\theta(0, t) = -\rho_{\mathrm{tot}}(0) \delta(x_1) - \partial_\lambda u^\theta(0, t) \delta \epsilon^\theta \delta(u^\theta(0, t))$$
$$+ \frac{1}{t} \Big( -a \rho_R^{\mathrm{tot}} \delta n^\alpha \delta(\xi_1 - \xi_*(\alpha)) \rho_{\mathrm{tot}}(\xi_1) \Theta(-v_{\mathrm{eff}}^\theta(0))$$
$$+ \left( (R^{-\mathrm{T}})^\theta{}_\alpha(u^\theta, 0) + a \rho_{\mathrm{tot}}(u^\theta, 0) n_\alpha(\xi_1) \right) \rho_{\mathrm{tot}}'(\xi_1) \Theta(u^\theta(0, t) - u^\alpha(x_1, t)) \Big), \quad (230)$$

where $\delta\epsilon^\theta = \epsilon_R^\theta - \epsilon_L^\theta$ and $\delta n^\theta = n_R^\theta - n_L^\theta$. To proceed, we note

$$(R^{-\mathrm{T}})^\theta{}_\alpha(u^\theta, 0)\Theta(u^\theta(0,t) - u^\alpha(x_1, t))$$
$$= \Theta(-x_1) - a\rho_{\mathrm{tot}}(u^\theta, 0)n_\alpha(u^\theta, 0)\Theta(u^\theta(0,t) - u^\alpha(x_1, t)), \qquad (231)$$

and

$$(n_\alpha(u^\alpha(x_1, t), 0) - n_\alpha(u^\theta(0,t), 0))\Theta(u^\theta(0,t) - u^\alpha(x_1, t)) = \delta n^\alpha \Theta(-v_{\mathrm{eff}}^\theta(0))\Theta(\xi_*(\alpha) - \xi_1). \quad (232)$$

Furthermore,

$$\rho_R^{\mathrm{tot}}\partial_{\xi_1}(\rho_{\mathrm{tot}}(\xi_1)\delta n^\alpha \Theta(\xi_*(\alpha) - \xi_1)) = \rho_R^{\mathrm{tot}}\partial_{\xi_1}(\rho_{\mathrm{tot}}(\xi_1)(n^\alpha(\xi_1) - n_R^\alpha))$$
$$= \rho_R^{\mathrm{tot}}\rho'(\xi_1) - \rho_R\rho_{\mathrm{tot}}'(\xi_1)$$
$$= \rho'(\xi_1). \qquad (233)$$

Using these the flow equation is now simplified to

$$\partial_\lambda\epsilon^\theta(0,t)$$
$$= -\rho_{\mathrm{tot}}(0)\delta(x_1) - \partial_\lambda u^\theta(0,t)\delta\epsilon^\theta\delta(u^\theta(0,t)) + \frac{1}{t}\Big(\rho_{\mathrm{tot}}'(\xi_1)\Theta(-x_1) + a\rho'(\xi_1)\Theta(-v_{\mathrm{eff}}^\theta(0))\Big). \qquad (234)$$

Next we turn to the term that involves $\partial_\lambda u^\theta$. When $\lambda = 0$ and $u^\theta(0,t) = 0$ we have,

$$\rho_{\mathrm{tot}}(0)\partial_\lambda u^\theta(0,t) = \int_{-\infty}^0 \mathrm{d}y\, \partial_\lambda\rho_{\mathrm{tot}}(y,t) - \int_{-\infty}^0 \mathrm{d}y\, \partial_\lambda\rho_{\mathrm{tot}}(y,0)$$
$$= -\int_0^t \mathrm{d}s\, \partial_\lambda v^\theta(0,s) = -a\rho_{\mathrm{tot}}(0)\rho_\phi(0)v_\phi^{\mathrm{eff}}(0)\int_0^t \mathrm{d}s\, \partial_\lambda\epsilon^\phi(0,s), \qquad (235)$$

which gives the full $\partial_\lambda\epsilon^\theta(0,t)$

$$\partial_\lambda\epsilon^\theta(0,t) = -\rho_{\mathrm{tot}}(0)\delta(x_1) + \frac{1}{t}\mathcal{E}^\theta(\xi_1) + a\delta\epsilon^\theta\delta(u^\theta(0,t))\rho_\phi(0)v_\phi^{\mathrm{eff}}(0)\int_0^t \mathrm{d}s\, \partial_\lambda\epsilon^\phi(0,s), \quad (236)$$

where

$$\mathcal{E}^\theta(\xi_1) := \rho_{\mathrm{tot}}'(\xi_1)\Theta(-x_1) + a\rho'(\xi_1)\Theta(-v_{\mathrm{eff}}^\theta(0))$$
$$= a\rho'(\xi_1)\Big(\Theta(x_1)\Theta(-v_\theta^{\mathrm{eff}}(0)) - \Theta(-x_1)\Theta(v_\theta^{\mathrm{eff}}(0))\Big). \qquad (237)$$

The term $-\rho_{\mathrm{tot}}(0)\delta(x_1)$ on the right hand side of Eq. (236) gives from Eq. (161) the local covariance matrix $\mathsf{C}_{00}(0,t)$ weight of the delta function in (168) (see also the discussion after Eq. (229)). The remaining terms determine the Euler-scaled long-range correlations $E_{00}(x_1, 0; t)$. Let us focus on this contribution and suppose $x_1 \neq 0$. It is then clear that solving the integral equation (236) recursively, the term containing $\delta(x_1)$, which is $t$-independent, merely acquires the multiplicative factor $t\delta(u^\theta(0,t))$. The latter is zero, hence the delta-correlated term remains the same. As for the other source term $\frac{1}{t}\mathcal{E}^\theta(\xi_1)$, it is clear that the iteration truncates after the second term because $v_{\theta_*(0)}^{\mathrm{eff}} = 0$. In conclusion, we have

$$\partial_\lambda\epsilon^\theta(0,t) = -\rho_{\mathrm{tot}}(0)\delta(x_1) + \frac{1}{t}\mathcal{E}^\theta(\xi_1) + a\delta\epsilon^\theta\delta(u^\theta(0,t))\rho_\phi(0)v_\phi^{\mathrm{eff}}(0)\int_0^t \mathrm{d}s\, \frac{1}{s}\mathcal{E}^\phi(x_1/s), \quad (238)$$

which gives

$$E_{00}(x_1, 0; t) = -\rho_{\text{tot}}(0)\rho_\theta(0)\left(\frac{1}{t}\mathcal{E}^\theta(\xi_1) + a\delta\epsilon^\theta\delta(u^\theta(0,t))\rho_\phi(0)v_\phi^{\text{eff}}(0)\int_0^t \mathrm{d}s\, \frac{1}{s}\mathcal{E}^\phi(x_1/s)\right),$$

(239)

provided $x_1 \neq 0$. Note again that the result above *does not* agree with numerical simulations we carried out, or to be more precise, it only gives the contribution that originates from hydrodynamics. But as we explained in the main text, the predominant contribution to the Euler scale correlator $S_{\hat{q}_0,\hat{q}_0}(x_1, t; 0, t)$ in the partitioning protocol stems from the transient physics that takes place at the early stage of the dynamics, which is not accounted for by hydrodynamics. One can, however, try to smear out the initial step function while keeping the asymptotic values of the distribution unchanged, in which case the correlator from BMFT is expected to agree with numerical simulations.

# I  Numerics: hard-rod model simulations

In this Appendix we provide details about the numerical simulations of the hard-rod model. For the numerical simulations with the partitioning protocol initial state, we focused on the case where the initial state presents a jump in the inverse temperature $\beta_{\text{ini}}^i$ at $x = 0$:

$$\beta_{\text{ini}}^i(x) = \delta^i_{\bar{i}}(\beta_L\Theta(-x) + \beta_R\Theta(x)),$$

(240)

which corresponds to the initial inhomogeneous state in Eq. (17) with the set $\mathcal{C}$ containing only the energy $q_{\bar{i}}$ conserved charge (and $\beta_0^i = 0$ otherwise). In the previous equation $\bar{i}$ is then the index corresponding to the energy conserved charge and the thermal state is identified by the source term $\beta_{L,R}^\theta = \beta_{L,R}\theta^2/2$ in Eq. (120). The thermal occupation function $n_{\beta_{L,R}}^\theta$ can be obtained by solving (176) with this source term, which turns out to admits a closed expression (see, e.g., Ref. [21]) in terms of the Lambert function $W(z)$ [115]:

$$n_{\beta_{L,R}}^\theta = \frac{e^{-\epsilon_{\beta_{L,R}}^\theta}}{2\pi} = \frac{e^{-\beta_{L,R}\theta^2/2}}{2\pi}e^{-W(ad(\beta_{L,R}))},$$

(241)

where $d(\beta) = 1/\sqrt{2\pi\beta}$. The thermal quasi-particle density $\rho_{\theta,\beta_{L,R}}$ is readily obtained from Eq. (241) and it reads as

$$\rho_{\theta,\beta_{L,R}} = f_\theta(\beta_{L,R})\rho(\beta_{L,R}),$$

(242)

where the rods spatial density $\rho(\beta)$ can also be expressed solely in terms of the Lambert function

$$\rho(\beta_{L,R}) = \frac{W(ad(\beta_{L,R}))}{a[1 + W(ad(\beta_{L,R}))]},$$

(243)

and $f_\theta(\beta)$ is the velocity distribution

$$f_\theta(\beta) = \sqrt{\frac{\beta_{L,R}}{2\pi}}e^{-\beta\theta^2}.$$

(244)

From the previous equation, it is evident that for a thermal state, the rods velocity distribution is a Gaussian with variance $1/\beta$ and mean $\mu$ zero. It is also possible to consider velocity

distribution with a non zero mean simply replacing $\theta \to \theta - \mu$ in Eq. (244), which corresponds to a boosted thermal distribution. In the simulations we always consider the case of $\mu = 0$.

The numerical simulations are done in infinite volume, but the rods are initially distributed in a symmetric interval $[-L_{\text{size}}/2, L_{\text{size}}/2]$ around the origin. In the numerical simulations, in addition to the discontinuity of the density at $x = 0$, there are, as a consequence, two depletion zones, where the density of rods jumps to zero, which move inwards as time elapses. The numerical results, as a consequence, deviate from the BMFT predictions in proximity of the depletion zones. The initial left ($[-L_{\text{size}}/2, 0]$) and right half ($[0, L_{\text{size}}/2]$) spatial distributions of rods are given in Eq. (242) with inverse temperature $\beta_L$ and $\beta_R$, respectively. Rods' velocities are sampled from the Gaussian distribution in Eq. (244) with $\beta_L$ for the rods initially in the left half, and with $\beta_R$ for the rods initially in the right one. The number $N$ of rods used in the simulations is then fixed by the initial size $L_{\text{size}}$, by the rod length $a$ and the inverse temperatures $\beta_{L,R}$ (according to the density $\rho(\beta_{L,R})$ in Eq. (243)). We emphasize that stochasticity in the simulations is only due to the initial condition according to Eqs. (240), (243) and (244), while the time evolution is purely deterministic according to the hard-rod dynamics. Numerical results are eventually obtained by averaging over a large number $M$ of independent realizations of the rods' positions and velocities.

Regarding the cumulants' analysis in the partitioning protocol we focus on the second cumulant $c_2^{\text{part}}$ for particle transport (single particle eigenvalue $N_\theta = 1$ in Eq. (174)), as explained in Subsec. 5.4 of the main text. The second cumulant $c_2^{\text{part}}$ provides a nontrivial test of the BMFT predictions as it probes fluctuations of the time-integrated particle current $\hat{J}(T)$ beyond the mean value, described, instead, by the first cumulant $c_1^{\text{part}}$. One expects, in particular, higher cumulants to be more sensible to rare events with very rapidly moving rods. As a consequence, for the numerical calculation of $c_2^{\text{part}}$, we exploit a large initial size $L_{\text{size}} = 10^5$ in order to be able to observe the Euler-scale predictions from BMFT before the boundary effects due to the aforementioned depletion zones become visible. The cumulant is numerically evaluated by computing the number $N_+$ and $N_-$ of rods on $x > 0$ and $x < 0$, respectively. We do this both at the initial time 0, getting $N_\pm(0)$ and at a variable observation time $T$, with result $N_\pm(T)$. The numerical value of $c_2^{\text{part}}$ is then obtained by computing the connected average of the transferred particle charge $\Delta N(T)$ and rescaling it by the time duration $T > 0$:

$$c_2^{\text{part}}(T) = T^{-1} \left\langle \Delta N(T)^2 \right\rangle^{\text{c}} = T^{-1}(\langle \Delta N(T)^2 \rangle - \langle \Delta N(T) \rangle^2), \tag{245}$$

with

$$\Delta N(T) = \frac{N_+(T) - N_-(T)}{2} - \frac{N_+(0) - N_-(0)}{2}. \tag{246}$$

Because of the continuity equation (6), it is immediate to verify that $\hat{J}(T) = \Delta N(T)$ and therefore the expression for $c_2^{\text{part}}(\text{T})$ in Eq. (245) coincides with the one given in Eq. (15) of the main text for $c_2(T)$ in the long time limit $T \to \infty$. The averages $\langle \bullet \rangle$ in Eq. (245) are done with respect to the initial partitioning inhomogeneous and non-stationary state in Eqs. (240). In the numerical simulations, averages $\langle \bullet \rangle$ of an observable $A$ are computed as the sample mean $\bar{A}$ over the independent realizations $A^{(i)}$, with $i = 1, 2 \ldots M$, of the initial rods' distribution as

$$\bar{A} = \frac{1}{M} \sum_{i=1}^{M} A^{(i)}. \tag{247}$$

This applies, for instance, both to $A_1(T) = \Delta N(T)$ and $A_2(T) = \Delta N(T)^2$ in Eq. (245). The resulting expression obtained for $c_2^{\text{part}}(T)$ is plotted in Fig. 3 of the main text. The statistical uncertainty $U(A)$ on the sample mean $\bar{A}$ is computed as the empirical standard deviation [116]

$$A = \bar{A} \pm U(A), \quad U(A) = \frac{1}{\sqrt{M}}\sqrt{\frac{1}{M-1}\sum_{i=1}^{M}(A^{(i)} - \bar{A})^2}, \tag{248}$$

which quantifies the dispersion of the samples $A_i$ around the sample mean $\bar{A}$. Note that the empirical standard deviation $U(A)$ drops as $1/\sqrt{M}$ as $M$ increases as a consequence of the central limit. For the calculation of the uncertainty on $c_2^{\text{part}}(T)$, we use the model for the propagation of uncertainties, see, e.g., Ref. [116], which assign to $c_2^{\text{part}}(T)$ the uncertainty $U(c_2^{\text{part}}(T))$ determined from the uncertainties $U(A_{1,2})$ as follows

$$U(c_2^{\text{part}}(T)) = \sum_{i=1}^{N}\left|\frac{\partial c_2^{\text{part}}(T)}{\partial A_i}\right|_{A_i = \bar{A}_i} U(A_i) = \frac{1}{T}(U(A_2) + 2\bar{A}_1 U(A_1)). \tag{249}$$

The previous equation has been used for the calculation of the error bars in Fig. 3, where $M = 8.8 \cdot 10^7$ samples have been used. Note that Eq. (249) leads to an excess estimation of $U(c_2^{\text{part}}(T))$ since the contributions from the uncertainties $U(A_{1,2})$ are summed in absolute value and therefore the possibility of a partial compensation of the the uncertainties $U(A_{1,2})$ is a priori excluded (cf., the discussion in Ref. [116]). The error bars in Fig. 3, which affect the third decimal digit, thereby show an excellent agreement between the numerical result for $c_2^{\text{part}}$ at long times and the BMFT prediction of Eq. (131), with the discrepancy between the two visible only on the fourth decimal digit and utterly within the error bars.

We conclude by reporting the formulas used for the calculation of the uncertainties'bars in Fig. (1) of Ref. [24] and in Fig. 4. The equal-time two-point correlator $S_{\hat{q}_0,\hat{q}_0}(x,t;0,t)$ (and analogously for $S_{\hat{q}_0,\hat{q}_0}(x,t;-x,t)$) is numerically obtained from Eq. (22) particularized the density conserved charge $\hat{q}_0$

$$S_{\hat{q}_0,\hat{q}_0}(x,t;0,t) = \lim_{\ell \to \infty} \ell \, \langle \overline{q_0}(\ell x, \ell t)\overline{q_0}(0,\ell t)\rangle_{\ell}^{\text{c}}, \tag{250}$$

with

$$\langle \overline{q_0}(x,t)\overline{q_0}(0,t)\rangle_{\ell}^{\text{c}} = \langle \overline{q_0}(x,t)\overline{q_0}(0,t)\rangle_{\ell} - \langle \overline{q_0}(x,t)\rangle_{\ell}\langle \overline{q_0}(0,t)\rangle_{\ell}. \tag{251}$$

In the numerical simulations, the fluid cell mean in Eq. (251) is implemented as in Eq. (35), thereby averaging only in space (see the discussion after Eq. (23) in the main text). The terms $C_2((x,t)) = \langle \overline{q_0}(x,t)\overline{q_0}(0,t)\rangle_{\ell}$, $C_1(x,t) = \langle \overline{q_0}(x,t)\rangle_{\ell}$ and $C_1(0,t) = \langle \overline{q_0}(0,t)\rangle_{\ell}$ appearing on the r.h.s. of Eq. (251) are evaluated in the numerical simulations by sample mean, as per Eq. (247). The statistical uncertainty $U(S_{\hat{q}_0,\hat{q}_0}(x,t;0,t))$ of the connected correlator in Eq. (250) is obtained from the uncertainties $U(C_2(x,t))$, $U(C_1(x,t))$ and $U(C_1(0,t))$ as

$$\begin{aligned}
U(S_{\hat{q}_0,\hat{q}_0}) &= \left|\frac{\partial U(S_{\hat{q}_0,\hat{q}_0})}{\partial C_2(x,t)}\right|_{\bar{C}_2(x,t)} U(C_2(x,t)) \\
&+ \left|\frac{\partial U(S_{\hat{q}_0,\hat{q}_0})}{\partial C_1(x,t)}\right|_{\bar{C}_1(x,t)} U(C_1(x,t)) \\
&+ \left|\frac{\partial U(S_{\hat{q}_0,\hat{q}_0})}{\partial C_1(0,t)}\right|_{\bar{C}_1(0,t)} U(C_1(0,t)).
\end{aligned} \tag{252}$$

In the previous equation we dropped the arguments of $S_{\hat{q}_0,\hat{q}_0}(x,t;0,t)$ for the sake of brevity. From Eqs. (250) and (251) one has

$$U(S_{\hat{q}_0,\hat{q}_0}(x,t;0,t)) = \ell\left(U(C_2(x,t)) + \bar{C}_1(0,t)U(C_1(x,t)) + \bar{C}_1(x,t)U(C_1(0,t))\right). \qquad (253)$$

The Euler-scaling limit in Eq. (250), which requires infinite variation lengths $\ell \to \infty$, is taken by considering large values of $\ell = 250, 500$ and $1000$. The previous equation, with the aforementioned values for the macroscopic length scale $\ell$, has been used in Fig. (1) of Ref. [24] and in 4 to compute the uncertainties'bars.

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
