# Peer review of "Ballistic macroscopic fluctuation theory"

_SciPost Physics_

## Round 3 · Referee Report · Anonymous (Referee 1) · 2023-2-3

Report

The paper develops a formalism for describing macroscopic fluctuations for the hydrodynamics of an arbitrary number of conserved quantities which possess ballistic modes. The main output of the approach is a formal setup for computing the full counting statistics of charges and currents, and dynamical correlation functions, both in the presence of a slowly space-dependent initial Gibbs ensemble. These quantities are then computed explicitly in the case of integrable systems.

Determining the general structures governing the emergent dynamics of many-body systems, classical and quantum, is presently a very active area of research. The paper is quite well-written and certainly contains results that can lead to further research on the topic. I therefore recommend the paper for publication, after the following remarks/questions have been addressed:

  1. This formalism strongly resembles the MSR approach to fluctuating hydrodynamics, which also uses an action formulation. What are the differences between this and the MSR approach, and what are the advantages of the present framework?

  2. The presence of non-local correlations in the Euler scaling limit is quite intriguing. Three- and higher-point functions should include the same type of information. Is the advantage of the BMFT in that it allows to infer this scaling somehow more systematically? Also, how are diffusive/noisy corrections expected to alter the result?

  3. The authors mention that non-local correlations require the presence of multiple conserved quantities. I'm confused by this statement: one can have a ballistic hydrodynamic mode with only a single conserved charge (at the cost of breaking time reversal), wouldn't such systems also possess non-local correlations in the same scaling limit?

Finally, there is a minor typo on p. 29 above eq. (84): "T(o1)T(o1)" -> "T(o1)T(o2)".

---

## Editorial Decision

resubmitted